# MGAT1-Guided complex N-Glycans on CD73 regulate immune evasion in triple-negative breast cancer

Junlong Jack Chi[1,2,3,4,17], Ping Xie[5,17], Mary Hongying Cheng [6,17], Yueming Zhu[1,2], Xin Cui[1,2], Joshua Watson [7], Lidan Zeng [1,2], Amad Uddin[1,2], Hoang Nguyen [6], Lei Li [8], Kelley Moremen [9], April Reedy [10], Megan Wyatt [2,11], Adam Marcus[2,10], Mingji Dai [2,12], Chrystal M. Paulos [2,11], Massimo Cristofanilli[13], William J. Gradishar [5], Shaying Zhao [7], Kevin Kalinsky [2,10], Mine-Chie Hung [14,15], Ivet Bahar [6,16] ✉, Bin Zhang [5] ✉ & Yong Wan[1,2,10] ✉

Despite the widespread application of immunotherapy, treating immune-cold tumors remains a significant challenge in cancer therapy. Using multiomic spatial analyses and experimental validation, we identify MGAT1, a glycosyl-transferase, as a pivotal factor governing tumor immune response. Over-expression of MGAT1 leads to immune evasion due to aberrant elevation of CD73 membrane translocation, which suppresses CD8[+] T cell function, especially in immune-cold triple-negative breast cancer (TNBC). Mechanistically, addition of N-acetylglucosamine to CD73 by MGAT1 enables the CD73 dimerization necessary for CD73 loading onto VAMP3, ensuring membrane fusion. We further show that THBS1 is an upstream etiological factor orchestrating the MGAT1-CD73-VAMP3-adenosine axis in suppressing CD8[+] T cell antitumor activity. Spatial transcriptomic profiling reveals spatially resolved features of interacting malignant and immune cells pertaining to expression levels of MGAT1 and CD73. In preclinical models of TNBC, W-GTF01, an inhibitor specifically blocked the MGAT1-catalyzed CD73 glycosylation, sensitizing refractory tumors to anti-PD-L1 therapy via restoring capacity to elicit a CD8[+] IFNγ-producing T cell response. Collectively, our findings uncover a strategy for targeting the immunosuppressive molecule CD73 by inhibiting MGAT1.

Despite the benefits of immunotherapy in treating certain types of cancers, the variability in patient response still remains[1,2]. Triple-negative breast cancer (TNBC) is among the most aggressive subtypes of breast cancer, primarily due to the lack of effective targeted therapies and limited treatment options beyond chemotherapy. Unlike HER2-positive breast cancer, which is also highly aggressive but benefits from targeted therapies like Trastuzumab deruxtecan (T-DXd) that have significantly improved outcomes, TNBC remains a major clinical challenge due to its heterogeneity, high recurrence rates, and poor prognosis. Importantly,

TNBC is the only breast cancer subtype for which immune checkpoint inhibitors have been approved, a distinction linked to the relatively higher immune infiltration in TNBC compared to other breast cancer subtypes, which are predominantly immune-cold[3,4]. However, even in TNBC, many tumors exhibit resistance to immunotherapy, driven by mechanisms of immune evasion and tumor-mediated immunosuppression. The immunotherapy resistance signature in breast cancer arises from several suppressive mechanisms, including lack of antigen, low T cell infiltration, and expression of critical immune checkpoints[5,6].

Our studies using multiomics analyses and experimental validation to identify targets for improving anticancer immune responses have drawn our attention to MGAT1.

MGAT1, mannosyl (alpha-1,3-)-glycoprotein beta-1,2-N-acetylglucosaminyltransferase, is a glycosyltransferase essential in the synthesis and maturation of complex N-glycans. The addition of N-acetylglucosamine (GlcNAc) to α-1,3-linked mannose of the substrate catalyzed by MGAT1 ensures the formation of complex glycans on target proteins, resulting in the orchestration of various cellular processes such as protein folding, protein translocation, and cell-cell communication[7–10]. Recent research has pinpointed the impact of MGAT1 on cancer progression, where elevated MGAT1 expression in cancer cells leads to increased branching of N-linked glycan structures on cell surface glycoproteins, facilitating a number of tumorigenic processes, including enhanced cell adhesion, migration, and invasion[11–13]. Dysregulated protein glycosylation was clearly shown to cause defects in tumor immunity and suppression of the tumor immune response, but the clinical relevance of MGAT1 in breast cancer immune evasion remains unknown. While previous reports have consistently demonstrated that silencing MGAT1 can impede cancer cell progression and metastasis[14–16], the absence of MGAT1 inhibitors underscores the tremendous untapped therapeutic potential of MGAT1.

CD73, also called ecto-5′-nucleotidase (NT5E), a membrane-bound enzyme, acts in association with CD39 in breaking down extracellular ATP to immunosuppressive adenosine[17,18]. Adenosinergic signaling regulates tumor immunity, suppressing cytotoxic T cells and creating an immunosuppressive tumor microenvironment[19,20]. It has been demonstrated that the oncogenic role of CD73 in advancing tumor progression involves interacting with cancer-associated fibroblasts through adenosine receptors (A1R, A2AR, A2BR, and A3R) on various types of immune cells such as regulatory (Foxp3+) T cells (Tregs), effector T cells, natural killer (NK) cells, myeloid-derived suppressor cells (MDSCs), B cells, and macrophages[21,22]. HIF-1α, estrogen receptor and certain inflammation factors have been linked to the regulation of CD73 in a transcriptional manner[1,23]. Recent studies showed CD73 to be a fast-turnover protein whose abundance is governed by the interplay between ubiquitin E3 ligase TRIM21 and deubiquitinase OTUD4[22,24,25]. Previous work characterized CD73 as an N-glycosylated protein. However, the upstream glycosyltransferase(s) governing CD73 glycosylation and the biochemical or cell biological consequences of CD73 glycosylation in relation to its immunosuppressive function remain largely unknown.

In the present study, we have identified a crucial role of MGAT1 in regulating tumor immune responses, with the demonstration of its clinical relevance in breast tumor immune evasion. We show that overexpression of MGAT1 leads to immune evasion through uncontrolled membrane trafficking and translocation of CD73, resulting in elevation of adenosine production. We mechanistically demonstrate that the addition of GlcNAc to CD73 at the Golgi apparatus by MGAT1 enables CD73 dimerization and ensures its membrane translocation. We further reveal that the molecular MGAT1-CD73-adenosine axis is regulated in response to THBS1 (thrombospondin-1). Clinically, we found that the MGAT1low/CD73low signature in a subset of human breast malignancies was associated with a favorable immune profile. Finally, we developed a pharmacological inhibitor of MGAT1, W-GTF01, that balances CD73 dimerization and restores CD8+ effector T cell responses, sensitizing anti-PD-L1 therapy in preclinical mouse models of TNBC. Our findings uncover a strategy for targeting immunosuppressive CD73 in treating immune-cold breast cancers.

## Results

### Accumulation of MGAT1, a glycosyltransferase, is associated with an unfavorable tumor immune response and prognosis in immune-cold breast cancers

To search for potential therapeutic targets for immune-cold breast cancer, we performed a hierarchical clustering on the gene expression data of 112 immune-relevant proteins from 935 breast cancer patients in The Cancer Genome Atlas (TCGA). Our analysis identified elevated levels of immune-relevant proteins, including VTCN1, CD274, and CD73, specifically within the basal-like subtype of breast cancer (BLBC), compared to other subtypes (Fig. 1a and Supplementary Fig. 1a). In addition, hierarchical clustering analysis suggested two distinct subgroups within TNBC patients, each characterized by a unique immune-related protein expression profile, as shown in Fig. 1a. One subgroup, highlighted in the dotted frame, exhibited a significantly higher expression of immune-related gene signatures compared to the rest of the TNBC samples. Further investigation via differential enriched pathway analysis, employing single-sample gene set enrichment analysis (ssGSEA), indicated that one subgroup of TNBC patients (red) exhibited an increased negative regulation of tumor immunity pathways (Fig. 1b). This regulation coincided with a distinct molecular signature of enhanced N-glycan processing and the prevalence of N-glycosylated proteins (Fig. 1c). To further validate these findings, we conducted the similar analysis to other breast cancer subtypes including LumA, LumB, Her2+, and Normal-like. As shown in Supplementary Fig. 1b−e, the data revealed that within these non-TNBC subtypes, a small subset of patients exhibited increased expression of immune-related genes, resembling the immune-cold subpopulation observed in TNBC. Interestingly, these samples also demonstrated significant enrichment in pathways associated with the negative regulation of the immune system, indicating the presence of a similar immunosuppressive environment. However, when analyzing glycosylation and glycan processing pathways, we found no significant differences compared to their main groups within the non-TNBC cohort (Supplementary Fig. 1b−e). Given the critical role of aberrant N-glycosylation in regulating tumor immunity[6,26,27], we undertook Pearson correlation analysis using a curated list of immune-related genes obtained from MSigDB[28] to identify if specific N-glycan processing-related genes are implicated in this particular TNBC subgroup. As shown in Fig. 1d, the results clearly showed that multiple enzymes involved in N-glycosylation correlate positively with genes driving immune suppression, and the detailed gene list is included in Supplementary Fig. 2a. Notably, MGAT1 emerged as the most significantly associated enzyme, topping the list of genes linked to this effect within the TNBC subgroup (Supplementary Fig. 2a).

To validate the findings depicted in Fig. 1a−d, immunohistochemical analysis was conducted on a tissue microarray (TMA) containing 224 unique breast tissue samples. We demonstrated a significant elevation of MGAT1 protein levels in the majority of TNBC specimens (Fig. 1e). Further systematic protein expression analysis across an extensive array of breast cancer cell lines confirmed that MGAT1 protein was prevalent in HER2+ and TNBC cell lines, reinforcing the potential link to immune escape mechanisms in TNBC (Fig. 1f and Supplementary Fig. 2b). We then took an unbiased approach to delve into MGAT1 protein expression within tumor cells and its correlation with CD8+ T cell presence, employing multiplex immunohistochemistry (Fig. 1g). Spatial analysis unveiled heightened interactions between CD8+ T cells and MGAT1lo tumor cells, as well as between proliferating CD8+Ki67+ T cells and MGAT1lo tumor cells, compared to interactions with MGAT1hi tumor cells (Fig. 1h). These findings strongly support the notion that lower levels of MGAT1 in tumors are linked to enhanced infiltration of CD8+ T cells. We further conducted a Gene Ontology Biological Process analysis using the TCGA breast cancer database. Supplementary Fig. 2c, d illustrates a significant correlation between MGAT1 expression and critical aspects of the adaptive immune response, including T cell activation and their antitumor functions. An increase in MGAT1 expression was correlated with the down-regulation of immune response, T cell activation, and proliferation. Furthermore, the results from immunohistochemical analysis with the TMA demonstrated a significant negative correlation of MGAT1 protein levels with the CD8+ T cell infiltration in TNBC patient

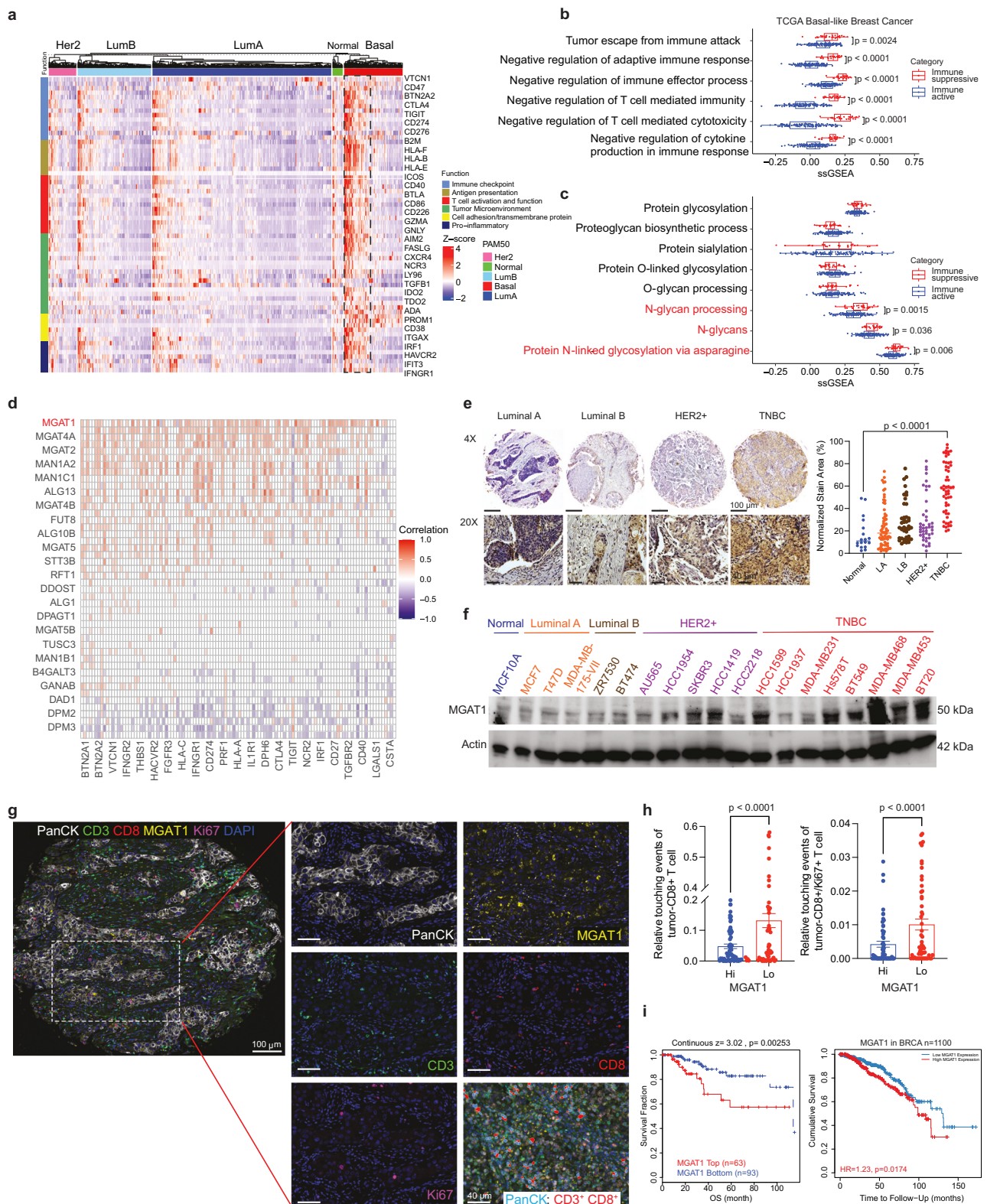

specimens (Supplementary Fig. 2e). In addition, data derived from TIDE, a computational model designed to assess gene correlations with T cell dysfunction, identified MGAT1 as a top-ranking N-glycan biosynthesis gene associated with a T cell dysfunction score (Supplementary Fig. 2f).

To further determine the prognostic value of MGAT1 expression in TNBC patients, we analyzed a cohort of 156 TNBC samples from the TCGA database. We observed a strong correlation

between elevated MGAT1 expression and negative overall survival outcomes (Fig. 1i). Moreover when evaluating cumulative survival with respect to both MGAT1 expression levels and the CD8+ T cell population, it was evident that patients with a high CD8+ T cell presence coupled with reduced MGAT1 expression exhibited a significant survival advantage and patients with low CD8+ T cell and high MGAT1 expression were associated with worse prognosis (Supplementary Fig. 2g). This data compellingly links increased

**Fig. 1 | Accumulation of MGAT1, a glycosyltransferase, is dramatically associated with an unfavorable tumor immune response and prognosis in immune-cold breast cancers. a** Heatmap showing hierarchical clustering of row-scaled breast cancer gene expression data from TCGA for a subset of immune-related genes differentially expressed in four breast cancer types. Each column represents a sample; each row represents a protein. **b, c** Distribution of single sample GSEA scores of BLBC separated by immune subgroup (immune suppressive subgroup, $n = 28$; immune active subgroup, $n = 125$), using gene sets involved in immune functions (**b**) and glycosylation (**c**). *P*-values are from two-sided Wilcoxon tests. Box plots are defined as follows: center line indicates the median (middle line), box bounds are 25th and 75th percentile, and whiskers extend to 1.5*interquartile range below the 25th percentile (minima) and above the 75th percentile (maxima), with outliers outside of that range shown. **d** Pearson correlation plot showing correlations between N-glycosylation genes (*y*-axis) and immune-related genes (*x*-axis). N-glycosylation genes are ordered from top to bottom by the number of significant positive correlations. White cells indicate no statistically significant correlation. **e** Breast cancer TMA stained with anti-MGAT1 antibody and representative pictures of different breast cancer subtypes are shown. Right: quantified TMA consisting of adjacent normal breast tissue ($n = 18$), Luminal A ($n = 68$), Luminal B ($n = 41$), HER2 + ($n = 43$), and TNBC ($n = 54$) samples immunostained for MGAT1. **f** Expression of MGAT1 in normal human mammary epithelial cells and various subtypes of breast cancer cells detected by immunoblotting using anti-MGAT1 antibody. **g** Representative composite image (left panel) with inset (right) of a breast cancer specimen ($n = 41$) stained by multicolor IHC comprising CD8, Ki67, CD3, MGAT1, PanCK, and DAPI. Scale bars: 100 μm. **h** Touching events between CD3$^+$CD8$^+$ T cells and PanCK$^+$ tumor cells, or CD3$^+$CD8$^+$Ki67$^+$ T cells and PanCK$^+$ tumor cells among MGAT1hi relative to MGAT1lo tumor regions ($n = 44$) were measured with R-based PhenoptrReports & Phenoptr. **i** Kaplan-Meier curves of TNBC and BC patients corresponding MGAT1 expression. The high MGAT1 expression leads to poor prognosis in both TNBC and BC patients. Data (represented as means ± SEM), images, and western blots are representative of three independent experiments. Statistical significance was determined using two-sided Wilcoxon tests (**b, c**), one-way ANOVA with Tukey's multiple comparisons test (**e**) or two-tailed paired t test (**h**), or two-sided logrank test (**i**). Source data are provided as a Source Data file.

MGAT1 expression to a worse prognosis in the context of immunosuppressive TNBC.

## MGAT1 regulates CD8$^+$ T cell function in 2D and 3D tumor/immune cell coculture systems

To determine the impact of MGAT1 in regulating tumor immune response, we have performed 2D coculture analyses with TNBC breast cancer cells and pre-activated peripheral blood mononuclear cells (PBMCs), followed by 3D coculture analysis using TNBC tumor spheroids and PBMCs, as illustrated in Fig. 2a[24]. To this end, we established Flag-tagged MGAT1 overexpression (OE) and MGAT1 stable knockdown (KD) in MDA-MB-468 cells based on the lentivirus and CRISPR/Cas9 system[22,24] and the stable cell lines were pre-stained with CellTracker Green CMFDA (ThermoFisher Scientific), a fluorescent dye stable for over 72 h. The relative survival of cancer cells in the presence of immune cell-mediated cytotoxicity was quantified based on the coverage of stained cells. After 2D coculture, the impact of MGAT1 in regulating CD8$^+$ T cell function that, in turn, influenced the cancer cell survival was measured by flow cytometry. As shown in Fig. 2b, MGAT1 OE in tumor cells conferred resistance to immune cell-mediated cell death, whereas tumor cells with KD of MGAT1 were more vulnerable to immune cell-mediated killing. Simultaneously, the flow cytometric evaluation of immune cells indicated that the up-regulation of MGAT1 in cancer cells dramatically inhibited the expression of TNFα, IFNγ, and Ki-67 in CD8$^+$ T cells, while depletion of MGAT1 dramatically enriched the expression of TNFα, IFNγ, Granzyme B, and Ki-67, indicating a potent role for MGAT1 in modulating T cell-mediated antitumor function in vitro (Fig. 2c). Similar results were observed in MDA-MB231 MGAT1-OE/KD stable lines (Supplementary Fig. 3a, b). Because of the advantage of a 3D coculture system in mimicking the tumor microenvironment that represents a more physiological scenario, we established tumor spheroids based on TNBC cells and cocultured them with stained PBMCs. The infiltration of stained immune cells in the 3D cancer cell spheroids was visualized with Z-stack imaging followed by Z-projection. Intriguingly, ablation of MGAT1 significantly boosted immune cell infiltration into MGAT1-KD TNBC spheroids, whereas elevated expression of MGAT1 in tumor cells drastically suppressed the infiltration of immune cells in the MGAT1-OE TNBC spheroids (Fig. 2d). Furthermore, flow cytometry analysis further revealed similar immune suppression of T cells cocultured with MGAT1-OE tumor cells and hyperactivation of T cell activities with MGAT1-KD tumor cells (Fig. 2e). Consistent with the PBMC coculture system, the coculture of MGAT1 stable cell lines with purified and pre-activated CD8$^+$ T cells leads to a similar inhibition on CD8$^+$ T cell activation (Fig. 2f, g). Collectively, these results indicate that the glycosyltransferase MGAT1 acts as a crucial negative regulator of CD8$^+$ T cells in immune-cold

TNBC, whose accumulation could contribute to tumor immune evasion through suppressing cytotoxic IFNγ-secreting CD8$^+$ T cell function.

## Identification of CD73, an immune checkpoint protein, as a putative substrate of MGAT1 that mediates MGAT1-initiated immune suppression

To identify the downstream substrates of MGAT1 that elicit immune suppression, we conducted tandem affinity purification coupled with mass spectrometry to isolate the MGAT1 interactome. Stable Flag-HA-tagged hMGAT1 protein was expressed in MDA-MB468 cells, and the MGAT1 protein complex was then purified by affinity capture in parallel with control cells, followed by mass spectrometry analysis (Fig. 3a and Supplementary Fig. 4a). The interactome of MGAT1, resolved through mass spectrometry, identified several tumor immune-responsive proteins, including antigen-presenting proteins such as HLA-A and HLA-B, as well as CD73 (Supplementary Data 1). Among the potential MGAT1 binding partners, our analysis identified several antigen-presenting proteins (e.g., HLA-A and HLA-B) but only one key immune-suppressive molecule—CD73. Given its well-established role in adenosine-mediated immunosuppression and tumor immune evasion, we prioritized CD73 as the primary focus for mechanistic investigation. To further decipher the biochemical consequences as well as the physiological relevance of the interaction between MGAT1 and CD73, we performed a series of immunoprecipitation analyses and immunostaining for colocalization. As shown in Fig. 3b, c and Supplementary Fig. 4b, c, we observed that endogenous CD73 co-immunoprecipitated with MGAT1, and MGAT1 co-immunoprecipitated with CD73. To examine the cellular compartmentalization of the interaction between MGAT1 and CD73, we employed multiple complementary approaches. First, we conducted immunofluorescence staining to co-stain MGAT1 and CD73 followed by imaging with confocal imaging. We observed that MGAT1 (green) colocalized with CD73 (red) in the Golgi body verified by GM-130, a well-established Golgi body indicator[29] (Fig. 3d, e). In addition, super-resolution microscopy further validated the colocalization of MGAT1 and CD73, providing superior spatial resolution and reinforcing the findings observed in confocal imaging (Fig. 3f). We also conducted a proximity ligation assay to corroborate the colocalization of MGAT1 and CD73 in the Golgi compartment (Fig. 3g). To complement the imaging data, we conducted subcellular fractionation to separate the Golgi and membrane compartments[30–32], followed by immunoblotting. The results revealed that both MGAT1 and CD73 were enriched in the Golgi fraction, supporting their co-distribution within this compartment (Supplementary Fig. 4d). Together, these approaches demonstrate that MGAT1 and CD73 colocalize in the Golgi apparatus, where MGAT1 likely mediates CD73 glycosylation to facilitate its functional

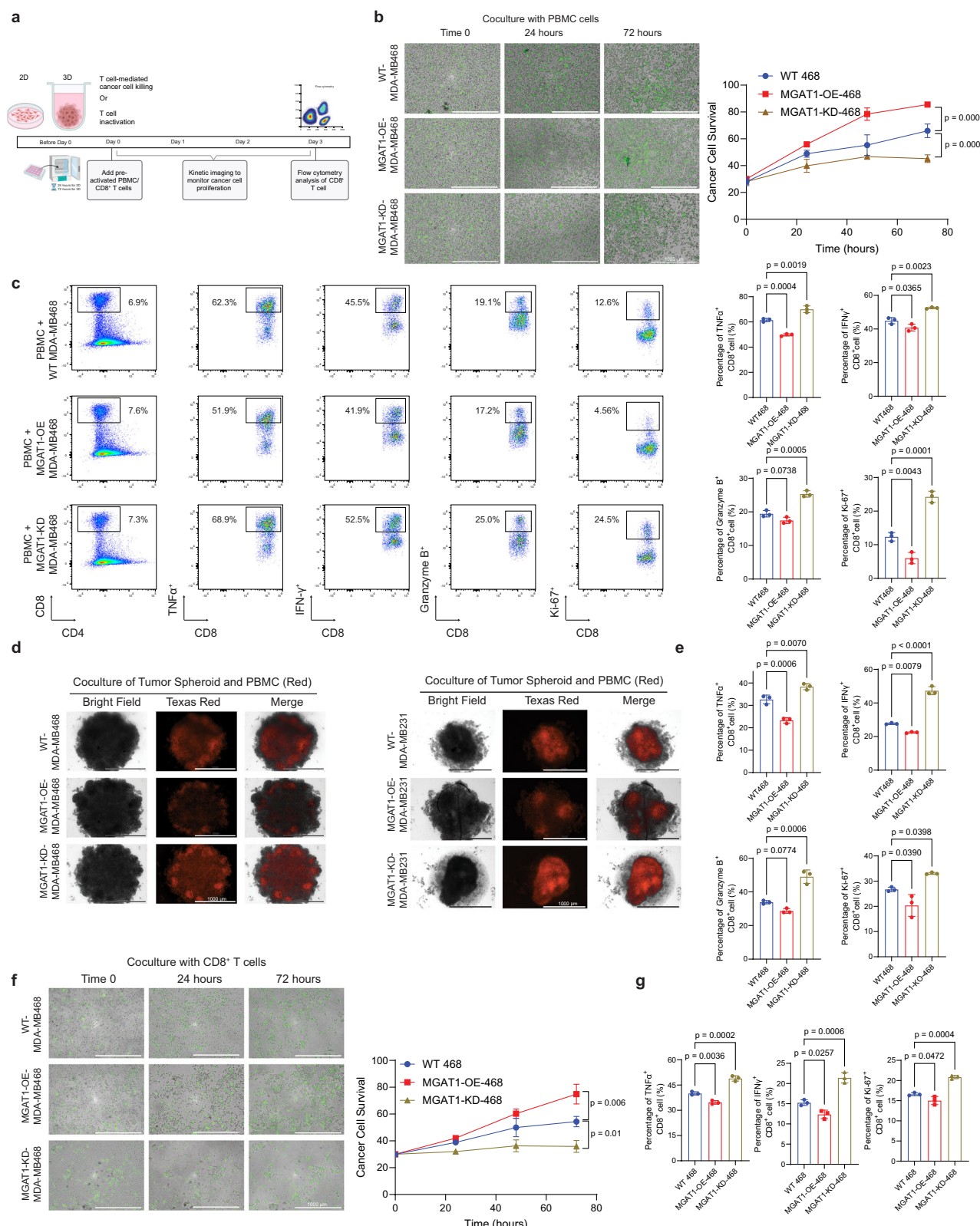

maturation. To ascertain the physiological relevance of the CD73 and MGAT1 colocalization, we measured adenosine production with an adenosine assay kit in both human TNBC breast cancer cells (MDA-MB231 and MDA-MB468) in response to altered MGAT1 expression. As shown in Fig. 3h, OE of MGAT1 in breast cancer cells significantly increased adenosine production, whereas MGAT1 KD resulted in a reduction of adenosine production. To investigate the dependency of

MGAT1's role in adenosine synthesis on CD73, we engineered MGAT1 overexpression and knockdown in CD73-KD stable lines. The results revealed that in the absence of CD73, MGAT1 overexpression or knockdown did not affect adenosine levels (Fig. 3h). These findings demonstrate that MGAT1's influence on adenosine synthesis is mediated specifically through its regulation of CD73. Consistent with the above results, the big datasets with Spearman's rank correlation

**Fig. 2 | Elevated expression of MGAT1 inhibits tumor immune response in 2D and 3D in vitro coculture systems. a** Schematic diagram of the coculture assays. Created in BioRender. Zhu, Y. (2025) https://BioRender.com/v30i433. **b** The proliferation of MDA-MB468, MDA-MB468-MGAT1-OE, and MDA-MB468-MGAT1-KD cells (green) in 2D coculture with pre-activated PBMCs was monitored by time-lapse image-based quantification. The data are presented as mean ± SEM from three replicates from a representative experiment. **c** MDA-MB468, MDA-MB468-MGAT1-OE, and MDA-MB468-MGAT1-KD, and MDA-MB231, MDA-MB231-MGAT1-OE, and MDA-MB231-MGAT1-KD cells were cocultured with pre-activated human PBMCs, and IFNγ⁺, TNFα⁺, Granzyme B⁺, Ki-67⁺ CD8⁺ T cell populations were measured and quantified using flow cytometry. **d** The infiltration of PBMCs (red) in 3D spheroids constructed with MDA-MB468, MDA-MB468-MGAT1-OE, and MDA-MB468-MGAT1-KD or MDA-MB231, MDA-MB231-MGAT1-OE, and MDA-MB231-MGAT1-KD was visualized with Z-stack imaging using a Lionheart microscope. **e** The IFNγ⁺, TNFα⁺, Granzyme B⁺, and Ki-67⁺ CD8⁺ T cell populations from 3D coculture with MDA-MB468, MDA-MB468-MGAT1-OE, and MDA-MB468-MGAT1-KD cells were measured and quantified using flow cytometry. **f** The proliferation of MGAT1-WT/OE/KD MDA-MB468 cells (green) in 2D coculture with pre-activated purified CD8⁺ T cells was monitored by time-lapse image-based quantification. The data are presented as mean ± SEM from three replicates from a representative experiment. **g** The IFNγ⁺, TNFα⁺, and Ki-67⁺ CD8⁺ T cell populations from 2D coculture with MDA-MB468, MDA-MB468-MGAT1-OE, and MDA-MB468-MGAT1-KD cells were measured and quantified using flow cytometry. Data (represented as means ± SEM), images, and flow cytometry are representative of three independent experiments. Statistical significance was determined using one-way ANOVA with Tukey's multiple comparisons tests (**c**, **e**, **g**) or two-way ANOVAs followed by Tukey's multiple comparison tests (**b**, **f**). Source data are provided as a Source Data file.

analysis using proteomic datasets from the PDC000408 cohort[33] clearly confirmed the correlation between MGAT1 protein levels with several immune regulators, especially CD73 (NT5E) (Supplementary Fig. 4e). This result further showed a strong positive correlation between CD73 protein levels and several N-glycan biosynthesis genes, including MGAT1 (Fig. 3i). In addition, in a TNBC TMA ($n = 23$), elevated MGAT1 protein levels were positively correlated with overexpression of CD73 (Fig. 3j). Taken together, the above results suggest that CD73 is a substrate of MGAT1 in mediating MGAT1-initiated immune suppression.

## Mapping and structural modeling of the interaction between MGAT1 and CD73, and catalysis of CD73 glycosylation by integrated experimental data and docking simulations

To elucidate the in-depth mechanism by which CD73 is regulated by MGAT1, we identified the interaction domains that facilitate the recognition and binding between MGAT1 and CD73. We made Flag/HA-tagged CD73 truncation constructs and Flag/HA-tagged MGAT1 truncation constructs (Fig. 4a–d) and co-transfected them into human embryonic kidney (HEK) 293T cells, followed by interaction mapping using coimmunoprecipitation. Among the MGAT1 truncation constructs, comprising amino acids 220–445 was the smallest fragment retaining interaction with CD73 (Fig. 4b). Next, we engineered a series of MGAT1 deletions within this construct, specifically deleting the amino acid segments 220–270, 271–320, 321–370, and 371–445. As illustrated in Fig. 4b, the MGAT1 mutant without the 321–370 segment lost its ability to bind with CD73. A similar strategy was used to identify the molecular motifs on CD73 responsible for the interaction with MGAT1. As shown in Fig. 4d, amino acid residues from 79 to 128 on CD73 were pinpointed as the region mediating the binding between CD73 and MGAT1.

To investigate the possible binding conformations of MGAT1-CD73, we conducted a series of docking and molecular dynamics (MD) simulations. The crystal structure of rabbit N-acetylglucosaminyltransferase I (GnT I) (PDB: 2am5)[34], which shares 94% sequence identity with human MGAT1, was used for modeling using SwissModel, along with the crystal structure of a soluble form of human CD73 (PDB:4h1s)[35]. The structures predicted by SwissModel for MGAT1 and CD73 truncated forms were observed to be almost identically reproduced by AlphaFold3 (with respective RMSDs of 0.6 and 0.2 Å).

MGAT1 was docked onto the CD73 structure using protein-protein rigid docking software ClusPro, guided by experimental data (Fig. 4a, c) on the identified interfacial regions of the two proteins. The top ClusPro-predicted CD73-MGAT1 complex model, consistent with experimental data, is shown in Fig. 4e. Its estimated docking occupancy is 10.5%, and the binding energy is −15.4 kcal/mol. AlphaFold3 was also employed to predict the binding interface between CD73 and MGAT1 (Supplementary Fig. 5a). When compared with experiments, ClusPro seems to provide a better prediction (No.1 in Supplementary Fig. 5a) than AlphaFold3 (No. 2–6 in Supplementary Fig. 5a) where the

interacting interface observed in co-IP position closely in ClusPro predicted model.

In order to assess the stability of the predicted complex conformation, we performed five runs of 200 ns MD simulations for the top-ranked cluster predicted by ClusPro. The root-mean-square deviations (RMSDs) of the MGAT1-CD73 complexes were evaluated by structurally aligning the simulated complexes with the initial ClusPro-predicted model. A RMSD profile over 200 ns simulation indicated a stable binding of MGAT1 and CD73 in 4 out of 5 runs (Fig. 4f). The result in Fig. 4g illustrates the time evolution of the most persistent inter-residue interactions at the interface between MGAT1 and CD73 over the 200 ns time frame of the MD simulations. To further elucidate how MGAT1 mediates CD73 glycosylation, we conducted docking simulations with CD73 carrying Man5 at all four Asn residues (N333, N403, N53, and N311). CD73 glycoprotein was modeled using CHARMM-GUI[36]; the binding of MGAT1 to CD73 glycoprotein was predicted using ZDOCK server[37] since ZDOCK can predict glycosylated protein binding (Supplementary Fig. 5b). Result shows that all four asparagine residues can be recognized by MGAT1. Consistent with the molecular docking, we observed that truncation of the interaction domain on CD73 with MGAT1 dramatically decreased CD73-mediated adenosine production (Supplementary Fig. 5c).

Since our data suggests that MGAT1 interaction with CD73 relies on the critical interfacial residue regions, it is worth validating if the truncated MGAT1 and CD73 maintain their native strictures and conformation. We examined them in two ways: 1. by comparing the structural alignment of the intact and truncated proteins, and 2. by comparing the dynamics of the intact and truncated proteins. In the comparison of intact and truncated conformation, the structure of the truncated MGAT1 and CD73 (fragment of interest; residues 220–445 and 27–224, respectively) predicted by AlphaFold3 predictions is almost identical to that of WT (intact) MGAT1. For the smaller fragments, when superimposed on its wildtype counterpart, MGAT1 220–445 Δ321-370 yielded an RMSD of 0.249 Å, and CD73 Δ79-128 differed from the WT by an RMSD 0.855 Å, which is typically within the resolution of protein structures. In the second way, we carried out MD simulations up to 200 ns for the truncated proteins in explicit water to assess their stability as well as possible conformational change and refolding/collapsing of flexible regions, in comparison to the dynamics of the corresponding regions in the intact proteins. Results showed that MGAT1 fragments were stable within an RMSD of 3.5Å with respect to the starting conformation (Supplementary Fig. 5d). CD73 WT, CD73 1-224 and CD73 Δ79-128, on the other hand, showed departures of up to 9Å from the initial state (Supplementary Fig. 5e). Notably, this behavior was shared by the wildtype CD73 as well as its truncated fragments, pointing to the higher conformational flexibility of CD73 in general. Further examination of the initial and final conformations of CD73 WT, CD73 1-224, and CD73 Δ79-128 confirmed that the change in conformations during simulations maintained the overall fold, with major rearrangements originating from the reconfiguration of the

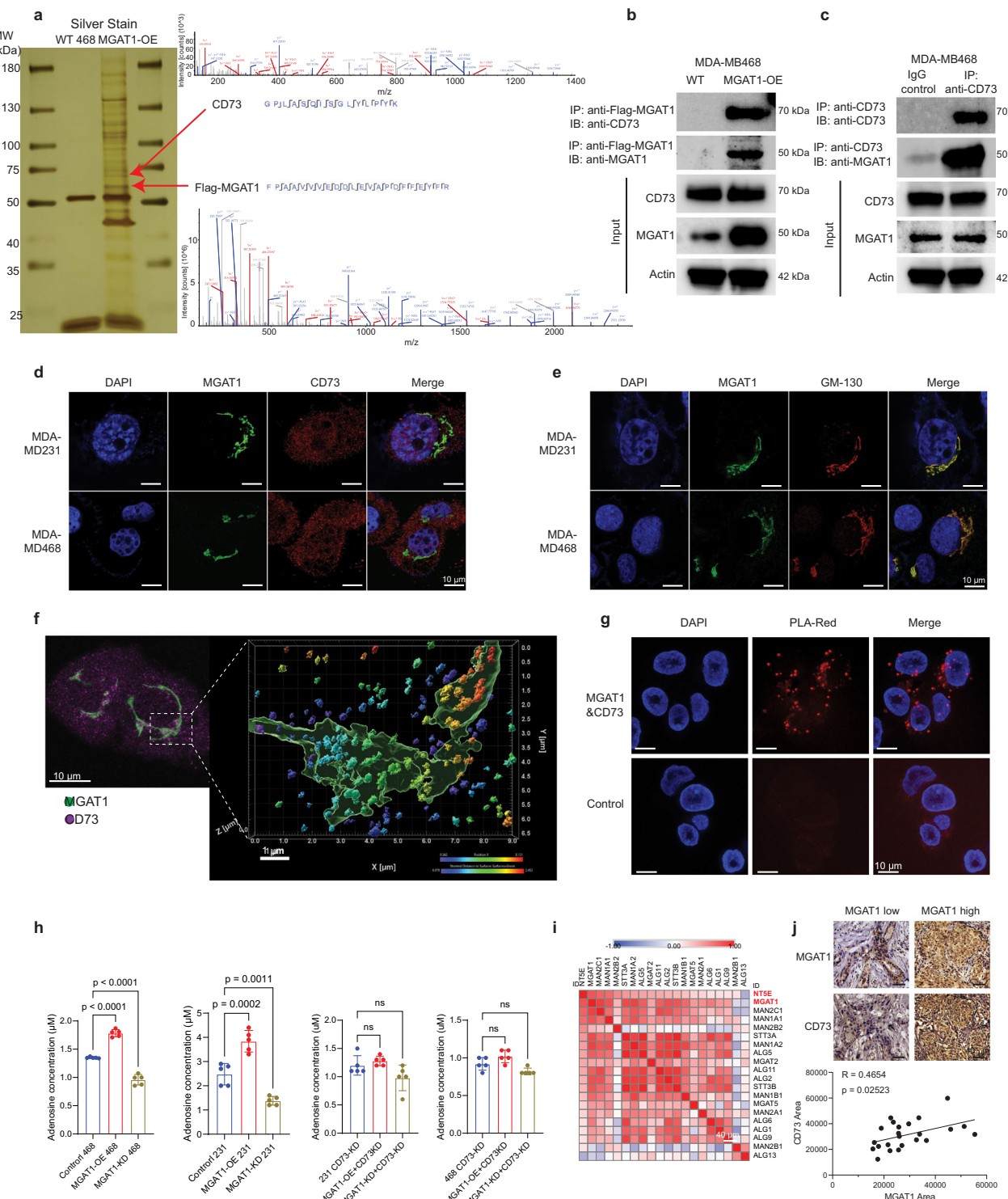

flexible (loop) regions as well as the relative position of the two domains.

## MGAT1 catalyzes CD73 glycosylation that, in turn, triggers CD73 dimerization, ensuring its translocation from the cytosol to the cell membrane

To further dissect the biochemical consequence of CD73 glycosylation by MGAT1, we systematically analyzed CD73 protein turnover, cellular localization, and interplay with other biochemical modifications such as ubiquitination. As described above, we initially engineered MGAT1 OE and KD in MDA-MB468 and MDA-MB231 cells. As shown in Fig. 5a, while

a minor change in CD73 abundance in response to MGAT1 OE was observed, reduced glycosylation of CD73 (running smaller molecular mass ~ 60 kDa) was clearly detected when MGAT1 was knocked down, confirming the aforementioned CD73 glycosylation catalyzed by MGAT1. To investigate the predominantly type of glycans on CD73, we further tested the effect of endoglycosidase H (specifically cleaving high mannose and hybrid N-glycan) and PNGase F (an N-glycosidase removing all N-linked glycan) on CD73 in TNBC cell lines. The CD73 band shift indicates that complex type N-glycan is the major type of asparagine-linked carbohydrate attached to CD73 (Supplementary Fig. 6a), but without sufficient MGAT1, a fraction of less glycosylated CD73 will be

**Fig. 3 | Identification of CD73, an immune checkpoint, as a critical MGAT1 substrate causing immune suppression in TNBC. a** MGAT1 complex was purified with a tandem-affinity purification protocol followed by mass spectrometry analysis in MDA-MB468-Flag/HA-MGAT1 cells. Silver staining of the purified MGAT1 complex is illustrated. CD73 was identified as a binding partner of MGAT1, and the representative spectra are shown. **b**, **c** The biochemical interaction between MGAT1 and CD73 in MDA-MB468 cells was validated by reciprocal coimmunoprecipitation of ectopic Flag-MGAT1 (**b**) and endogenous CD73 (**c**). The samples derived from the same experiment but different gels for CD73 MGAT1, and β-ACTIN were processed in parallel. **d** The colocalization of immunostained MGAT1 (green) and CD73 (red) in MDA-MB468 and MDA-MB231 cells was visualized by confocal imaging. **e** The subcellular localization of MGAT1 was detected through colocalization of MGAT1 (green) and Golgi indicator, GM-130 (red), in MDA-MB468 and MDA-MB231 cells by confocal imaging. **f** The intracellular interaction between MGAT1 (green) and CD73 (red) in MDA-MB468 cells was

visualized by immunofluorescence stimulated emission depletion microscopy imaging followed by 3D reconstruction by Imaris. **g** The intracellular interaction between MGAT1 and CD73 was validated by a proximity ligation assay (PLA-red) with anti-MGAT1 and anti-CD73 antibodies or control IgG followed by confocal imaging. **h** Adenosine levels were determined in WT or CD73-KD MDA-MB231/ MDA-MB468 breast cancer cells with MGAT1 OE or MGAT1 KD. **i** Spearman's rank correlation analysis shows that CD73 protein expression is highly positively correlated with several N-glycan biosynthesis genes, and MGAT1 is the most positively correlated one. **j** Representative IHC staining of MGAT1 and CD73 in human TNBC tissue sections ($n = 23$). Quantification of positive staining areas using QuPath reveals a significant positive correlation between MGAT1 and CD73 expression. Data (mean ± SEM), images, and western blots are representative of at least three independent experiments. Statistical significance was determined using one-way ANOVA with Tukey's multiple comparisons test (**h**) or simple linear regression (**j**). Source data are provided as a Source Data file.

covered with high mannose and hybrid N-glycan (Fig. 5b). CD73 owns four N-glycosylation motifs including [53]NAS, [311]NSS, [333]NYS, and [403]NGT[38]. To investigate whether all four N-glycan sites on CD73 are regulated by MGAT1, we transfected the triple-mutant CD73 construct into WT and MGAT1-KD HEK-293T cells and analyzed CD73 molecular size via Western blotting. The results demonstrated that glycosylation at all four sites requires the participation of MGAT1, as MGAT1 knockdown significantly reduced the CD73 glycosylation (Supplementary Fig. 6b). In addition, the triple-mutant CD73 retained partial glycosylation, further supporting the hypothesis that each of the four Asn sites on CD73 can independently be glycosylated by MGAT1.

To investigate the role of MGAT1 in regulating the membrane localization of CD73, we measured the membrane CD73 distribution in MGAT1 OE and KD stable lines, and the results (Fig. 5c and Supplementary Fig. 6c) demonstrated that MGAT1 overexpression leads to a significant increase in CD73 membrane localization, while MGAT1 knockdown reduces its presence on the membrane. The above result was further confirmed by confocal microscopy imaging using Biotium's CellBrite® NIR680 Cytoplasmic Membrane Dye, which specifically labels the cytoplasmic membranes of live or formaldehyde-fixed cells (Fig. 5d) and whole-cell lysate fractionation followed by immunoblot (Supplementary Fig. 6d).

In response to MGAT1-mediated CD73 modification, we observed a dramatic change in CD73 dimerization. We initially confirmed the dimerization of CD73 when the CD73 complex was stabilized in the presence of a crosslinking agent (Fig. 5e). The dimerization was further confirmed in several TNBC cell lines by running semi-native gels (Supplementary Fig. 6e), and the knockdown of MGAT1 dramatically decreased the CD73 dimerization (Fig. 5f and Supplementary Fig. 6f). The point mutations, D210N and R301W, on MGAT1 previously reported to reduce its enzymatic activity[39] also reduce the glycosylation of CD73, leading to decreased dimerization and adenosine production (Supplementary Fig. 6g, h), which provide direct evidence that MGAT1 enzymatic activity is critical for stabilizing CD73 dimerization. To further determine if branched N-glycans produced from MGAT1 could directly affect CD73 dimerization and adenosine production, we treated both cell lines with kifunensin and swainsonine to ensure efficient inhibition of N-glycan branching, and the results demonstrated the critical role of branched N-glycans in stabilizing CD73 dimerization and adenosine production (Supplementary Fig. 6i, j). While CD73 dimer was robustly measured for wild-type CD73 in native gels, replacement of Asn 53, 311, 333, and 403 by Gln (CD73-4NQ mutant) led to a significant decrease in CD73 dimerization on native PAGE (Supplementary Fig. 7a). We further identified the CD73 dimerization interface based on the resolved CD73 dimer in the open (PDB: 4H2G)[38] and closed state (PDB: 6TVX)[40] and thereafter engineered a dimerization-deficient mutant of CD73 (Fig. 5g). As shown in Supplementary Fig. 7a, the deletion of amino acids 480-537 on CD73 diminished CD73 dimerization as measured by native PAGE. Similarly,

MGAT1 KD dramatically decreased CD73 dimerization and increased the fraction of the CD73 monomer (Fig. 5f), confirming the biochemical role of MGAT1-mediated glycosylation in regulating CD73 dimerization.

To visualize the CD73 dimer in cells, we conducted split-GFP assays to validate the influence of glycosylation on CD73 dimerization. Plasmids of CD73 with GFP 1–10 or GFP11 × 7 extensions were engineered and co-transfected into HEK-293T cells (Fig. 5h), and the green fluorescence was visualized with a confocal microscope. As shown in Fig. 5i, we captured the strongest green fluorescence for WT CD73, while the signal from CD73-4NQ and dimer-deficient mutant CD73 was largely diminished. This observation was further confirmed by stably expressing CD73 with GFP 1−10 or GFP11 × 7 in MDA-MB231 and MD-MB468 cells (Supplementary Fig. 7b, c). To evaluate how MGAT1 expression affects the CD73 dimerization, the WT CD73 plasmids with GFP 1−10 or GFP 11 × 7 extensions were transformed into the MGAT1-WT/OE/KD MDA-MB468 stable lines, and the intensity of green fluorescence was measured by flow cytometry. As illustrated in Supplementary Fig. 7d, e, MGAT1 OE significantly enhanced the dimerization of CD73, whereas MGAT1 KD decreased the CD73 dimerization. In addition, the impact of MGAT1 on the total cellular glycan distribution was measured by flow cytometry with fluorophore-conjugated lectins. As shown in Supplementary Fig. 7f, g, MGAT1 OE dramatically increased the membrane complex type glycan recognized by PLA-H, while MGAT1 KD decreased the membrane distribution of complex glycan and sialic acid. To examine whether MGAT1 also influences the membrane distribution of other critical regulators in adenosine production, we measured membrane CD39 in MGAT1-WT, MGAT1-OE, and MGAT1-KD cell lines with flow cytometry and the results (Supplementary Fig. 7h) showed no significant alteration in CD39 membrane distribution across these conditions, indicating that MGAT1 does not play a prominent role in regulating CD39 glycosylation.

To elucidate the mechanism by which MGAT1-mediated CD73 glycosylation and dimerization orchestrates CD73 membrane translocation from the trans-Golgi network to the plasma membrane, we conducted mass spectrometry to identify CD73 binding partners in TNBC cells. In theory, the spatial distribution of cellular organelles within exocytic pathways is distinctive, and they communicate via a complex vesiculotubular transport system. This process is orchestrated by RAB and v-SNARE proteins, which manage consecutive transport stages, including vesicle formation, movement, and docking at target sites. Indeed, we detected several vesicle-associated proteins, including RAB8A, RAB13, and VAMP3, tightly interacting with CD73, with validation by coimmunoprecipitation and proximity ligation assays (Supplementary Fig. 8a−c). We deciphered the molecular detail on the functional interaction between CD73 and VAMP3 (a member of the vesicle-associated membrane protein family that facilitates vesicle exocytosis, docking, and fusion) and found that binding of CD73 to VAMP3 enables CD73 membrane fusion and translocation, whereas

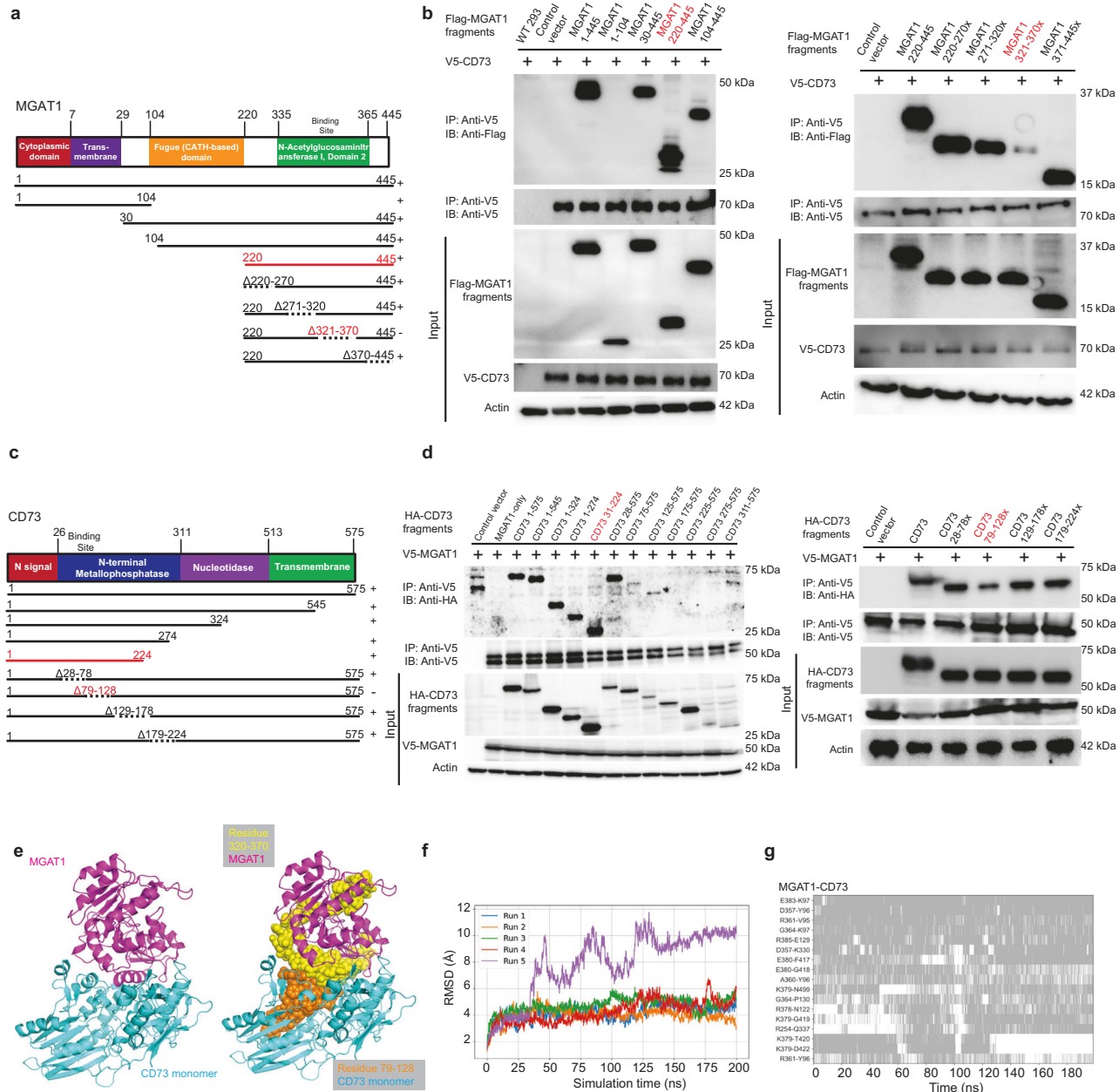

**Fig. 4 | Mapping of molecular regions facilitating the interaction between MGAT1 and CD73, and 3D structural modeling of MGAT1-mediated CD73 glycosylation. a** Schematic diagram of human MGAT1 domains and strategy to engineer a series of MGAT1 truncation and deletion mutants. **b** The interactions between CD73 and MGAT1 fragments were examined by co-IP experiments in HEK-293T cells. Red text indicates the smallest truncation mutant that bound CD73 (left) and the deletion mutant with reduced binding to CD73 (right). Amino acids 321–370 were identified as the region on MGAT1 that mediates the interaction with CD73. The samples derived from the same experiment but different gels for Flag V5, and β-ACTIN were processed in parallel. **c** Schematic diagram of human CD73 domains and strategy to engineer a series of CD73 truncation and deletion mutants. **d** The interactions between MGAT1 and CD73 fragments were examined by co-IP experiments in HEK-293T cells. Amino acids 79–128 on CD73 were identified as the region that mediates the interaction between MGAT1 and CD73. The samples

derived from the same experiment but different gels for V5 and HA, and β-ACTIN were processed in parallel. **e** The top CD73-MGAT1 complex model predicted by ClusPro with an estimated docking occupancy of 10.5% and binding affinity of −15.4 kcal/mol. The interacting domains discovered by mapping were highlighted. **f, g** The time evolution of the RMSDs of the MGAT1-CD73 complex in five 200-ns MD simulations of the complex formed with MGAT1-CD73 using the start points in (**e**) is shown in (**f**). The RMSD was evaluated after structurally aligning the conformers observed during MD trajectories with respect to the initial ClusPro-predicted complex. The corresponding time evolution of residue-residue interactions between MGAT1 and CD73 residues is shown in (**g**), and regions shaded in gray refer to time intervals during which the indicated residue pairs made interfacial contacts. Data and western blots are representative of at least three independent experiments. Source data are provided as a Source Data file.

deficiency of glycosylation-mediated CD73 dimerization resulted in failure to translocate to the membrane (Fig. 5j and Supplementary Fig. 8d–g). Taken together, the above observations suggest that MGAT1 catalyzes CD73 glycosylation that, in turn, triggers CD73 dimerization, ensuring its translocation from the cytosol to the

membrane via the VAMP3-RAB cascade (Fig. 5k). To investigate the upstream signaling that potentially modulates MGAT1 in the context of tumor invasion, we conducted Spearman's rank correlation analysis using proteomic datasets from the PDC000408 cohort[33] and found a tight correlation between elevated expression of MGAT1 and

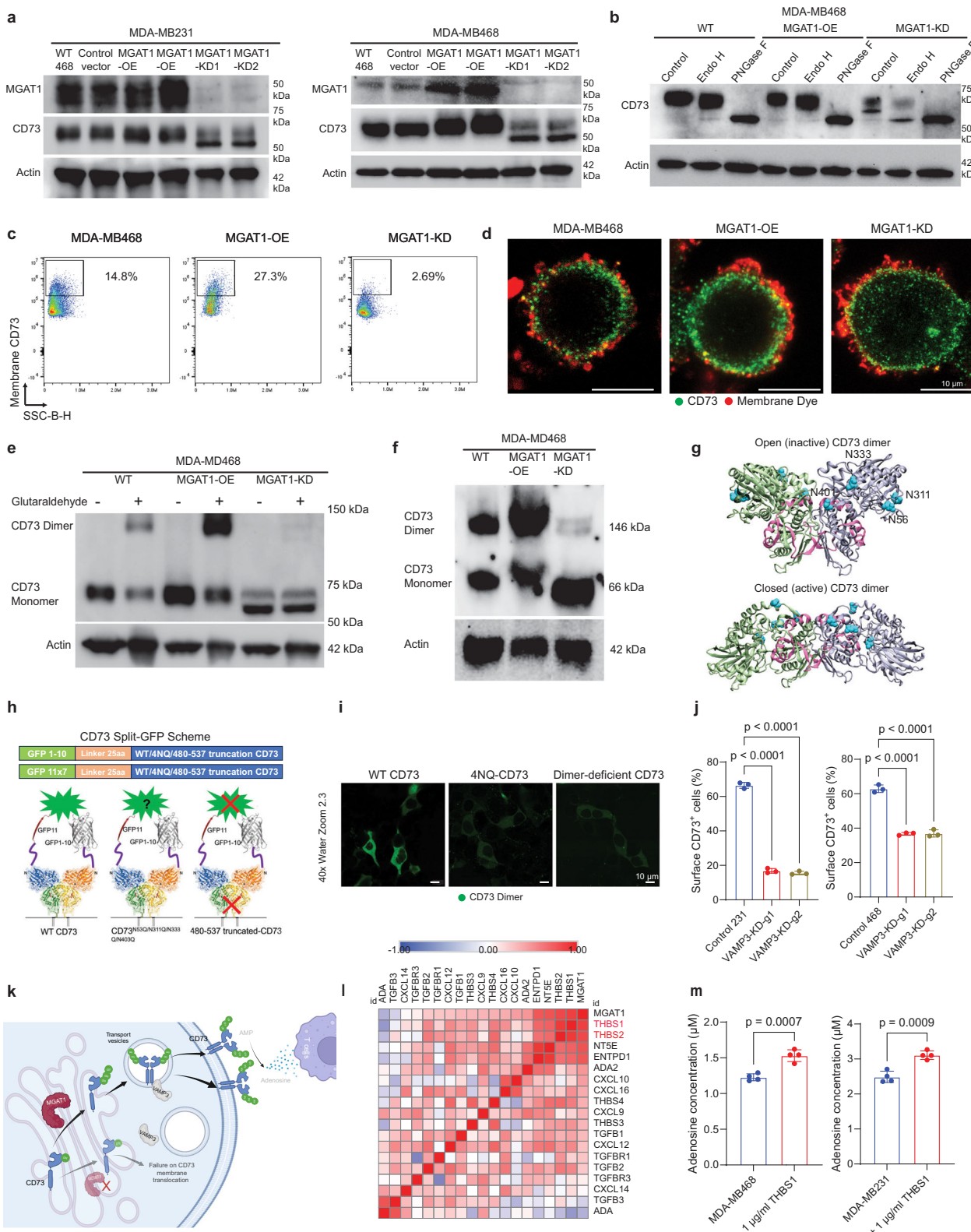

up-regulation of THBS1 and THBS2 signaling pathways (Fig. 5l). To confirm this observation, we validated the effect of THBS1 and THBS2 on the MGAT-CD73-adenosine cascade and observed that stimulation of TNBC cells with THBS1 leads to increased MGAT1 abundance and up-regulated membrane-bound CD73 levels as well as enhanced adenosine production (Fig. 5m and Supplementary Fig. 8h–j), suggesting the overactivation of MGAT1 in tumor immune invasion could be due to abnormal THBS1 signaling.

## Development of a pharmacological inhibitor of MGAT1 that restores tumor immune responses in immune-cold breast cancer cells

We next aimed to determine whether blocking the aberrant accumulation of CD73 on the tumor cell surface could revive tumor immune response in immune-cold breast cancers. We initially developed an in vitro high-throughput screening assay to monitor the effect on MGAT1 enzymatic activity of a library of putative inhibitors (Fig. 6a).

**Fig. 5 | MGAT1-mediated glycosylation of CD73 orchestrates its dimerization and translocation to the membrane. a** The CD73 protein levels and molecular sizes in MDA-MB468 and MDA-MB231 cells with MGAT1 OE/KD were determined by immunoblotting. The samples derived from the same experiment but different gels for CD73 and MGAT1, and β-ACTIN were processed in parallel. **b** Immunoblotting of CD73 in MGAT1 OE/KD MDA-MB468 lysates treated with Endo H/PNGase F. **c** Flow cytometry analysis of membrane CD73 in MGAT1 OE/KD MDA-MB468 cells. **d** The membrane distribution of CD73 (green) was visualized with stimulated emission depletion microscopy imaging in MDA-MB468 cells with MGAT1 OE/KD with membrane dye (red). **e, f** CD73 dimerization in MGAT1 OE/KD MDA-MB468 cells was assessed by glutaraldehyde cross-linking (**e**) and semi-native gel (**f**) immunoblotting. **g** Structural analysis of CD73 dimerization in inactive (PDB: 4H2G) and active (PDB: 6TVX) states, highlighting N-glycosylation sites (cyan) and C-terminal dimerization interface (magenta). **h** The scheme of investigating CD73 dimerization with a split-GFP assay. The GFP 1-10 or GFP 11 × 7 was attached to the N-terminus of WT CD73, CD73-4NQ, or CD73Δ480-537 and co-transfected into cells to investigate

their dimerization abilities. **i** The split-GFP signal generated from dimerized WT CD73, CD73-4NQ, or CD73Δ480-537 was visualized by confocal microscopy in HEK-293T cells. **j** Flow cytometry of membrane CD73 + cells in MGAT1 OE/KD MDA-MB231 and MDA-MB468 with or without VAMP3 KD. **k** Schematic model showing how MGAT1-mediated glycosylation of CD73 orchestrates its dimerization and further translocation to cell membranes. Failure of CD73 glycosylation impedes CD73 dimerization and membrane translocation. Created in BioRender. Zhu, Y. (2025) https://BioRender.com/i34w446. **l** Spearman's rank correlation analysis shows the MGAT1 protein expression is highly positively correlated with the THBS1 signaling pathways. **m** Adenosine levels in MGAT1 OE/KD MDA-MB468 and MDA-MB231 cells treated with THBS1. Data (mean ± SEM), images, western blot, and flow cytometry are representative of at least three independent experiments. Statistical significance was determined using one-way ANOVA with Tukey's multiple comparisons test (**j**) or two-tailed unpaired *t* test (**m**). Source data are provided as a Source Data file.

---

An anti-cancer compound library (HY-L025 from MedChemExpress) was subjected to screening of inhibitors that could block the assembling glycan chain. We obtained several hits, and a compound named TW-37, an inhibitor of recombinant Bcl-2, Bcl-xL, and Mcl-1[41], was the most potent candidate for inhibiting MGAT1 activity (Fig. 6b). We then used a computational model to search for more potent and specific MGAT1 inhibitors based on the structure of TW-37.

We employed two distinct strategies for pharmacophore modeling to optimize the likelihood of discovering highly potent anti-MGAT1 inhibitors. The first pharmacophore model (PM_1) was constructed based on the docking-predicted binding of TW-37 to MGAT1 (refer to Fig. 6c and Supplementary Fig. 9a, b). Through FTMap analysis[42], we identified high-affinity probe-binding residues of MGAT1, notably R115, Y182, I185, E209, L267, W288, S320 and R413. Then, combining high-affinity binding residues with their functional significance, we incorporated essential pharmacological features (depicted in Fig. 6d): (1) Hydrogen donor near D289 (active site of MGAT1); (2) Hydrogen donor near E209, D210, and D211 (Mn$^{2+}$ binding site); (3) Hydrogen acceptor near R115 (substrate binding site); and (4) Aromatic near F324 (comparably less conserved among MGAT1 homologs). Then, we used PM_1 to screen the MolPort library using Pharmit server[43]. The predicted binding poses of the top 10 compounds screened by Pharmit are illustrated in Fig. 6e. While PM_1 was devised through static docking simulations, we recognized the necessity of accounting for the dynamic nature of MGAT1. To achieve this, we employed MD simulations of the target protein in the presence of explicit water and probe molecules representative of drug-like fragments (detailed in the Methods section). Specifically, we focused on the snapshot identified by Pharmmaker[44], which exhibited the highest affinity for the probe molecules (as depicted in Supplementary Fig. 9c). Subsequently, leveraging the insights gleaned from these druggability simulations, we constructed the second pharmacophore model (PM_2), utilizing the residues demonstrating high affinity for the probe molecules (as illustrated in Supplementary Fig. 9d). PM_2 was then screened against the MolPort library using Pharmit server. Collectively, virtual screening utilizing both PM_1 and PM_2 yielded a selection of 14 compounds for experimental validation (see Supplementary Data 2).

The 14 lead compounds from the virtual screening were initially validated in an MGAT1 enzymatic activity assay. Several compounds, including No.2, No.8, No.9, and No.14, showed potent inhibition of MGAT1 catalytic function (Fig. 6f). We next examined the effect of these candidate compounds on CD73 membrane translocation in cancer cells. The candidate compounds were added to the growth medium of MDA-MB231 and MDA-MB468 cells, and the membrane-bound CD73 abundance was measured with flow cytometry and immunostaining (Fig. 6g, h and Supplementary Fig. 9e). The adenosine production response to identified compounds was measured as well (Supplementary Fig. 9f). The results in Fig. 6f–i demonstrated that

No.2, No.8, and No.9, whose structures are shown in Fig. 6j, are ideal candidates based on the dual criteria of MGAT1 activity inhibition and suppression of membrane translocation of CD73. They all showed dose-dependent inhibition of MGAT1 catalytic function (Fig. 6k). These candidates were further evaluated using cancer and immune cell coculture to assess the effect on the antitumor immune response. The CD8$^+$ T cell function was detected using flow cytometry (Fig. 6l–o), and cancer cell survival was examined with image-based viability (Fig. 6p). Based on evaluation standards from multiple layers, W-GTF01 (No.8 compound) was confirmed as a potent inhibitor of MGAT1 activity and stimulator of CD8$^+$ IFNγ-producing T-cell response, leading to lowest cancer cell survival. To investigate whether W-GTF01 directly affects the MGAT1-CD73 axis, we established CD73-KD stable lines in MDA-MB468 cells and repeated the coculture experiment with immune cells under W-GTF01 treatment. The data demonstrated that the addition of W-GTF01 did not present further strengthened efficacy to immune cells in the absence of CD73 (Fig. 6q). Furthermore, to determine if No.2, No.8 (W-GTF01), and No.9 exhibit off-target effects against Bcl-2, we measured Bcl-2 abundance following treatment in MDA-MB231 and MDA-MB468 cells. The results confirmed that these compounds do not affect Bcl-2 expression (Supplementary Fig. 9g), further supporting their specificity toward MGAT1.

To investigate the mechanism by which W-GTF01 leads to superior efficacy in inhibiting MGAT1, we docked W-GTF01 onto human MGAT1 (Fig. 6r). The computed binding affinity for W-GTF01 was −10.4 ± 0.3 kcal/mol. A favorable π-stacking interaction (marked by a blue transparent oval) is formed between F324 (yellow stick) and W-GTF01 (magenta stick). MGAT1 binding site residues within 4 Å of W-GTF01 included R115, I185, E209, P265, G266, L267, D289, R293, G318, V319, F324, D406, and R413. Besides the favorable π stacking interaction, four putative hydrogen bonds were formed between W-GTF01 and R115, E209, R293, and the backbone of V319.

## Modulation of MGAT1-mediated CD73 membrane translocation affects tumor growth and capacity to elicit antitumor CD8$^+$ T cell responses

To evaluate the role of MGAT1 in tumor growth in vivo, we created murine cell lines with modified MGAT1 expression. We developed MGAT1-OE, MGAT1-KD 4T1, and E0771 cells, with an empty vector as a control (Supplementary Fig. 10a). The impact on tumor growth was then assessed using a syngeneic mouse model (Fig. 7a). Engineered 4T1 control and 4T1 MGAT1 OE breast cancer cells were subcutaneously injected into the mammary fat pad of female BALB/C mice. The volume of mammary tumors was measured with calipers and calculated using the formula: $V = (W^2 \times L)/2$, where V is the tumor volume, W is the tumor width, and L is the tumor length. A significant increase in both the tumor size and tumor weight was observed 25 days post-injection of the MGAT1-OE tumor cells (Fig. 7b, c). Furthermore, consistent with

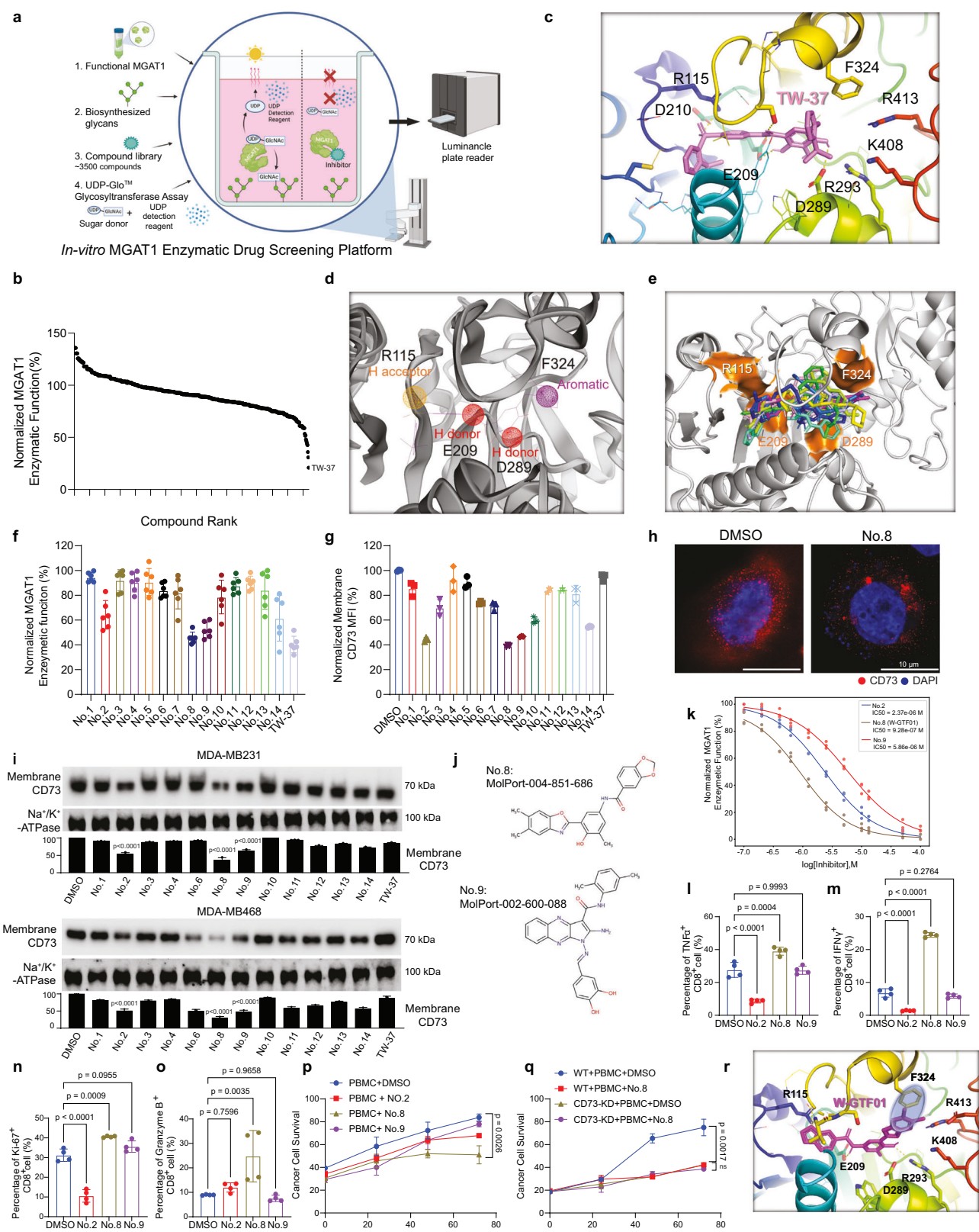

the aforementioned in vitro results (Fig. 5c, d and Supplementary Fig. 6c, d), MGAT1 OE resulted in an increase in membrane-bound CD73 on tumor cells from tumor-bearing mice (Fig. 7d). Similar results were observed in E0771 and E0771 MGAT1 OE models (Supplementary Fig. 10b, c). These data support a tumor-promoting role of the MGAT1-CD73 axis in vivo.

To evaluate how MGAT1 influences tumor immune composition, we utilized high-dimensional spectral flow cytometry (CyTEK) with a custom panel designed to capture key markers across major immune cell types, based on previously established protocols[45]. We applied viSNE, a dimensionality-reduction algorithm, to compare immune landscapes between 4T1 control and MGAT1-overexpressing tumors

**Fig. 6 | Development of a high-throughput screening system that can quantify MGAT1 enzyme activity with luminance signal to develop specific MGAT1 inhibitors. a** Schematic of an MGAT1 function-based drug screening assay converting enzymatic activity to a luminance signal. Key components include MGAT1, mannose-5-AEAD (substrate), UDP-GlcNAc (sugar donor), and a compound library. MGAT1 inhibition reduces free UDP, decreasing luminance. Created in BioRender. Zhu, Y. (2025) https://BioRender.com/t24b827. **b** Normalized MGAT1 enzymatic function in high-throughput screening with an anti-cancer compound library (~ 3500 compounds). **c** AutoDock Vina-predicted most energy-favorable binding of TW-37 (pink) to MGAT1, with interacting residues (within 3.5 Å) shown. **d** Pharmacophore model 1 (PM1) based on binding of TW-37 to MGAT1. **e** Pharmit-predicted binding of the top 10 compounds (colored differently) screened against PM_1. **f** Normalized MGAT1 enzyme function upon treatment with 15 screened compounds was measured by function-based screening assay. **g** Flow cytometry analysis of membrane CD73 in MDA-MB468 cells treated with 15 compounds.

**h, i** The membrane-fractionated protein levels of CD73 in MDA-MD468 cells upon treatment with screened compounds were determined by (**h**) immunostaining and (**i**) immunoblotting. j The chemical structures of two top-ranked compounds (No.8 and No.9). **k** Dose-dependent MGAT1 inhibition by No.2, No.8, and No.9 in the function-based assay. **l–o** MDA-MB468 cells were cocultured with PBMC cells and the indicated compounds. Percentages of TNFα⁺ (l), IFNγ⁺ (m), Ki-67⁺ (**n**), and Granzyme B⁺ (**o**) in CD8+ T cells were measured using flow cytometry. **p, q** Time-lapse quantification of MDA-MB468 survival in PBMC coculture treated with No.2/No.8(W-GTF01)/No.9 (**p**) or No.8 in MDA-MB468-WT/CD73-KD cells (**q**). **r** Binding of W-GTF01 (MolPort-004-851-686) to MGAT1, highlighting π-stacking interaction between F324 (yellow) and W-GTF01 (magenta). Data (mean ± SEM), images, western blot, and flow cytometry are representative of at least three independent experiments. Statistical significance was determined using one-way ANOVA with Tukey's multiple comparisons test (**i**, **l–o**) or two-way ANOVAs followed by Tukey's multiple comparison tests (**p**, **q**). Source data are provided as a Source Data file.

(Fig. 7e). CD45⁺ live single cells from tumor infiltrates clustered into well-defined populations, including CD4⁺ T cells (CD3⁺CD4⁺), CD8⁺ T cells (CD3⁺CD8⁺), proliferating CD8⁺ T cells (CD3⁺CD8⁺Ki-67⁺), CD8⁺ tissue-resident memory T cells (CD69⁺CD103⁺, TRM), exhausted CD8⁺ T cells (PD-1⁺TIM3⁺, $T_{EX}$), stem-like progenitors of exhausted CD8⁺ T cells (CD69⁺Ly108⁺, $T_{PEX}$), Treg (CD3⁺CD4⁺ Foxp3⁺CD25⁺), non-Treg CD4⁺ T cells (CD3⁺CD4⁺ Foxp3⁻), B cells (CD19⁺CD3⁻), dendritic cells (MHC-II⁺CD11c⁺CD11b⁺, DCs), tumor-associated macrophages (Gr1⁻F4/80⁺CD11b⁺, TAMs), polymorphonuclear myeloid-derived suppressor cells (Ly6G⁺ Ly6C$^{lo}$CD11b⁺, PMN-MDSC), monocytic MDSCs (Ly6G$^{lo}$ Ly6C⁺CD11b⁺, M-MDSCs), NK (CD3⁻ NKp46⁺), and NKT (CD3⁺NKp46⁺) cells. Notably, MGAT1-overexpressing tumors showed elevated proportions of exhausted CD8 + T cells, Tregs, monocytic MDSCs, and TAMs among CD45⁺ cells (Fig. 7e). Furthermore, the fractions of PD-1⁺TOX⁺ and PD-1⁺CD101⁺ CD8⁺ cells were significantly increased in MGAT1 OE tumors (Fig. 7f). Consistent with the Fig. 7f, our additional analysis in Supplementary Fig. 10d further supports the observation that MGAT1-OE tumors exhibit an increased percentage of exhausted-like CX3CR1⁻CD101⁺ CD8⁺ T cell populations compared to control tumors, reinforcing the impact of MGAT1 overexpression on T cell exhaustion and immunosuppressive mechanisms in the tumor microenvironment. We also found that MGAT1 OE in tumor cells enhanced a tumor-promoting M2-like phenotype in TAMs, usually defined by the expression of CD163 and MHC II (Fig. 7g and Supplementary Fig. 10e). Furthermore, there was a significant reduction in IFNγ and TNFα secretion by infiltrating CD8⁺ T cells in MGAT1 OE tumors (Fig. 7h).

Conversely, MGAT1 KD in either 4T1 or E0771 murine breast cancer cell lines hindered tumor development compared to the control cell line (with empty vectors) as determined by both tumor size and tumor weight measurements (Fig. 7i, j and Supplementary Fig. 10f, g). In addition, there was decreased fraction of Tim3⁺PD1⁺ and increased fraction of CD8⁺ T cells expressing a higher level of Ki67 (Fig. 7k, l) in MGAT1 knockdown tumor-bearing mice compared to that in the control mice. To evaluate the importance of the physical interaction between MGAT1 and CD73 for tumor growth, we overexpressed both MGAT1 and CD73 WT (MGAT1 OE + CD73$^{WT}$) or MGAT1 and CD73-4NQ (MGAT1 OE + CD73$^{4NQ}$) in 4T1 and E0771 murine TNBC cell lines. E0771, E0771-MGAT1-CD73$^{WT}$, and E0771-MGAT1-CD73$^{4NQ}$ cells were subcutaneously injected into the mammary fat pad of female C57BL/6 mice. While there was no significant difference in tumor growth between mice with E0771-MGAT1-CD73 $^{4NQ}$ and E0771 vehicle control, tumor growth was accelerated in E0771-MGAT1-CD73$^{WT}$ tumor-bearing mice (Fig. 7m, n). MGAT1-CD73 co-overexpression resulted in a significant increase in membrane-bound CD73 on tumor cells followed by immune suppression from tumor-bearing mice, but it is not observed in E0771-MGAT1- CD73$^{4NQ}$ (Fig. 7o and Supplementary Fig. 10h–j). Similar results were observed in 4T1

models (Supplementary Fig. 10k, l). These results underscore the importance of MGAT1-mediated glycosylation of CD73 for tumor growth.

## Pharmacological blockade of the MGAT1-CD73 axis promotes tumor immunogenicity and inhibits tumor progression in immune-cold breast cancers

To determine the therapeutic relevance of our inhibitor W-GTF01, we examined the antitumor effect of W-GTF01 by single treatment or in combination with the anti-hPD-L1 drug durvalumab, using a TNBC 4T1 model where the endogenous mouse PD-L1 was knocked out and replaced with the human counterpart. W-GTF01 was injected twice per week at the dose of 5 mg/kg, and PD-L1 antibody durvalumab was injected three times at the dose of 5 mg/kg. PBS and IgG control were used in control groups. Combining W-GTF01 with durvalumab led to enhanced suppression in tumor growth without evident toxic effects (Fig. 7p), as seen in consistent mouse body weights (Supplementary Fig. 10m). In addition, we also observed that the combination of W-GTF01 and anti-PD-L1 significantly improves survival outcomes compared to either treatment alone (Fig. 7q), reinforcing the conclusion that targeting MGAT1 enhances the efficacy of immune checkpoint blockade. The IHC images showed that the combination of W-GTF01 and PD-L1 antibody markedly increased CD8⁺ T cell infiltration and activation (Supplementary Fig. 10n). In addition, we observed a moderate decrease in T cell exhaustion, as evidenced by reduced Tox staining, when compared to treatment with anti-PD-L1 antibody alone. These outcomes suggest the therapeutic potential of targeting MGAT1 pharmacologically to enhance immunogenic cell death in response to current immunotherapy.

To further validate the role of the MGAT1-CD73 axis in immune checkpoint blockade responses, we conducted gain- and loss-of-function experiments in MGAT1-KD, MGAT1-OE, and CD73-4NQ tumor models (Supplementary Fig. 11a–f). MGAT1 overexpression impaired the anti-PD-L1 ICB efficacy (Supplementary Fig. 11a, b). Conversely, the MGAT1 knockdown significantly enhanced the efficacy of anti-PD-L1 therapy, characterized by reduced tumor progression (Supplementary Fig. 11c, d). Similarly, the CD73-4NQ tumors exhibited significant tumor growth inhibition under anti-PD-L1 therapy compared to WT-CD73 tumors (Supplementary Fig. 11e, f). These findings highlight the critical role of MGAT1-mediated CD73 glycosylation in driving immune suppression and resistance to ICB therapy.

## Spatially resolved signatures pertaining to MGAT1$_{high}$/CD73$_{high}$ are associated with unfavorable immune responses in clinical samples

To assess the spatial interaction of MGAT1 and CD73 in tumor cells with immune signaling programs within the tumor microenvironment,

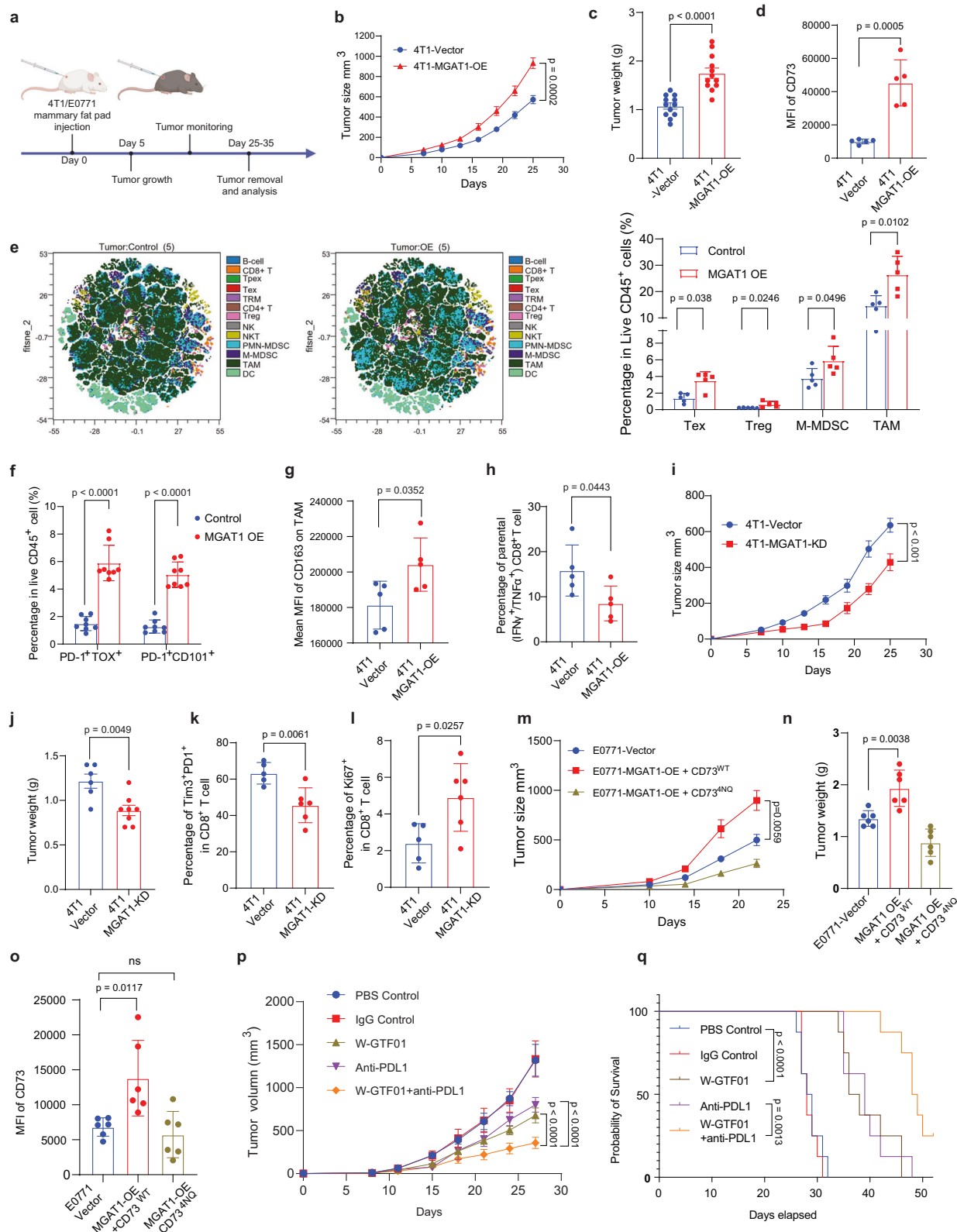

we conducted spatially indexed transcript profiling (GeoMx digital spatial profiling) of tumor tissue cores (3 per patient) from 50 TNBC patients as described previously[24]. We profiled 32 regions of interest (ROIs) per tissue section and further segmented each region into epithelial versus non-epithelial areas based on PanCK staining (Fig. 8a and Supplementary Fig. 12a). Co-staining of MGAT1 and CD73 was used subsequently for spatial segmentation analysis. We then scanned these

regions to construct digital maps of cellular content and ROIs containing heterocellular populations representative of the whole section or regions exclusively containing immune cells. Individual ROIs were subject to independent transcriptional analysis by next-generation sequencing. The major cell types were classified using the CIBERSORT digital cytometry platform[42,46]. We found that ROIs contained a diverse, spatially variable population of constituent cells, in line with

**Fig. 7 | MGAT1-mediated glycosylation of CD73 orchestrates tumor immune evasion in vivo. a** Schematic of mouse experiments: 4T1 or E0771 breast cancer cells were injected orthotopically into WT BALB/c or C57BL/6 mice. Tumor weight was measured at the endpoint, and tumor growth was monitored. Created in BioRender. Zhu, Y. (2025) https://BioRender.com/y73s493. **b–d** Tumor growth ($n = 6$) (**b**), endpoint tumor weight ($n = 6$) (**c**), and membrane CD73 levels (**d**) in 4T1-MGAT1 OE ($n = 5$) vs. control tumors ($n = 5$). **e** tSNE visualization of immune cell composition, comparing TEX, Tregs, MDSCs, and TAMs among live CD45 + cells in 4T1-MGAT1 OE ($n = 5$) vs. control tumors ($n = 5$) at day 25. **f** Percentages of PD-1[+]TOX[+] and PD-1[+]CD101[+] CD8 + T cells among CD45[+] cells between 4T1 control tumors ($n = 8$) and 4T1 MGAT1 OE tumors ($n = 8$). **g** CD163 expression levels were compared in tumor-associated macrophages (TAMs) between 4T1-MGAT1 OE ($n = 5$) vs. control tumors ($n = 5$). **h** Comparison of the percentage of IFNγ[+]/TNFα[+] among tumor-infiltrated CD4[+] or CD8[+] T cells between 4T1 control ($n = 5$) and MGAT1 OE tumors ($n = 5$). **i, j** Tumor growth (**i**) and endpoint tumor weight (**j**) in 4T1-MGAT1 KD ($n = 8$) vs. control tumors ($n = 6$). **k** Percentage of Tim3[+]/PD1[+] CD8[+] T cells in 4T1-MGAT1 KD ($n = 6$) vs. control tumors ($n = 5$). **l** Ki-67[+] tumor-infiltrating CD8[+] T cells in 4T1-MGAT1 KD ($n = 6$) vs. control tumors($n = 5$). **m–o** Tumor growth (**m**), endpoint tumor weight (**n**), and membrane CD73 levels (**o**) in E0771-MGAT1 OE + CD73WT ($n = 6$) vs. CD73-4NQ tumors ($n = 6$) in C57BL/6 mice. **p, q** 4T1-hPD-L1 tumors (the endogenous mouse PD-L1 was replaced with its human counterpart) were treated with W-GTF01 (10 mg/kg, i.p.) twice/week and durvalumab (10 mg/kg, i.p.) three times/week ($n = 8$). PBS and IgG were used in the control groups. Tumor growth (**p**) and survival (**q**) were monitored. Data (means ± SEM), images, and flow cytometry are representative of at least three independent experiments with 5–10 independently analyzed mice per group. Statistical significance was determined using one-way ANOVA with Tukey's multiple comparisons test (**n, o**) or two-tailed unpaired $t$ test (**c–h, j–l**) or two-way ANOVAs followed by Tukey's multiple comparison tests (**b, i, m, p**). Source data are provided as a Source Data file.

our prior work[24]. There were relatively higher average fractions of immune system cell types in the non-epithelial stromal areas (Fig. 8b, c) as compared to PanCK[+] tumor areas (Supplementary Fig. 12b, c). Although the non-tumor regions displayed similar proportions of endothelial cells, fibroblasts, and individual immune subsets between MGAT1lo and MGAT1hi areas (Supplementary Fig. 12b) or between MGAT1loCD73lo (DL) and MGAT1hiCD73hi areas (DH) (Supplementary Fig. 12c), memory B cell and CD8[+] memory T cells were enriched at higher levels in MGAT1lo areas relative to MGAT1hi areas (Fig. 8b) or DL relative to DH areas within the non-epithelial stromal compartment (Fig. 8c).

We further validated a positive correlation between MGAT1 and membrane CD73 expression in malignant epithelial areas across all regions per tumor using multiplexed immunohistochemistry (mIHC) (Fig. 8d–f). Notably, the subsequent spatial analysis revealed a greater abundance of CD8[+] T cells touching DL tumor cells compared with that touching DH tumor cells (Fig. 8g). Using gene pathway analysis (KEGG/Reactome/Gene ontology) (Fig. 8h, i) based on spatial transcriptomes, we found the Glycoprotein pathway pertaining to CD73 glycosylation modulation was particularly enriched in DH compared to DL tumor areas, further supporting a specific role of MGAT1 in regulating CD73 glycosylation and protein expression in tumor cells.

Moreover, we found Innate immune response, B cell receptor signaling and B cell differentiation, T cell activation, and TCR signaling pathways related closely to the tumor immune response were significantly enriched in non-tumor compartments from DL areas compared to that from DH areas (Fig. 8j, k). In addition, Gene Set Enrichment Analysis (GSEA) revealed the enriched pathways of TGF-β signaling (Supplementary Fig. 12d) in non-tumor compartments and IL-10 signaling (Supplementary Fig. 12e) in tumor compartments from DH areas compared to that from DL areas. Similar results are also obtained from a comparison between MGAT1lo and MGAT1hi areas (Supplementary Fig. 13a–g) in non-tumor and tumor compartments across all ROIs. The gating strategies for the Cytek spectral flow cytometry data analysis in tumor-infiltrated lymphocytes (Fig. 7e) is shown in Supplementary Fig. 14a. These findings indicate that distinct immune features of the DH areas are implicated in the suppression of anti-tumor responses and the promotion of tumor growth, further supporting the importance of MGAT1 for CD73 glycosylation in tumor immune evasion (Fig. 8l).

## Discussion

This study integrates multiomics analyses to comprehensively define a distinct immunosuppressive signature in a subset of TNBCs, followed by in-depth mechanistic dissection with multidisciplinary approaches. We identified MGAT1, a critical glycosyltransferase, as an inhibitory tumor immune regulator whose overexpression drives tumor immune evasion in TNBC. Mechanistically, we have demonstrated the essential role of MGAT1 in governing CD73 translocation from the Golgi body to

the cell membrane. The addition of GlcNAc to CD73 in the Golgi compartment by MGAT1 triggers the dimerization of CD73 that, in turn, allows its recognition by VAMP3 and enables CD73 fusion to the membrane. Dysregulated MGAT1 caused uncontrolled accumulation of CD73 on the tumor plasma membrane that elevates levels of adenosine, leading to suppression of CD8[+] T cell function and subsequent tumor immune evasion. In the context of tumor evasion, aberrant upstream THBS1 and THBS2 provoke overactivation of MGAT1, leading to enhanced CD73 glycosylation and membrane-bound accumulation, especially in immune-cold breast tumors. To explore the therapeutic potential, we have developed a pharmacologic inhibitor, W-GTF01, that potently inhibits MGAT1, down-regulating CD73 glycosylation and thereby scaling down CD73-facilitated adenosine production and reviving CD8[+] T cell function. Notably, treatment with W-GTF01 sensitized TNBC tumors to anti-PD-L1 therapy, even in the case of tumors with high expression of MGAT1 and CD73 (Fig. 7q). This study introduces a paradigm by specifically targeting glycosylation of immune checkpoint proteins for improving tumor immunogenicity and overcoming resistance to immune checkpoint inhibitors in immune-cold TNBCs.

### MGAT1-mediated glycosylation of CD73 orchestrates Golgi-associated CD73 membrane translocation that determines the adenosine-dependent suppression of CD8[+] T cell function

Dysregulation of T cell function leads to tumor escape from immunosurveillance that enhances tumor initiation and progression[47,48]. Suppressed T cell proliferation and enhanced T cell exhaustion are major barriers to immunotherapy in treating immune-cold tumors. Recent genetic and biochemical studies have pinpointed the critical role of CD73 in tumor malignancy and immunogenicity, emphasizing its great potential as a therapeutic target[22,24,47–49]. Extensive research has explored the transcriptional and etiological regulation of CD73 at the gene level[21,50]. During malignant tumor progression, CD73 expression is dynamically modulated, particularly in response to hypoxic signaling[49,51]. Beyond HIF-1α-mediated regulation, CD73 levels in tumor cells are also influenced by Wnt and cyclic AMP (cAMP) signaling, as well as cytokine-driven pathways, highlighting its integration into multiple oncogenic signaling networks[52–54]. While significant research has focused on the transcriptional regulation of CD73, limited attention has been given to CD73 protein regulation, especially how CD73 is regulated by post-translational modifications. A recent publication demonstrated that CD73 expression level is regulated through ubiquitylation, which controls its protein stability and turnover, thereby influencing its overall abundance on the tumor cell surface[25]. In contrast, our study reveals a distinct mechanism that CD73 is tightly regulated by glycosylation regarding its cellular distribution, including its protein synthesis/quality control, trafficking, and membrane translocation. Our fluorescent imaging studies revealed that once CD73 passed the quality control checkpoint in the endoplasmic

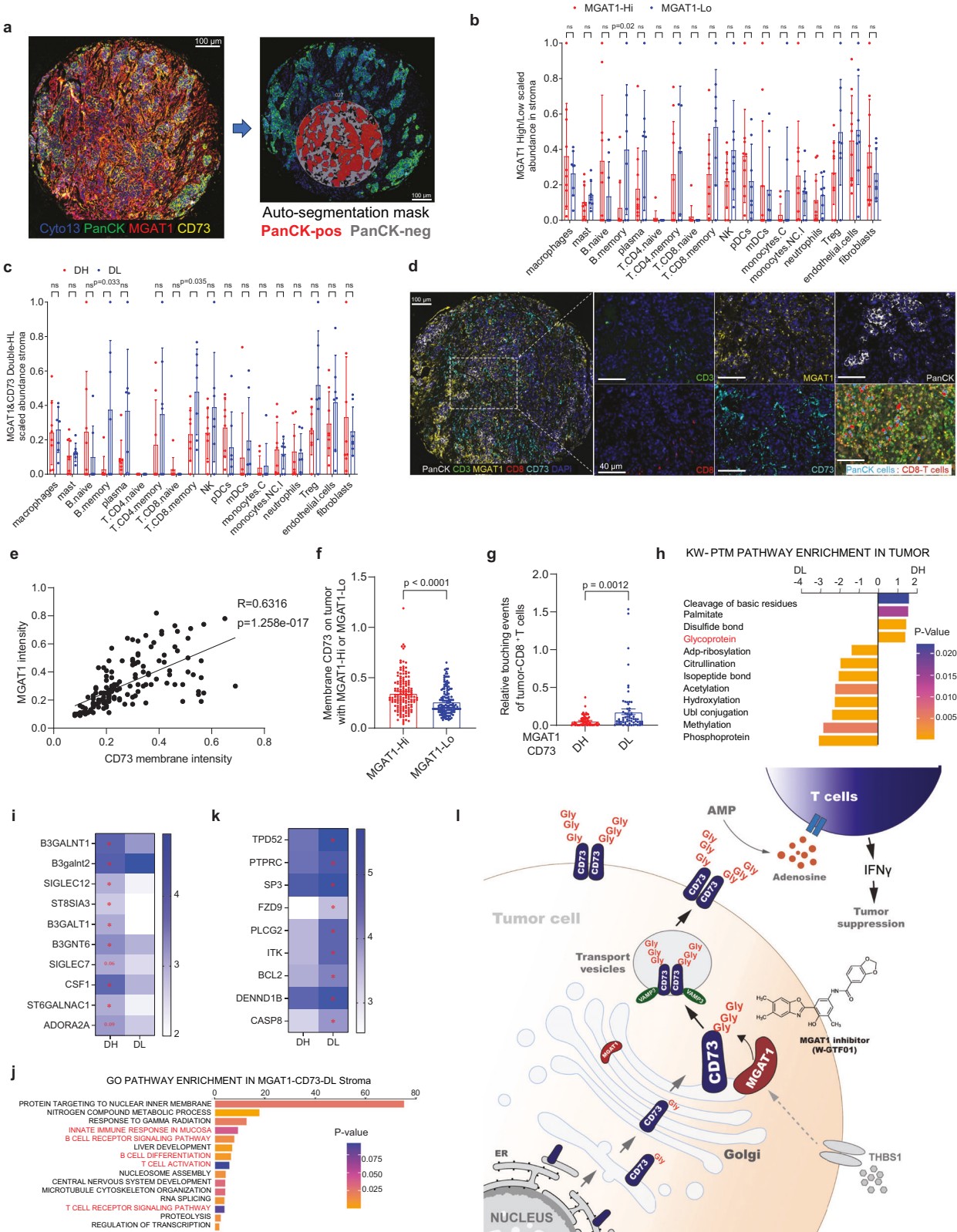

reticulum, MGAT1 interacted with CD73 in the Golgi compartment, an essential cellular apparatus allowing protein trafficking to a functional destination. Using molecular docking and simulation coupled with biochemical and cell biological validation, we deciphered the in-depth dynamics of how MGAT1 recognizes CD73, how MGAT1 interacts with the substrate CD73, and how GlcNAc is added to CD73. The functional interference analyses based on the four asparagine residues

(glycosylation sites) on CD73 identified by mass spectrometry further revealed the biochemical consequences of MGAT1-mediated CD73 glycosylation in terms of its membrane translocation, where the replacement of all asparagine residues by glutamine disrupted translocation to the plasma membrane, as demonstrated by biochemical fractionation, fluorescent image, and flow cytometry. This work shows that, before its integration into the membrane, MGAT1-dependent

**Fig. 8 | Spatially resolved signatures pertaining to MGAT1$_{high}$/CD73$_{high}$ are associated with unfavorable immune responses. a** Representative staining of MGAT1, CD73, and PanCK, followed by the selection of regions of interest (ROI) (*n* = 32) and subsequent auto-segmentation in TNBC tissue sections using GeoMX DSP. **b, c** CIBERSORTx analysis of the relative abundance of individual cell populations between MGAT1lo and MGAT1hi areas (**b**) or between MGAT1loCD73lo (DL) and MGAT1hiCD73hi areas (DH) (**c**) among nontumor compartments using transcriptomics data. **d** Representative composite image (left panel) with inset (right) of a breast cancer specimen stained by multicolor IHC comprising CD8, Ki67, CD3, CD73, MGAT1, PanCK, and DAPI. Scale bars: 100 µm. **e** The analysis of MGAT1 and membrane CD73 protein expression on TNBC (*n* = 146) using mIHC shows the positive correlation between MGAT1 with CD73. **f** Intensity of membrane CD73 was compared between MGAT1hi and MGAT1lo tumor regions. **g** Touching events between CD3$^+$CD8$^+$ T cells and PanCK$^+$ tumor cells among DH relative to DL tumor regions. **h, i** UP-KW functional annotation (UniProt KeyWord Functional Annotations) analysis of post-translational modification (PTM) on the DEGs (different expressional genes) in PanCK$^+$ epithelial tumor compartment between DL cases and

DH cases (**i**), revealing selected Glycoprotein at increased levels in DH tumors (**h**). *$p$ < 0.05. **j, k** GO Pathway analysis of DEGs in non-tumor areas between DL cases and DH cases, highlighting selected genes involving B cell receptor signaling and T cell activation at increased levels in DL cases (**k**). *$p$ < 0.05. **l** The proposed working model. MGAT1, a glycosyltransferase, catalyzes CD73 for glycosylation at the Golgi body resulting in CD73 dimerization that enables CD73 plasma membrane translocation mediated by transport vesicles. Overactivation of MGAT1 due to aberrant THBS1 signaling leads to increased membrane-bound abundance of CD73 that enhances adenosine production and subsequently suppresses CD8$^+$ T cell function. Pharmacological blockade of MGAT1 by an MGAT1 inhibitor (W-GTF01) reduces tumor growth by reinstating the capacity to elicit CD8$^+$ effector T-cell responses through inhibition of CD73 glycosylation and membrane translocation. Data (means ± SEM), images, and flow cytometry are representative of at least three independent experiments. Statistical significance was determined using a two-tailed unpaired *t* test (**b, c**), two-tailed paired *t* test (**f, g**), or simple linear regression (**e**). Source data are provided as a Source Data file.

glycosylation of CD73 is a critical step in the medial Golgi, ensuring CD73 membrane-bound enzymatic function in producing adenosine for down-regulating T cell activities. Unlike ubiquitylation, which controls CD73 degradation, N-glycosylation fine-tunes its spatial organization and functional output, underscoring the complexity of its post-translational regulation. By revealing a previously unrecognized role for MGAT1 in CD73 dimerization and immune evasion, our study suggests that disrupting CD73 glycosylation through MGAT1 inhibition may serve as a strategy to enhance immunotherapy responses in TNBC patients. In future studies, mass spectrometry-based quantification of CD73 glycosylation in vivo and detailed biochemical study of the CD73 monomer, dimer, and oligomer in the cytosol and at the membrane will advance our understanding of the regulatory mechanism and physiological relevance of post-translational modification of CD73 in tumor immunogenicity.

## Dimerization of CD73 triggered by MGAT1-mediated glycosylation ensures CD73 cytosolic protein trafficking and membrane translocation

Glycosylation is a fundamental post-translational modification that regulates protein folding, stability, localization, protein-protein interactions, and enzymatic activity[55–57]. Highly conserved across species, glycoconjugates—including O-fucose, O-mannose, N-glycans, mucin-type O-glycans, proteoglycans, and glycosphingolipids—play essential roles in cellular function[55,56,58,59]. Over the past three decades, studies have demonstrated that complex N-glycans are key drivers of tumor progression, metastasis, and immune evasion. These branched glycans, covalently linked to the membrane and secreted proteins, regulate cell-cell and cell-extracellular matrix (ECM) interactions, facilitating tumor proliferation, invasion, and immune escape[60–63]. Aberrant N-glycosylation has been shown to impair immune surveillance, allowing tumor cells to evade recognition and destruction by the immune system[25–27,57,64]. Given their critical role in tumor biology, understanding the mechanisms that regulate complex N-glycan biosynthesis has become a major focus in cancer research, particularly for identifying therapeutic targets in cancer immunotherapy. We demonstrated that MGAT1 catalyzes N-glycosylation of CD73, which results in the dimerization of CD73. Proteins rarely show function in their isolated form in cells, and protein dimerization/oligomerization facilitates protein interaction and functional assembly. We showed that dimerization of CD73 triggered by MGAT1-catalyzed N-glycosylation offers temporal and spatial regulation on the Golgi body, enabling its recognition and interaction with VAMP3, a master protein facilitating vesicle exocytosis, docking, and fusion[65,66]. We analyzed the amino acid residues mediating CD73 homodimerization and showed that loss of dimerization precludes its loading onto VAMP3 for membrane integration. Our work further unraveled the importance of

protein glycosylation on the Golgi body, where CD73 was processed by post-translational modification and sorted for subsequent vesicle-mediated trafficking and membrane fusion. Nevertheless, how VAMP3 coordinates with other vesicle-associated proteins, such as RAB8A and RAB13, in transporting CD73 from the trans-Golgi network to the plasma membrane and how VAMP3 accomplishes CD73 membrane translocation requires further study. In addition to protein translocation, whether the glycosylation-mediated dimerization of CD73 is required for its catalytic function at the cell membrane also needs to be investigated.

Our data indicate that the MGAT1-CD73 axis is not significantly enriched in ER + breast cancers, likely due to the distinct immune landscape and immune evasion strategies of these tumors. ER + tumors exhibit poor T-cell infiltration and low antigen presentation, resulting in minimal selective pressure for cancer cells to upregulate immunosuppressive molecules such as CD73. Consequently, the necessity for MGAT1-mediated glycosylation to enhance CD73 function is reduced. In addition, emerging evidence suggests that estrogen receptor signaling suppresses glycosylation-associated pathways, including MGAT1 regulation, further limiting the role of this axis in ER+ tumors. In contrast, TNBC lacks ER signaling and relies on alternative oncogenic pathways that promote glycosylation changes, enhancing CD73-mediated immunosuppression. While CD73 plays a crucial role in shaping the immunosuppressive tumor microenvironment in TNBC, ER + tumors predominantly evade immune attack through metabolic and hormonal crosstalk rather than direct immune suppression via checkpoint molecules like CD73. These findings underscore the specificity of MGAT1-mediated CD73 upregulation in TNBC, where immune evasion is driven by high checkpoint expression and adenosine-mediated suppression, whereas ER + breast cancers employ distinct, non-CD73-dependent immune escape mechanisms. This distinction highlights the need to develop therapeutic strategies tailored to the immune landscape of each breast cancer subtype.

## W-GTF01, a potent pharmacologic inhibitor, blocks MGAT1-mediated immune suppression and restores CD8$^+$ cytotoxic T cell function in immune-cold TNBC

Aberrant glycosylation has been consistently documented as a critical hallmark of cancer[56–59]. Hyperactivation of glycosyltransferases may result in abnormal glycosylation of immune checkpoint proteins, altering protein stability, trafficking, membrane translocation, and function, resulting in impaired tumor immune response and undermining the efficiency of immunotherapy. There are nearly 200 glycosyltransferases and glycosidases in the human genome, but there has not been a successful inhibitor applied to this protein class. Targeting glycosyltransferases in modifying immune checkpoint proteins could open an avenue to sensitize tumors to immune checkpoint blockade.

However, substrate recognition specificity, the enzyme-substrate complex's dynamic nature, and the diversity of potential glycosylation patterns cause difficulty in designing potent inhibitors with sufficient specificity. In addition, the similarity across the catalytic domains of glycosyltransferases demands the need for inhibitors that can either selectively target a single glycosyltransferase or effectively modulate an entire biosynthetic pathway without adverse off-target effects. The above challenges have left the development of glycosyltransferase inhibitors largely behind those targeting other enzymes, requiring a deep understanding of glycosyltransferase structure-function relationships and drug design and delivery approaches. Recent advances in high-throughput screening approaches and structure-based virtual screening shed light on methodology development. In the present work, we proposed a systematic high-throughput screening system integrating both an in vitro function-based screening step and a virtual screening step based on protein structure to develop an MGAT1 inhibitor. Compared to previous approaches based on protein-compound interaction, the function-based screening assay directly converts the inhibition of MGAT1 enzymatic activity to a quantifiable luminance signal, and the lead compounds from these experiments build the necessary foundation for virtual screening to increase the chance of discovering potent MGAT1 inhibitors. This combinatorial strategy allowed the development of W-GTF01 as a potent small molecule inhibitor that efficiently inhibits MGAT1, limiting CD73 membrane abundance by impeding its dimerization on the Golgi body. As a key enzyme in the N-glycosylation pathway, MGAT1 is critical for the proper folding, stability, and function of numerous glycoproteins, including membrane receptors and transporters. Inhibiting MGAT1 activity in vivo with compounds like W-GTF01 has demonstrated promising efficacy, particularly in immune-cold TNBC models, by promoting tumor regression through the revival of cytotoxic CD8$^+$ T cell responses. However, MGAT1's integral role in glycoprotein biosynthesis also raises concerns about systemic toxicity and off-target effects, as W-GTF01 binds to the UDP-binding domain shared by other GlcNAc-transferases. This could result in suboptimal glycosylation of essential proteins, impairing cellular signaling, and receptor-mediated interactions, ultimately leading to adverse physiological outcomes. To mitigate these risks, strategies such as antibody-drug conjugates (ADCs) and combination therapies that enhance selectivity and minimize off-target effects are being actively explored. Furthermore, comprehensive characterization of W-GTF01, including its toxicity, selectivity, and pharmacokinetics, is critical for therapeutic development. Future efforts should also focus on identifying biomarkers to monitor potential side effects and guide therapeutic interventions. These approaches will enable the development of more effective and safer MGAT1-targeting therapies, leveraging their therapeutic potential while minimizing unwanted biological consequences.

### Overactivation of MGAT1 by deregulated THBS1 signaling down-regulates CD8$^+$ cytotoxic T cell function and enhances tumor immune evasion

During tumor progression, the CD73-adenosine axis is thought to be a major contributor to the creation of an immunosuppressive environment that favors tumor expansion and distal migration. The binding of adenosine to A2AR on immune cells leads to the suppression of CD8$^+$ T cell function, thus destroying antitumor immune responses. Regulation of CD73 in the context of the etiological microenvironment is sophisticated and could happen at multiple levels. Previous studies demonstrated, at the transcriptional level, that the up-regulation of CD73 in response to tumor hypoxia or TGF-β signaling stimulates the transcription of CD73, resulting in accelerated CD73 catalysis of adenosine production[23,24]. Recent studies further revealed that membrane-bound CD73 protein accumulation can be achieved at the post-translational level, where CD73 protein turnover can be governed by the interplay between ubiquitination and deubiquitination of CD73[22,24]. While the basal level of CD73 is controlled by TRIM21-mediated proteolysis, OTUD4 could counteract ubiquitin E3 ligase and stabilize CD73 by removing ubiquitin conjugates attached to CD73, especially when TGF-β is overactivated[24]. Intriguingly, our work demonstrated that, in immune-cold tumors, glycosyltransferase MGAT1 can be hijacked by the immunosuppressive action of THSB1 that orchestrates Golgi body-mediated CD73 membrane translocation via rendering CD73 dimerization. Dimerization of CD73 facilitated by glycosylation is critical, enabling CD73 in the Golgi apparatus to approach the subsequent vesicle transporters such as VAMP3, RAB8A, and RAB13 for membrane translocation. Increased expression of MGAT1 induced by abnormal THSB1 causes severe accumulation of CD73 on the tumor surface that overproduces extracellular adenosine, extrinsically inhibiting CD8$^+$ T cell function and promoting tumor evasion. Our discovery of the THBS1-MGAT1 axis in modulating immune suppression through regulating CD73 dimerization provides a target for drug design. MGAT1 inhibitors that restore the IFNγ-producing CD8$^+$ T cell response could be an efficient way to augment the efficacy of current immunotherapeutic agents in treating immune-cold tumors such as TNBC. Consistent with our other analyses, our bioinformatic study revealed that IFNγ, TGF-β, and THBS1 are the upstream nodes that coordinate the balance of CD73 in immune suppression, with the signature of high CD73 expression appearing with increased THBS1 as well as TGF-β signaling in the context of TNBC. Furthermore, our GSEA analyses showed a correlation between high MGAT1 expression and increased THBS1 signaling activity, implying a cascade of THBS1-MGAT1-CD73 in maintaining a tumor-promoting immune microenvironment. In the context of TNBC, THBS1 could be secreted by tumor cells, cancer-associated fibroblasts (CAFs), and immune cells such as macrophages and dendritic cells, particularly in response to hypoxia or stress signals within the tumor microenvironment. Elevated THBS1 expression is often associated with immune suppression and poor clinical outcomes. Further studies are needed to determine the precise sources and functional contributions of THBS1 within MGAT1$^{high}$ and MGAT1$^{low}$ tumor regions. Mechanistically, how MGAT1 is elevated by THBS1 and how THBS1 is coordinated with other CD73 regulatory pathways, for instance, TGF-β and hypoxia, requires further study.

In summary, this study revealed a previously unidentified role for MGAT1 in regulating CD73 and its enzymatic activity to determine tumor immune suppression in the context of TNBC. Golgi apparatus-mediated CD73 membrane translocation is regulated by MGAT1-catalyzed CD73 glycosylation, leading to dimerization, which modulates adenosine-dependent immune suppression. Targeting MGAT1-mediated CD73 glycosylation, translocation, and dimerization, combined with PD-L1 blockade, appears as an effective strategy to treat TNBC, particularly when MGAT1 and CD73 are abundantly expressed.

## Methods

This research complies with all relevant ethical regulations, and all animal procedures were approved by the Institutional Animal Care and Use Committee of Northwestern University and Emory University.

### Bioinformatics analysis

Human breast cancer RNA-seq data were downloaded from the National Cancer Institute (NCI) Genomic Data Commons (GDC) database, and the PAM50 classification of these cancers was obtained from the cBioportal database. Immune subgroups for basal-like breast cancer (BLBC) were determined using the R package dendextend to cut the clustering dendrogram (Supplementary Fig. 1). Single sample gene set enrichment analysis (ssGSEA) was used to compare functional genesets in immune subgroups of BLBC. Wilcoxon rank-sum tests were used to perform statistical comparisons.

## Cell lines and cell culture

HEK-293T, Human Mammary Epithelial Cells (HMECs), 19 human breast cancer cell lines [American Type Culture Collection (ATCC) Breast Cancer Cell Panel (ATCC 30-4500K)], 4T1, and E0771 were obtained from the ATCC (Manassas, VA). The HEK-293T and MDA-MB231 cells were cultured in Dulbecco's Modified Eagle Medium supplemented with 10% fetal bovine serum [FBS; streptomycin (100 μ/ml) and penicillin (100 U/ml)] at 37 °C with 95% humidity. MDA-MB468 cells were cultured in L-15 medium supplemented with 10% FBS [streptomycin (100 μ/ml) and penicillin (100 μ/ml)] at 37 °C with 95% humidity. Normal (non-immortalized) human mammary epithelial cells (HMEC) (passage 2–5) were cultured in Mammary Epithelial Cell complete medium (basal medium plus growth kit) and maintained in a humidified 37 °C incubator with 5% $O_2$, $CO_2$, and $N_2$. Other breast cancer cell lines were maintained as per the manufacturer's protocol.

## Tissue microarray, immunohistochemistry, and image analysis

A breast cancer tissue microarray (TMA) containing 110 cores, including matched adjacent normal tissues (US Biomax, BC081116d), was utilized for immunohistochemical (IHC) analysis. Sections were deparaffinized, followed by blocking with 10% normal goat serum to minimize nonspecific binding. Primary antibodies targeting MGAT1 (Sigma-Aldrich, SAB1400165) and CD73 (Cell Signaling Technology, #13160) were applied and incubated overnight at 4 °C on a shaker.

After washing, sections were treated with biotin-conjugated secondary antibodies and developed using a Vectastain ABC kit (PK-6100, Vector Labs). DAB (DAKO) was used for chromogenic detection, with hematoxylin as the counterstain. Stained slides were scanned at 4x and 20x magnification using a Nanozoomer (Hamamatsu) under standardized conditions by a blinded investigator. Digital slide images (NDPI format) were analyzed semi-quantitatively using QuPath v0.2.0-m4.

## Plasmids and Transfection

The CD73 plasmids and vectors were purchased from Sino Biological (Human: HG10904-UT, Mouse: MG50231-UT). The MGAT1 plasmids were purchased from GeneScript (human MGAT1 (NM_001364392.2) ORF Clone and Mgat1_OMu04487D_pcDNA3.1+/C-(K)-DYK). The full-length and truncation mutant fragments of MGAT1 and CD73 were cloned into Flag/HA-tagged mammalian expression vectors by Gibson Assembly based on full-length or partial coding region. The human and mice CD73 with a point mutation at four n-glycan sites were constructed with a site-directed mutagenesis kit (Agilent) following the protocol. The sgRNAs in pLentiCRISPR v2 targeting human and mice genes were purchased from GeneScript and listed as follows, for human: MGAT1 CRISPR Guide RNA1_pLentiCRISPR v2 (gRNA sequence: CGTTGGGTAGAAAAATGGGC), MGAT1 CRISPR Guide RNA2_pLenti-CRISPR v2 (gRNA sequence: GTGGCGGTGGTGACGGGCTG), VAMP3 CRISPR Guide RNA1_pLentiCRISPR v2 (gRNA sequence: AGAGCGA-GACGTCCCCTCCC), VAMP3 CRISPR Guide RNA2_pLentiCRISPR v2 (gRNA sequence: GGCTCTGAAACGCAGAACGC), NT5E CRISPR Guide RNA1_pLentiCRISPR v2 (gRNA sequence: AAACCTTAGTGATAAAA-CAG), NT5E CRISPR Guide RNA2_pLentiCRISPR v2 (gRNA sequence: ATTGTGAATAAACAAGGAAC); for mouse: Mgat1 CRISPR Guide RNA1_pLentiCRISPR v2 (gRNA sequence: GCA-CACGCCGGCCCCGAAAC), and Mgat1 CRISPR Guide RNA2_pLenti-CRISPR v2 (gRNA sequence: GCGCATCCAGTCGTCCCAAA). The pcDNA3.1-GFP(1-10), pHRm-NLS-dCas9-GFP11×7-NLS, MCS-13X Linker-BioID2-HA were purchased from Addgene. HEK-293T cells were plated one day in advance at the confluence of 70% and transfected with specific plasmids together with PEI (Invitrogen).

## 2D and 3D coculture of cancer cells and immune cells

In the 2D coculture, the cancer cells were plated in 48 well plates and incubated overnight. The PBMC or purified CD8⁺ T cells were pre-activated with anti-CD3 and anti-CD28 antibodies and added into the well to initiate coculture. The images of specific regions were taken every 24 h with BioTek Lionheart Automated Microscope, and the confluence of cancer cells was analyzed by Gen5. After 72 h of coculture, the T-cell and cancer cells were harvested by washing and proceeded to flow cytometry analysis.

In the 3D coculture, the 3D spheroids of MGAT1-WT/OE/KD stable TNBC cell lines were constructed by centrifuging within U-shaped bottom 96-well plates with 5% Matrigel at 300 G for 10 min. The cancer cell spheroids were incubated for 72 h, and pre-activated PBMC cells stained with CellTracker Deep Red Dye (ThermoFisher) following the protocol were added to initiate coculture. After 72 h of culture, the infiltration of immune cells was visualized by Z-stack imaging with BioTek Lionheart Automated Microscope, followed by Z-projection.

## Lentiviral construction, generation, and Infection

The human MGAT1, CD73, Cd73-4NQ, and mice Mgat1, Cd73, and Cd73-4NQ were cloned into the pLENTI vector through Gibson Assembly Kits following protocol. At least four clones from each ligation were individually picked, followed by validation, including restriction enzyme digestion, sequencing, and transient transfection into HEK-293T cells. The clones that passed all validations were selected for transient transfection and stable line construction. To package lentivirus, the targeted expression plasmids were co-transfected with the lentiviral particle packaging plasmids (pVSV-G and psPAX2) and PEI. After overnight incubation, the medium with transfection reagents was replaced with a fresh medium. The virus was collected, followed by centrifuge and filtration with 0.45 μm filters after 24 and 48 h of culture. The filtered virus was utilized to transduce the targeted human and mice TNBC cancer cell lines with polybrene, and the stable cell lines were selected by the addition of specific antibiotics (puromycin, blasticidin, and hydromycin) at suggested working concentrations. The targeted protein expression was validated with WB.

## Purification of MGAT1 complex (affinity capture purification)

The WT MDA-MB468 (control) and MDA-MB468 stably expressing Flag/HA-tagged MGAT1 were washed with PBS and lysed by NP-40 buffer (1% NP40, 10% glycerol, 25 mM Tris-HCl [pH 7.9] and protease inhibitor cocktails) on ice. The Flag-MGAT1 and its binding partners were purified by immunopurification with Flag-M2 beads followed by three times washing with TBST buffer (137 mM NaCl, 20 mM Tris-HCl [pH 7.6], 0.1% Tween-20). The purified complex was eluted with 3 × Flag peptide in TBS buffer and validated with SDS-PAGE gel followed by silver staining following the manufacturer's protocol. The bind partners of MGAT1 were discovered by mass spectrometry.

## Enzymatic deglycosylation

The protein deglycosylation was conducted with Peptide N-Glycosidase F (PNGase F) and Endoglycosidase H (Endo H) from New England Biolabs following the manufacturer's protocol. Briefly, the cell lysis from indicated cells was mixed with water and Glycoprotein Denaturing Buffer (10X) and denatured by heating at 100 °C for 10 min. The samples were cooled on ice and further mixed with GlycoBuffer 2, 10% NP-40, $H_2O$, and PNGase F and incubated at 37 °C for 1 h. Sampling buffer with 2-mercaptoethanol was added to stop the reaction and the samples were separated on SDS-PAGE gel.

## Measurement of CD73 Activity

The targeted cells were plated at the same density in the same volume of medium. The medium with adenosine was harvested by 5 min centrifuge at 10,000 rpm to remove insoluble materials. The adenosine concentration was measured with an Adenosine Assay from Cell Biolabs following the manufacturer's protocol. Briefly, the adenosine is converted to inosine by adenosine deaminase (ADA), and inosine is converted into hypoxanthine by purine nucleoside phosphorylase

(PNP). Eventually, the hypoxanthine is converted to xanthine and hydrogen peroxide by xanthine oxidase (XO) and the concentration of hydrogen peroxide was measured with a highly specific fluorometric probe. A series of reagents with known adenosine concentration was utilized as references to generate standard, and samples without adenosine deaminase were utilized as controls.

## Immunoprecipitation analysis

Cells were lysed at 4 °C in ice-cold immunoprecipitation assay buffer (Cell Signaling Technology, 9806), and cell lysates were cleared by centrifuge at $13,000 \times g$ for 15 min. Concentrations of protein extract were determined with BCA assay following protocol. For the immunoprecipitation assay, cell lysate was mixed with anti-Flag M2 beads or IP-proved antibodies and incubated overnight at 4 °C on a rotator. After five washes with lysis buffer supplemented with protease inhibitor mixture, complexes were released from the anti-Flag M2 beads by boiling for 5 min in 2 × SDS-PAGE loading buffer, and the targeted proteins were detected with WB.

## Immunofluorescence

Cells were plated on the coverslip and incubated overnight before treatment. The cells were fixed with 4% paraformaldehyde in PBS for 10 min, and the fixed cells were permeabilized with 0.5% Triton X-100 for 15 min. The membrane staining of the indicated proteins was performed without membrane permeabilization. After one hour of blocking with blocking bugger, the cells were incubated with the indicated primary antibodies at 4 °C overnight. After three washes with PBST, cells were incubated with Alexa 488/594 IgG or STAR RED/STAR ORANGE from Abberior diluted in PBST for one hour. Cells were washed three times, mounted with UltraCruz DAPI containing mounting medium (Santa Cruz), viewed, and photographed under a Leica SP8 confocal Microscope or Abberior STED Super-Resolution Microscope.

## Duolink proximity ligation assay

The co-localization of MGAT1 and CD73 was investigated by Duolink In Situ PLA kit from Sigma-Aldrich following the manufacturer's protocol. In brief, the cells were seeded on a glass coverslip and incubated overnight. Then the cells were fixed with 4% formaldehyde and penetrated with 0.5% Triton X-100. After washing, the cells were blocked with 2% BSA, followed by overnight indicated primary antibody incubation on a shaker in the cold room. The cells were then incubated with PLA probes and ligation mix for 1 h at 37 °C. The amplification mix was added to cells and incubated for 2 h at 37 °C followed by three times of washing. In the end, the coverslips were mounted with Duolink® In Situ Mounting Medium with DAPI, and the images were captured with Leica SP8 confocal microscope.

## Split-GFP

The pcDNA3.1-GFP(1-10), pHRm-NLS-dCas9-GFP11 × 7-NLS, MCS-13X Linker-BioID2-HA were purchased from Addgene. The GFP and linker coding regions were cloned into pLENTI WT CD73/4NQ-CD73/dimer deficient CD73. The GFP(1-10)-CD73 and GFP11 × 7-CD73 plasmids were cotransfected into HEK-293T cells, and the GFP positive cells were detected by Leica Stellaris 8 live cell imaging confocal and flow cytometry with Cytek Aurora.

## Native gel immunoblot

The experiment was conducted following the protocol of the NativePAGE Novex Bis-Tris Gel System. Briefly, the native protein samples from indicated cells were collected with Native PAGE Sample Prep Kit with 1% DDM. The lysate was centrifuged at $20,000 \times g$ for 30 min at 4 °C. The supernatant was collected, and NativePAGE 5% G-250 Sample Addictive was added right before electrophoresis. The collected samples were separated with NativePAGE Electrophoresis system or Mini-PROTEIN TGX Precast Gel. For the NativePAGE Electrophoresis system, samples were loaded on NativePAGE 4–16% Bis-Tris 1 mm Mini Protein Gel and ran with Dark Blue Cathode Buffer until the dye front migration was 1/3rd of the gel. Then, the Dark Blue Cathode Buffer was removed with a pipet, and the buffer was replaced with the Light Blue Cathode Buffer before resuming the run. For Mini-PROTEIN TGX Precast Gel, samples were loaded on 4–15% Mini-PROTEIN TGX Precast Gel and ran with Tris/glycine running buffer without SDS. After gel running, the samples were further transferred to the PVDF membrane and detected after blocking, primary antibody and secondary antibody. The NativeMark Unstained Protein Standard was run on a native gel and detected with Coomassie blue staining as a reference.

## Plasma membrane and Golgi fractionation

Plasma membrane and Golgi fractions were prepared from discontinuous sucrose gradient centrifugation method as described previously[30–32]. Briefly, Cultured TNBC ($10 \times 10^6$ cells) were harvested by scraping and homogenized the cells with 0.25 M sucrose and 5 mM Tris-HCl pH 7.5 using Dounce homogenizer and initial centrifugation at t low speed ($400 \times g$, 12 min) and further the collected pellet was mixed with isotonic Percoll solution (20% (w/v) Percoll, 0.25 M sucrose, and 2.5 mM Tris-HCl pH 7.5. This suspension was spun at low speed ($1500 \times g$, 15 min) and collected the top floating membranes using a Pasteur pipette. The membranes were then pelleted in an ultracentrifuge ($100,000 \times g$, 1 h and characterized using sodium potassium adenosine triphosphatase as a plasma membrane marker. The supernatant of initial centrifugation was submitted to successive increasing the $g$ forces ($1500 \times g$ for 10 min, $10,000 \times g$ for 20 min, and $100,000 \times g$ for 1 h, 5 min) to obtain the fractions obtained were mitochondria, lysosomes, and peroxisomes, microsomes, and cytosol, respectively. The microsomes were collected and mixed with 1.3 M sucrose density gradient medium and overlaid with 1.15 M, 0.9 M, 0.6 M, and 0.25 M sucrose. After centrifugation for 1 h, 35 min at $100,000 \times g$ in a Beckman Optima MAX-XP ultracentrifuge and centrifuged at $100,000 \times g$ for 30 min, the interphases of middle layers were further pooled and centrifuged at $5000 \times g$ at 20 min to collect Golgi fraction, which was confirmed based on GM130 as Golgi marker.

## Structural modeling and protein-protein binding

Human MGAT1 (UniProt ID: P26572) protein model was constructed based on crystal structure of rabbit N-acetylglucosaminyltransferase I (GnT I)[34] (PDB: 2am5); and human CD73 (Uniprot ID: P21589) protein was generated after the crystal structure of a soluble form of human CD73 (PDB:4h1s)[35]. Both structural models were built using SwissModel[67]. MGAT1-CD73 protein complex models were generated using the docking software ClusPro[68]. Residues in the proximity of MGAT1 (L271-V370) and CD73 (L79-I128) were set as attractor sites of MGAT1 and CD73. We obtained 30 clusters of MGAT1-CD73 complexes, upon clustering ~900 models generated by ClusPro. The clusters were rank-ordered by cluster size[68] as recommended, and the binding free energies of MGAT1-CD73 were calculated using PRODIGY[69]. The docking pose with the largest cluster size was selected to conduct two independent 100 ns unbiased molecular dynamics (MD) simulations to further evaluate the stability of the predicted binding poses. NAMD package and CHARMM36m force field were used in MD simulations. MD simulations and trajectory analysis were carried out following our previous protocol[6].

## Modeling of CD73 glycoproteins and their binding to MGAT1

The Man5 glycans on the CD73 monomer or dimer were added to four ASN residues, N53, N311, N333, and N403, using CHARMM-GUI[70]. Active hMGAT1 conformer bound with UD1 and Mn[2+] was modeled after the resolved MGAT1 homolog (PDBs: 2am5 and 1foa). The binding poses of active hMGAT1 to CD73 glycoprotein were predicted in Supplementary Fig. 5b was generated using ZDOCK server[37], which is

based on a rigid-body docking algorithm. The binding probability of MGAT1 to the four ASN residues on CD73 were roughly estimated based on 40 computed complexes.

## Docking simulation and Druggability simulation

Ligand docking simulations were performed using AutoDock Vina[71]. Simulations were carried out using a grid with dimensions set to $64 \times 64 \times 64$ Å$^3$ to encapsulate the entire structure of MGAT1. The exhaustiveness of the simulation was set to 50, and the algorithm returned 20 binding poses of interest for each compound. Ligand binding affinity was computed using Vina. Druggability simulations[72] were carried out for MGAT1 in the presence of seven probes: isobutene, imidazole, acetamide, isopropanol, isopropylamine, acetate, and 4H-pyran. Preparation of druggability simulation was performed using the DruGUI[72] module implemented in VMD[73]. All-atom druggability simulations were carried out using NAMD with the CHARMM36 force field for proteins, the TIP3P water model, and the CGenFF force field[74] for the probes, following our previous simulation protocols[46]. The trajectories were analyzed using the DruGUI module of ProDy[75,76]. Three independent runs of 40 ns druggability simulations were carried out, following our previous protocol[46].

## Pharmacophore modeling and virtual screening of MGAT1 inhibitors

For the virtual screening of MGAT1 inhibitors, we constructed two pharmacophore models (PMs), namely PM_1 and PM_2, based on docking and druggability simulations, respectively. PM_1 was generated from the most energy-favorable binding pose of TW-37 predicted by Vina[71], and the high-affinity binding residues from MGAT1 were assessed using FTMap[42]. PM_2 was constructed based on druggability simulations following our previous protocol[46,72]. In particular, the snapshots from druggability simulations were analyzed using Pharmmaker[44], and the snapshot that exhibited the highest number of the most frequently occurring interaction pairs was selected as the template to construct the second pharmacophore model (PM_2)[46]. Those two PMs were then screened against MolPort libraries using Pharmit[43]. Data visualization was performed using PyMOL 1.8.6. (Schrödinger L. The PyMOL Molecular Graphics System, Version 1.8.4.2. (2016)).

## Tumor challenge

Sex as a biological variable. Our study exclusively examined female mice because the disease modeled is only relevant in females. BALB/C and C57BL/6 mice (8-week-old females) were sourced from Charles River Laboratories and acclimated in a temperature-controlled animal facility with a 12-h light/dark cycle for one week. Engineered 4T1 cells (Balb/C background) ($5 \times 10^5$) or E0771 cells (C57BL/6 background) ($1 \times 10^6$) with MGAT1 knock-down or over-expression were subcutaneously injected into the mammary fat pad of female BALB/C or C57BL/6 mice on day 0. Tumor volumes were measured along three orthogonal axes (x, y, and z) and calculated as xyz/2 every 2 to 4 days. At the experiment's endpoint, the mice were euthanized, and tumors were harvested and weighed. Tumors were processed to obtain single-cell suspensions by digesting with collagenase/DNase and passing through 70-μm pore strainers.

Mice after tumor cell injection were monitored based on measurement of body weight and observation for severity of clinical signs (e.g., presentation of ulcers/lesions). A monitoring log sheet was initiated by the person checking the tumor-bearing animals as identified with pink cards per the Policy on Medical Records for Rodents. Tumor volumes exceeding 1500 mm$^3$ were designated as the endpoint and recorded for survival curve analysis. Other humane endpoints for euthanasia include (i) Weight loss greater than 20% from their base weight; (ii) When the tumor significantly interferes with eating, drinking, ambulation, grooming, or sleep; (iii) Tumor ulcerations that are significantly pitted or cavitated; (iv) Body Condition Score (BCS) less than 2 and (v) BCS 2 which doesn't improve with supportive care. All animal procedures in this study were approved by the Institutional Animal Care and Use Committee of Northwestern University and Emory University.

## Flow cytometry staining

The single-cell suspensions underwent stimulation with phorbol 12-myristate 13-acetate (50 ng/ml), ionomycin (5 μg/ml), and brefeldin A (10 μg/ml) to induce IFNγ and TNFα production. To block non-specific antibody binding to Fc receptors on immune cells, the stimulated single-cell suspensions were incubated with anti-mouse CD16/32 antibody (2.4G2). Following this, live/dead dye and fluorophore-conjugated anti-mouse antibodies targeting cell surface markers were added. After PBS washing, cells were fixed with 1x Fixation/Permeabilization reagent (eBioscienceTM, 00-5223-56) for 20 min at room temperature. Subsequently, cells were washed twice with 1× Permeabilization/wash buffer (eBioscienceTM, 00-8333-56) and subjected to intracellular staining (ICS) with an antibody cocktail for 45 min at room temperature. Finally, cells were washed and suspended in phosphate-buffered saline (PBS) for acquisition using a three-laser CYTEK AURORA spectral flow cytometry or LSRII flow cytometry. Single staining of each antibody with corresponding fluorescence on splenocytes served as the reference for spectral unmixing (CYTEK). Data analysis was performed using either the OMIQ web application (https://www.omiq.ai/) or the FlowJo application, following previously described methods[77].

## Multiplex immunohistochemistry (mIHC)

Multiplex IHC was carried out on a TMA comprising 150 invasive triple-negative breast cancer samples (US Biomax). Staining was performed using the Opal multiplex system (AKOYA Biosciences) according to previously published protocols[78]. After deparaffinization and antigen retrieval with AR9 buffer, six iterative staining rounds were conducted. Each cycle involved sequential application of primary antibodies (targeting MGAT1, CD73, CD3, CD8, Ki67, and PanCK), HRP-conjugated secondary antibodies, and visualization with distinct Opal fluorophores.

Between each round, AR6 buffer was used to strip antibodies without disrupting existing fluorophores. Following staining, DAPI was applied for nuclear visualization. Spectral unmixing was carried out using single-marker control slides to assign fluorophores accurately. Details of antibodies and fluorophores are provided in Supplementary Data 3.

## Spatial analysis and quantification

The spatial interactions between PanCK$^+$ tumor cells and CD8$^+$ T cells or CD8$^+$Ki67$^+$ T cells were analyzed using the "Spatial Map Reviewer" tool within the R-based phenoptrReports software (AKOYA Biosciences). This tool was applied to quantify touching interactions between cellular populations in tissue microarray (TMA) sections comprising 41 specimens. The expression levels of MGAT1 in PanCK + tumor cells were quantified using the automated image analysis software inForm (AKOYA Biosciences). MGAT1 expression was stratified into high and low groups based on the median expression value, serving as an unbiased threshold for comparison. All analyses were performed on multiplex-stained TMA slides to ensure precise quantification of both MGAT1 expression and cellular interactions.

## Acquirement of multispectral images (MSI) and data analysis

The Opal fluorophore signals from the completed staining TMA slide were captured using the Vectra 3 Automated Quantitative Pathology Imaging System from Perkin Elmer at 200x magnification. These signals were then subjected to spectral unmixing, separating them into seven individual fluorophores based on their unique emission spectra,

utilizing the InForm Advanced Image Analysis software from Akoya Biosciences.

Following spectral unmixing, the resulting images underwent cell segmentation based on DAPI staining and cell phenotyping based on specific cellular markers, utilizing the trained algorithm within InForm. The data, including composite images, cell segmentation, and cell phenotyping results, exported from InForm, were subjected to further quantitative analysis of cellular densities, protein intensities, and interaction between CD8 T cells with tumor cells. The relative touching events were defined by the total touching events divided by the total number of MGAT1 positive or negative tumor cells. This analysis was conducted using R-based phenoptrReports & phenoptr software from AKOYA Biosciences.

To spatially profile gene expression in human triple-negative breast cancer (TNBC), we employed the GeoMX Digital Spatial Profiler (DSP) platform. TNBC tissue microarrays (TMAs) underwent deparaffinization, rehydration, and proteinase K digestion using a Leica BOND-RX system. Following that, in situ hybridization was performed using the Human Whole Transcriptome Atlas probes, with overnight incubation at 37 °C. After stringent washes, the TMAs were blocked with buffer R (Nanostring) and stained with a panel of morphology markers, including anti-PanCK (AF532), anti-CD73 (CoraLite594), anti-MGAT1 (with secondary AF647-conjugated IgG), and nuclear dye Syto13.

The DSP instrument was used to scan stained slides, from which 32 regions of interest (ROIs) were selected based on the co-localization of MGAT1, CD73, and epithelial marker PanCK. Tumor and non-tumor compartments were distinguished using PanCK-based segmentation. UV light was used to cleave barcoded probes from each segment, which were then collected into separate wells for library preparation and RNA sequencing, following Nanostring's protocols. After sequencing and quality control, differential expression analysis was conducted across high vs. low MGAT1/CD73 expression in tumor and stromal areas. Gene Ontology (GO) and GSEA were then used to interpret functional pathways. ROIs with expression levels above or below the median were categorized as high or low, respectively.

### Statistical analysis

Data are presented as mean ± SEM unless otherwise noted. Each experiment was independently repeated at least twice. Statistical analyses were conducted using SPSS (Chicago, IL), GraphPad Prism (v9), or R (v3.3.3). Significance was assessed via unpaired Student's $t$ test or one-/two-way ANOVA, followed by Tukey's post-hoc test for multiple comparisons. Survival and tumor-free intervals were compared using the Log-rank (Mantel-Cox) test. Correlations between variables were examined using either Pearson or Spearman methods, with $p$-values < 0.05 considered statistically significant.

### Reporting summary

Further information on research design is available in the Nature Portfolio Reporting Summary linked to this article.

## Data availability

The authors declare that all data supporting the findings of this study are available within the article and its supplementary information. The mass spectrometry proteomics data generated in this study have been deposited to the ProteomeXchange Consortium via the PRIDE partner repository with the dataset identifier PXD055157. The proteomic data utilized in this research are publicly accessible through Breast cancer proteomic data through the National Cancer Institute Clinical Proteomic Tumor Analysis Consortium (https://proteomic.datacommons.cancer.gov/pdc/; accession number: PDC000173). TNBC immune-related mRNA data utilized in this research can be accessed through the Gene Expression Omnibus database (accession number: GSE88847). Immune analysis using CAMOIP, TIMER2, and TIDE online platforms are available with the following links: https://www.camoip.net, http://timer.cistrome.org, and http://tide.dfci.harvard.edu/login/. Source data are provided with this paper.

## Code availability

Code generated for this manuscript and associated data was deposited in Github: https://github.com/Watson-40/MGAT1-Figure-1-code.

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

## Acknowledgements

We appreciate the Winship Cancer Institute Flow Cytometry Facility, Integrated Cellular Imaging Core (RRID:SCR_023534), Integrated Proteomics Core (RRID: SCR_023530), and Animal Resources Facility at Emory University School of Medicine (Winship Cancer Institute-NIH 5P30CA138292). We appreciate the proteomic core at the Feinberg School of Medicine Northwestern University for Mass Spectrometry Analysis and post-data analysis (NCI CCSG P30 CA060553 and P41 GM108569). This work was also supported by the Northwestern University Immunotherapy Assessment Core, Pathology Core Facility, RHLCCC Flow Cytometry Facility, and Animal Resources Facility (NCI CA060553). We appreciate Dr. Hanai Fu and Dr. Xiulei Mo's help on the Split-GFP protocol. We appreciate Kevin Li and Yikai Wang in sample preparation. We thank all members of the Wan, Zhang, and Bahar laboratories for their helpful discussion. This work was supported by the Emory SOM Endowed fund, and part of the effort of Y. Wan is covered by NIH R01CA258857, NIH R01CA258765, NIH R01CA250110, NIH R01CA202948, Winship Invest$ fund and Award of therapeutic discovery pipeline from Emory Center for New Medicines. Support from the NIH grants P01 DK096990 and R01 GM139297 is gratefully acknowledged by I. Bahar. This work was partially supported by NIH R01CA222963, R01CA250101, and R01CA258857 for B. Zhang.

## Author contributions

JJ. Chi: Conceptualization, formal analysis, investigation, methodology, drug discovery, writing-original draft, writing-review, and editing. P. Xie: Study of preclinical models. H. Cheng: Structural modeling and docking simulation analyses, virtual screening, writing-original draft, writing-review, and editing. Y. Zhu: Formal analysis, investigation, methodology. J. Watson: Bioinformatic analyses. L. Zeng: Construction for plasmids. X. Cui: Study of preclinical models. A. Uddin: Drug validation. H. Nguyen: Structural modeling and docking simulation analyses. M. Wyatt: Collection of purified CD8+ T cells. C. Massimo: Clinical data analysis and editing. W. Gradishar: Clinical data analysis and editing. M.A. Bhave: Clinical data analysis and editing. K. Kevin: Clinical data analysis and editing. C. Paulos: Collection of purified CD8+ T cells. L. Lei: Synthesis of MGAT1 substrate. M. Hung: Pre-clinical model and editing. M. Dai: Discussion for drug validation. K. Morman: MGAT1 enzyme protein purification. A. Reedy: Imaging. A. Marcus: Imaging and editing. S. Zhao: Bioinformatic analyses and editing. I. Bahar: Molecular dynamic analyses, drug development, review, and editing of the manuscript. B. Zhang: Data curation, review, and editing of the manuscript. Y. Wan: Conceptualization, drug discovery, data curation, supervision, funding acquisition, project administration, writing, review, and editing of the manuscript.

## Competing interests

The authors declare no competing interests.

## Declaration of Interests

W-GTF01 is under the process of patent application.

## Additional information

[1]Department of Pharmacology and Chemical Biology, Emory University School of Medicine, Atlanta, GA, USA. [2]Winship Cancer Institute, Emory University School of Medicine, Atlanta, GA, USA. [3]DGP graduate program, Northwestern University Feinberg School of Medicine, Chicago, IL, USA. [4]Department of Obstetrics and Gynecology, Northwestern University Feinberg School of Medicine, Chicago, USA. [5]Department of Medicine, Robert H. Lurie Comprehensive Cancer Center, Northwestern, University Feinberg School of Medicine, Chicago, IL, USA. [6]Laufer Center for Physical & Quantitative Biology, Stony Brook University, Stony Brook, NY, USA. [7]Department of Biochemistry and Molecular Biology and Institute of Bioinformatics, University of Georgia, Athens, USA. [8]Department of Chemistry, Georgia State University, Atlanta, USA. [9]Complex Carbohydrate Research Center, University of Georgia, Athens, USA. [10]Department of Hematology and Medical Oncology, Emory University School of Medicine, Atlanta, GA, USA. [11]Department of Surgery/Microbiology & Immunology, Emory University School of Medicine, Atlanta, GA, USA. [12]Department of Chemistry, Emory University School of Medicine, Atlanta, GA, USA. [13]Department of Medicine, Weill Cornell Medicine, New York, NY, USA. [14]Graduate Institute of Biomedical Sciences, Institute of Biochemistry and Molecular Biology, Cancer Biology and Precision Therapeutics Center, and Center for Molecular Medicine, China Medical University, Taichung, Taiwan. [15]Department of Biotechnology, Asia University, Taichung, Taiwan. [16]Department of Biochemistry and Cell Biology, School of Medicine, Stony Brook University, Stony Brook, NY, USA. [17]These authors contributed equally: Junlong Jack Chi, Ping Xie, Mary Hongying Cheng.
✉e-mail: bahar@laufercenter.org; bin.zhang@northwestern.edu; yong.wan@emory.edu

