## [Transparent Peer Review file · Nature Communications]

MGAT1-Guided Complex N-Glycans on CD73 Regulate Immune Evasion in TNBC

Corresponding Author: Professor Yong Wan

Version 0:

Reviewer comments:

Reviewer #1

(Remarks to the Author)

-----This manuscript provides evidence in support of the proposal that modification of the N-glycans on CD73 by MGAT1 is critical for the dimerization of CD73 and interaction with VAMP3 which facilitates its translocation to the membrane and production of adenosine which produces an immunosuppressive environment leading to increased tumor growth and progression. The authors have characterized their reagents in some depth and tested their hypothesis from several angles. However, there are important key questions that need to be addressed to more fully support the authors' hypothesis:

1. Are the branched N-glycans produced from MGAT1 responsible for any or all of the consequences of MGAT1 KD or OE? The authors should prevent branching by treating cells with kifunensin and test the effects on CD73 dimerization and adenosine production.
2. Does overexpression of MGAT1 with a point mutation that reduces activity but not medial Golgi localization have reduced effects on CD73 dimerization and adenosine production? See PMID: 11467936.
3. Do monomeric forms of CD73 produced from control versus Mgat1 null cells form dimers in vitro? That is, does CD73 with branched N-glycans form dimers whereas CD73 with Man5GlcNAc2 at the 4Ns does not?
4. Do cells producing only monomeric CD73 form fewer tumors?
5. The CD73-4NQ mutant is not a good test molecule as it does not mimic CD73 from an Mgat1 null cell, is trapped intracellularly and probably its misfolding causes ER stress. Are CD73 KO or KD mutant cells defective in tumor formation?
6. The data implicating VAMP3/THSB1 and 2 and adenosine should be in the main text, not just in the supplement. These aspects of the hypothesis are referenced in the abstract and highlights and should be supported by robust evidence in the main text.
7. Additional concerns that should be addressed:
 - a. The title is very broad and should be more focused on the empirical findings.
 - b. Is there evidence in immune-cold tumors that CD73 is a dimer and carries complex N-glycans?
 - c. "MGAT1 on Golgi bodies." Is an inappropriate term. Change to "MGAT1 in the Golgi compartment" or something similar throughout the text.
 - d. The reaction catalyzed by MGAT1 which initiates of branched N-glycan synthesis leading to complex/branched N-glycans should be shown and the hypothesis that branched N-glycans are necessary for CD73 dimerization should be clearly stated and tested.
 - e. "Glycosylation of CD73 in the cytosol determines its membrane abundance by regulating CD73 translocation." This statement is grossly inaccurate. The catalytic domain of MGAT1 operates in the luminal, juxta-membrane region of the medial Golgi, not in the cytosol.
 - f. "complex N-linked glycan structures." Should be simply "complex N-glycans" throughout the text.
 - g. "The addition of N-acetylglucosamine (GlcNAc) to growing glycan chains catalyzed by MGAT1 ensures the completion of glycan chain assembly on target proteins," This is inaccurate. MGAT1 transfers one GlcNAc to a specific Man in the substrate Man5GlcNAc2Asn which allows the removal of two Man residues and the subsequent action of several branching GlcNAc-transferases (MGAT2, MGAT3, NGAT4, MGAT5). MGAT1 does not add GlcNAc to growing glycan chains.
 - h. Fig 1B, C – define red and blue categories in the legend.
 - i. Fig. 1D – while MGAT1 is at the top, other GlcNAc-transferases are not far below. All GlcNAc-transferases contribute to increased branching and potentially affect tumor formation and progression similarly to MGAT1.
 - j. Fig. 1F – MGAT1 signal seems most consistently high in Her2+ rather than TNBC tumors. Need to quantitate images and normalize to actin. Gel is not convincing.

- k. Ref 35 Flynn et al is not cited appropriately.
- l. Fig. 2C. Are histograms meant to show “%IFN γ CD8 T cells” or “%IFN γ in CD8 T cells” as labeled? Should “in” be removed from labels of all 4 histograms? Same comment for Fig. 2E and 2G and later figures in which the % cells expressing a certain cytokine are what is being measured.
- m. Fig. 2D. Taxes Red should be Texas Red.
- n. Fig. 3A. Any of the many, many proteins in the silver-stained gel could be a “partner” of MGAT1. Showing co-IP with very OE proteins is not a proof of specificity. What are the appropriate controls? This story is a tale of cherry-picking and following the positive results.
- o. Fig. 3F. CD73 seems to be expressed throughout the cell. It is not confined to the Golgi and not only found in association with MGAT1.
- p. Fig. 3I. What are the “big datasets” from which these data were obtained?
- q. Fig 4D. Why is there a double band for IP anti-V5(MGAT1) and IB anti-V5?
- r. Fig. 5A. KD of MGAT1 only partially converts CD73 to lower MW in both cell lines. Should do MGAT1 KO to validate.
- s. Fig. 5H-J. Removing 4 N-glycans in the 4NQ mutant traps CD73 in the cell – probably in the ER due to misfolding. This is quite different from changing the N-glycans from complex to high mannose in the medial Golgi. Cannot interpret the data in Fig 5J in relation to dimerization. In addition, the quality of the images is too poor to publish. Figure label is MGAT1 KO but should be MGAT1 KD.
- t. Fig. 6F “Several compounds, including No.2, No.8, No.9, and No.14, showed potent inhibition of MGAT1 catalytic function(Figure 6F).” Inhibition is only a 60% reduction in activity at most. Heterozygote Mgat1 mice have 50% enzyme and are viable whereas Mgat1 null mice are embryonic lethal. Only 50-70% inhibition of enzyme activity is obtained for the 3 best compounds at a concentration of 10 μ M which is very high (Fig. 6K). The “best” inhibitor W-GTF01 needs to be characterized in activity and dimerization assays like Nos.2, 8, and 9. There are no data that W-GTF01 is “best”. Also, no evidence that it is specific for MGAT1. If the inhibitors compete with UDP-GlcNAc they would be expected to inhibit any GlcNAc-transferase. The authors should discuss this point.
- u. Fig. 7. Were No. 2, 8, or 9 inhibitors tested in the tumor formation assay? If so, the data should be included.
- v. Fig. 8. It is not clear why the authors highlight neutrophils as TILs responsive to low MGAT1. Data for neutrophils are not significantly different in DH vs DL tumors or MGAT1 OE vs KD tumors.
- w. Discussion. “enabling CD73 on the Golgi apparatus” sounds like CD73 is facing the cytosol. Should be “enabling CD73 in the Golgi apparatus” as most of CD73 faces the lumen of the Golgi compartment. Applies throughout the manuscript.
- x. Supplement. Fig. S3B has MGAT1-OB instead of OE.
- Check that all supplementary figure panels are referenced in the main text.
- Fig. S4E “Spearman’s rank correlation analysis shows the MGAT1 protein expression is highly positively correlated with the THBS1 an THBS2 signaling pathways.” Data from which the correlation analysis was performed should be described. Also there is no evidence for the source of THBS1/2 in the tumor environment.

Reviewer #2

(Remarks to the Author)

In this study, the authors investigate how the glycosyltransferase MGAT1 participates in regulating immunosuppression by influencing the glycosylation of CD73. They demonstrate that in immune-cold triple-negative breast cancer (TNBC), MGAT1 expression is closely associated with tumor immune evasion. Mechanistically, MGAT1-mediated glycosylation promotes CD73 dimerization, enhances its binding to VAMP3, and increases CD73 expression on the tumor cell membrane, which in turn suppresses the activation of anti-tumor CD8+ T cells. Furthermore, the authors identified a small molecule inhibitor, W-GTF01, through high-throughput drug screening, which inhibits CD73 glycosylation and dimerization. The combination of W-GTF01 with PD-L1 antibody therapy effectively suppresses the growth of immune-cold breast cancers. Thus, authors conclude that MGAT1 catalyzes glycosylation which is critical for CD73 membrane translocation and immunosuppression of TNBC cells.

This is an interesting paper that tries to address an important question in the field. However, some of the conclusions are not well supported by the data and there are multiple issues that authors need to address before publication.

Overall Comments:

1. In Figures 1E-F, immunohistochemistry suggests that MGAT1 expression is higher in TNBC than in other breast cancer subtypes. However, in breast cancer cell lines, MGAT1 expression is not high and is significantly lower in TNBC cell lines compared to HER2+ cell lines. This inconsistency needs to be addressed.
2. The manuscript predominantly focuses on TNBC as an immune-desert environment and correlates MGAT1 expression with immunosuppression in TNBC. However, the co-culture experiments were conducted using HER2+ cell lines, with no experimental validation using TNBC cell lines. This limits the study’s relevance to TNBC, and additional validation with TNBC models is necessary.
3. The paper lacks schematic diagrams illustrating the knockdown and overexpression results of MGAT1 in both mouse and human cell lines. These results should be included to support the conclusions drawn from the experiments.
4. The data presented in Figures 3D-E do not provide sufficient evidence to confirm the colocalization of MGAT1 and CD73 in the Golgi apparatus. Additional experimental data or higher-resolution images are needed to substantiate this claim.
5. The co-immunoprecipitation experiment shown in Supplementary Figure S3H only includes the IP results. The input data is missing and should be provided to fully validate the findings.
6. In several instances, the loading controls in the Western blot data are faint and unclear. The clarity of these bands should be improved for better interpretation.
7. The E0771 cell line name is repeatedly misspelled as “EO771” throughout the manuscript. This typographical error should be corrected.

Reviewer #3

(Remarks to the Author)

The authors describe the identification of MGAT1, a glycosyltransferase, as a key factor in immune evasion in immune-cold tumors, particularly in triple-negative breast cancer (TNBC). The authors show that MGAT1 overexpression promotes CD73 translocation, suppressing CD8+ T cell activity. The authors show that MGAT1 adds N acetylglucosamine to CD73, which facilitates its dimerization and membrane fusion. Finally, the authors present a new inhibitor called W-GTF01, which it is shown to blocks MGAT1 activity thereby restoring immune response and improving anti-PD-L1 therapy in TNBC models. The manuscript is well written, and the results are well documented.

However I have the following remarks:

Major remarks:

Figure 1:

- Figure 1A: the authors poorly describe the figure, assuming that the reader will understand where the data is. This is a complex panel that needs to be properly explain for the reader to understand the rest of the figure. For instance, the clusters mentioned in the results are not clear to see, since many proteins are shown and many seem to be elevated according to the results. Please explain what is what and guide the reader.
- Figure 1B: indicate what is red and blue, now it only says category. It is explained in the text, but the figure should speak by itself.
- What is BLBC? Don't expect the reader to know.
- Figure 1D: explain what the correlation is against immune suppression genes, how is this list obtained? Manually or through the analysis?
- Figure E: Basal is missing, are samples available? If so, it should be included.
- Figure 1G: how were the interactions quantified? How many samples were analysed? How were the levels of MGAT1 assessed to divide into high and low?

Figure 2:

- Figure 2B: what are the green cells? Are these really the MDA-MB468-MGAT1-KD as indicated in the legend? It looks as if the cells are cultured in the presence of a Caspase 3/7 substrate that when cleaved due to the ongoing apoptosis, the cells turn "green" fluorescent? This is a typical co-culture experiment to show that the OE and KD are differentially killed by activated T-cells. Please clarify.
- Figure 2B: the cell proliferation curves don't seem to correspond to the images. The OE cells are not proliferating at the same rate as the WT (as it seems the case in the quantification) according to the images. Please clarify.
- Figure 2F vs 2B: what is the difference between these two experiments? In 2F the T-cells are activated? But the outcome is the same?

Figure 3:

- Figure 3I: what does "big datasets" in line 261 of the text mean?
- Figure 3K: what do CD73 and MGAT1 Area mean? What samples were measured? Please clarify.

Figure 5:

- In line 314 of the main text, the authors state that MGAT1 KD resulted in a reduced CD73 glycosylation. While this seems to be true for MDA-MD468 cells, it does not seem to be the case for MDA-MB231, or at least, there seems to be equally much CD73 glycosylated and non-glycosylated levels. Can the authors speculate about these cell-specific differences?
- Figure 5D: what is the membrane dye for? Please clarify what is the meaning of this observation in the text.
- In line 376 of the main text, the authors again elude to a Spearman's rank correlation, but it is totally unclear what data was used. Please clarify.

Figure 6:

- What exact compound library was used for this screen? Is it an "in-house" library? A repurposing library? Please clarify.
- What is known about TW-37? Any references to its specificity and uses? Please provide references.
- Why trying to identify other compounds based on the structure of TW-37? What was the rationale? Please clarify.
- Figure 6L: Compounds No.2, No.8 and No.9 induce a very different response, however they were identified as hits. Can the authors elaborate?
- Figure 6M: have the authors run this experiment for compounds No.2 and No.9? If so, the results should be shown too.

Figure 7:

- In line 478 of the main text, the authors conclude that there is an increased TAM presence in the in vivo experiment with MGAT1 OE cells, and that this is a sign of a pro-inflammatory M2-like macrophage response. However, the levels of IFNg+ and TNFa+, two pro-inflammatory cytokines are reduced. Can the authors please clarify or elaborate?
- Figure 7K: why did the authors measure Tim3+PD1+ in CD8+ Tcells? Please elaborate. It's just that it comes a bit as a surprise and there is no explanation.
- Figure 7Q: these results are remarkable. Did the authors measure the amount of exhausted T-cells on the mice treated with W-GTF01+anti-PDL1?

Figure 8:

- What data was used for Figures 8H to 8I?

Minor remarks:

- Figure 2A: in my personal opinion, the comical cells are unnecessary and don't bring rigor to the manuscript, on the contrary.
- Figure 3A: the mass spectrometry results are impossible to read, suggestion to make them bigger and put them in the supplementary
- Figure 5H comes after I in the text, suggestion to consider swapping.
- Line 392, it is not needed in a screen result to say that the hit was compound number 72...it could have been compound number 450 and it would have still been irrelevant. The important here is the TW-37 compound name.

Reviewer #4

(Remarks to the Author)

In the manuscript titled 'MGAT1-Mediated Glycosylation Orchestrates Immune Checkpoints and Antitumor Immunity', the authors investigate MGAT1/GnT1, a glycosyltransferase involved in CD73-mediated inhibition of CD8+T cells in immune-cold triple-negative breast cancer. The authors performed bioinformatic analysis of a subset of the TCGA database, and found that a number of enzymes in protein N-glycosylation pathway, particularly MGAT1, are associated with unfavourable tumor immune responses and poor prognosis in immune-cold breast cancers. Next, the authors co-cultured PBMCs with a breast cancer cell line overexpressing or knocked down for MGAT1 and conducted flow cytometry analysis. They found that MGAT1 expression in cancer cells modulates the expression of pro-inflammatory genes, including TNF α and IFN γ in CD8+ T cells. Confocal microscopy analysis further demonstrated the colocalization of MGAT1 and CD73 in the Golgi apparatus. Based on these findings, the authors explored the molecular mechanism of MGAT1 and CD73 interactions using in silico docking and simulation. They validated the predicted MGAT1-CD73 binding and proposed a model of the complex. Notably, the authors discovered that MGAT1-mediated glycosylation of CD73 promotes its dimerization and localization to plasma membrane. Finally, the authors screen a small compound library and identified W-GTF01 as a potent inhibitor of MGAT1 activity.

Protein N-glycosylation is prevalent in most membrane receptors and transporters. MGAT1/GnT1 is a key glycosyltransferase in N-glycosylation biosynthesis pathway, controlling the conversion of oligomannose glycan to hybrid/complex N-glycan structures. Previous studies have underscored the importance of MGAT1/GnT1 in various diseases. The loss of function mutations of MGAT1/GnT1 leads to aberrant glycosylation of cell surface receptors and transporters, resulting in dysregulated down-stream pathways. Here, the authors focus on elucidating MGAT1-mediated glycosylation of CD73 and investigate its biological consequences. In summary, these results links MGAT1 activity and CD73 glycosylation to immune checkpoints and antitumor immunity. I find this work is not suitable for publication in Nature Communications in current version.

General points.

1. CD73 is a nucleotidase with at least three N-glycans. Therefore, it is a plausible substrate for MGAT1. However, the authors presented some conflicting results regarding the proposed interface of MGAT1- CD73 interactions (Figure 4E, 4F 4G and 4H). The authors should re-analysis the in silico predictions of MGAT1-CD73 interaction. If MGAT and non-glycosylated CD73 can form a stable complex (Figure 4F and 4G), does this imply CD73 is an essential component of the glycosylation machinery? Does non-glycosylated CD73 exist in cells? Is the glycan conformation rigid in Figure 4H? Furthermore, do the authors have any control for their in silico predictions? Have they performed AlphaFold prediction of the truncated MGAT1/CD73 (Figure 4A and 4C). Can the truncated MGAT/CD73 maintain their native structures and conformations?
2. MGAT1 is the key enzymes in protein glycosylation pathway. Inhibiting its activity in vivo leads to dysregulation of various membrane receptors and transporters across all cell types. The authors should expand their discussion about the biological consequence of inhibiting MGAT1 activity using W-GTF01.

Major points

1. Figure 3A, there are so many other bands darker than the CD73 band in the right lane (MGAT1-OE). Is there any other glycotransferase or immune-response related protein that co-precipitated with MGAT1? The quality of MS/MS spectra can be improved. Did you only observe one peptide from CD73 and one peptide from MGAT1? How many replicates did you repeat?
2. MGAT1 facilitates N-glycosylation across a range of glycosylated membrane receptors and transporters. Did you observe any other highly glycosylated membrane receptors involved in immune check point and antitumor immunity in the TAP experiment? For example, CD39 is also highly glycosylated and it is in the same pathway with CD73.
3. It is essential to perform a glycoproteomics experiment of determine the glycan types in CD73, and confirm that its glycosylation is regulated by MGAT-KO and OE.
4. It would be beneficial to clarify the reason for using two different software, Cluspro and ZDOCK for the prediction of MGAT1-CD73 complex.
5. Figure 4F, is this the best RMSD from one MD simulation or averaged RMSD from several MD simulations?
6. Line 304, how do you know all four Asn residues can be glycosylated by MGAT1? The proximity of two protein in space does not necessarily imply a catalytic reaction. Additionally, why does ZDOCK predict four different complex structure? Does this mean that all four structures are plausible or that none of them are significant?
7. Figure 5E, there appear to be two contradictory results in a single gel. The middle lines (MGAT1-OE enhances CD73 dimerization) suggests that CD73 carries oligomannose type N-glycans and the overexpression of MGAT1 promotes the formation of complex N-glycans in CD73. However, the right lanes (MGAT1-KD suppresses CD73 dimerization) indicate CD73 carries complex-type N-glycans, and that MGAT1-KD results in oligomannose type N-glycan in CD73.

8. Can two well-known tool compounds, namely kifunensine and swainsonine also inhibit CD73 dimerization?

Reviewer #5

(Remarks to the Author)

The study focuses on a glycosyltransferase, MGAT1, which the authors identify as a regulator of CD73 glycosylation. This regulation drives the transport and dimerization of CD73 at the cell surface, enabling its activity and the generation of adenosine, which leads to immunosuppression. Notably, the study includes the development of an inhibitor that disrupts this interaction, preventing MGAT1-CD73-mediated immunosuppression. This is a key highlight of the research; however, data supporting the inhibitor's effects should be further validated. Overall, the study is comprehensive, encompassing molecular biology, biochemistry, as well as preclinical and clinical validations. While the data is compelling, additional analyses and revisited claims are needed before it can be considered for publication.

Major comments:

- In the introduction the authors should contextualize better TNBC as one of the most aggressive subtypes due to lack of therapies (in general, not only immunotherapy). Indeed, HER2 BC is more aggressive but has efficient therapies reducing the mortality and aggressiveness of this subtype. Authors should also mention that immunotherapy has only been approved in TNBC, but not in the other subtypes since these are immune cold tumors. TNBCs are not as cold as the rest of BC tumors.

- Authors should discuss and consider in their statements why MGAT1 is more expressed in TNBC rather ER+BC which is much colder than TNBC. It could help to provide a similar analysis as in Fig.1b including all type of BC patients.

- In fig.3 please show all candidates obtained with the MS data alongside CD73.

- In order to validate that MGAT1 influence adenosine synthesis through its interaction with CD73. MGAT1 OE and KD experiments of Fig.3 should be also done in CD73-KD cells.

In addition, all functional experiments should include this condition of CD73-KD to validate that MGAT1 is mediating all the claimed effects through CD73 or not. This is partially addressed with CD73-4NQ in in vivo experiments, however its own CD73-4NQ control is missing.

- Importantly, the effect of the inhibitor should be shown alongside the CD73-KD conditions to claim that is mediated to CD73 and not by other MGAT1 effects or by other targets of the designed inhibitor (i.e. teste whether it also affects Bcl-2,...). Indeed, it would be strongly recommended to perform RNA-seq of the treated cells with the inhibitor to have a full picture of transcriptomic changes mediated by the inhibitor.

- CX3CR1 reflects a transitory T cell effector state and CD101 a differentiated exhausted state, thus authors should show percentages of these combined populations, since the increase of CX3CR1+ CD8 T-cells do not support their conclusions.

- The immune-checkpoint experiments in fig.7 should include survival events. In addition, waterfall plots would help the interpretation of the data.

- The immune-checkpoint experiments in fig.7 should include gain-and-loss of function conditions with MGAT1-KD/OE and CD73-4NQ (or KD).

Minor comments:

- Indicate category legend in fig. 1B

- Fig. 1G not very convincing data. Please indicate better the number of tumors, and the number of regions per tumor analyzed.

- Fig. 1I is not novel since the differences are likely due to high or low CD8 as known in the literature. Show better the supp data fig. 1 with MGAT1 high vs low CD8 infiltration and the prognosis of MGAT1 high and low

- Fig. 1F indicate better cell lines per subtype. In addition, authors should discuss why HER2 BC show higher expression of MGAT1.

- Fig. 2B indicate % of tumor cell survival.

- Fig. 2 should be addressed with an additional TNBC model

- Fig. 6K should include the analysis CD8 T cell activation with Granzyme B to be consistent with previous figures.

Additional activation markers would also reinforce the data such as CD69 or 41BB. Again, coculture tumor cell survival should be shown.

- In line 515, include a header for the section of spatial analysis of clinical samples

- Tittle change: change "orchestrates immune checkpoints" to "modulated CD73"...or similar.

Version 1:

Reviewer comments:

Reviewer #1

(Remarks to the Author)

The authors have responded to each of my concerns and added considerable new data to support their conclusions. However, on re-reading the manuscript, I was struck by the awkwardness of the logic. The focus of the manuscript is tumor immune evasion and, while MGAT1 did top the list of molecules to target, the manuscript could just as easily focus on CD73. In fact, CD73 provides a specific focus and mechanism whereas MGAT1 can be replaced by swainsonine or kifunensine, and most probably by the deletion of any of a number of N-glycan branching GlcNAc-transferases. A 2024 paper on CD73 in TNBC and immune evasion must be quoted (J Clin Invest (2024) 10.1172/JCI180914). This paper shows that CD73 levels controlled by ubiquitination are essential to its functions in tumor immune evasion. The present manuscript makes a strong case that the complex N-glycans on CD73 are essential to its functions in immune evasion, a complementary and elegant story. The authors should rewrite this manuscript in the context of the published JCI paper and any other literature on CD73 in breast cancer. A search for "CD73 TNBC" in Pubmed brings up 27 papers. All literature on CD73 in cancer should be accounted for in the context of the authors' findings and with respect to the novel role for complex N-glycans on CD73 they have uncovered. The main contribution of the present manuscript is to show that complex N-glycans on CD73 (generated by MGAT1 (or MGAT2, MGAT4, MGAT5)) are important regulators of its role in tumor immune evasion via inducing the dimerization of CD73 and its plasma membrane localization. MGAT1 is in several senses a red herring as a "regulator" and a focus. It is the complex N-glycans themselves that regulate CD73. In trying to simplify the narrative, the authors have instead made it much more difficult for cancer biologists to comprehend in relation to the big picture. It is important that the literature be presented as clearly as possible in this complex area of cancer biology. The title should reflect the central role of CD73 in immune evasion and the importance of its complex N-glycans. Thus "Complex N-glycans on CD73 Regulate Immune Evasion in TNBC" would be an appropriate title. Evidence that complex N-glycans promote tumor growth and tumor progression/metastasis has been published over the last 30 years, particularly by J. W. Dennis and colleagues. This literature should be appropriately discussed.

Reviewer #2

(Remarks to the Author)

The authors addressed all point raised, and did good work on the revised manuscript.

Reviewer #3

(Remarks to the Author)

I thank the authors for addressing my remarks and concerns.

Reviewer #4

(Remarks to the Author)

The authors have addressed all my previous comments. However, I regret to note that the first author has graduated and taken a new position, making glycoproteomics experiments unavailable. Instead, the authors performed PNGase F and Endo H digestions to investigate the glycosylation status of CD73 under MGAT-KO and OE conditions. They also conducted point mutations to validate the presence of complex-type N-glycans at four potential glycosylation sites.

One further comment: the quality of the mass spectra in Figure 3a is still poor. The authors should re-plot these spectra to improve clarity."

Reviewer #5

(Remarks to the Author)

The authors have addressed many of the concerns raised by this reviewer and others. As highlighted in my previous revision, a key novelty of this study is the development of an inhibitor targeting MGAT1-CD73-mediated immunosuppression. Part of the newly provided data reinforce this message; however, some minor concerns still persist.

- Based on the new data presented in Figure 1, which compare this mechanism across various BC subtypes, the authors should clarify whether the mechanism studied is specific to TNBC. It does not appear to be significantly enriched in immune-cold tumors, such as ER+ BC, thus this distinction should be explicitly stated.
- As previously requested, the authors should elaborate on the criteria used to select CD73 from among other interacting partners. Additionally, they should reference the table containing the list of proteins in the main text and provide examples of other immunosuppressive proteins, since it is mentioned in the text. The table should also include FDR values, as CD73 is listed at position 226.
- The new data in Supplementary Fig. 4e should be integrated into Fig. 3H. The functional assays lack a CD73-KD condition, which is necessary to evaluate whether the moderate changes in adenosine concentration result in functional effects.
- The new data presented in Supplementary Fig. 9g should be incorporated into the main figures for better visibility and emphasis.
- The combined T-cell state populations for different CX3CR1+/- vs. CD101+/- combinations should be shown. These results

appear inconsistent with the authors' claims, as CX3CR1 is not a marker of exhaustion and is shown to be higher with MGAT1-OE.

- Statistical analysis is missing in Fig. 7q and should be included.
- Finally, the title of the study overstates its findings. This could lead to confusion regarding the scope of immune checkpoints tested, which have not been tested in the study.

Version 2:

Reviewer comments:

Reviewer #1

(Remarks to the Author)

The authors have addressed my comments appropriately.

Reviewer #4

(Remarks to the Author)

The authors have addressed all my comments.

Reviewer #5

(Remarks to the Author)

The authors have addressed all my concerns. The manuscript has been improved.

Response Letter

Dear Reviewers,

We greatly appreciate your comprehensive evaluation of our manuscript, including both the positive feedback and the thoughtful critiques, technical concerns, and mechanistic questions. Over the past three months, our team has made extensive efforts to address your comments and enhance the quality of our manuscript. The revised version now includes **8 main Figures (89 panels)** and **14 Supplementary Figures (93 panels)**, with additional experimental data to directly address the concerns raised by the reviewers. These new findings have further solidified our conclusions regarding: (1) Multiomic spatial profiling reveals distinct tumor-immune interactions associated with MGAT1 and CD73, offering insights into mechanisms shaping immune-cold phenotypes. (2) MGAT1-catalyzed GlcNAc addition to CD73 at the Golgi is essential for its dimerization and membrane translocation, driving immune evasion through elevated adenosine production in immune-cold TNBC. (3) THBS1 regulates the MGAT1-CD73-adenosine axis by orchestrating tumor-stroma interactions, promoting immune suppression and highlighting its therapeutic relevance. (4) Clinically, the MGAT1^{low}/CD73^{low} signature correlates with favorable immune profiles and better outcomes in breast malignancies, emphasizing its translational significance. (5) W-GTF01, a novel MGAT1 inhibitor, disrupts CD73 dimerization, restores CD8+ T cell function, and sensitizes tumors to anti-PD-L1 therapy. In addition to these experimental advancements, we have revised the manuscript to enhance clarity, address all reviewer comments comprehensively. Below, we provide a detailed ***point-by-point*** response to each reviewer's comment:

Reviewer #1: Pages 2-36

Reviewer #2: Pages 37-47

Reviewer #3: Pages 48-70

Reviewer #4: Pages 71-93

Reviewer #5: Pages 95-116

To avoid any confusion with the Fig. numbers, which have changed in the revised version, all new Fig. numbers in the descriptions below are shown in **red** text.

Reviewer #1, General Comments:

This manuscript provides evidence in support of the proposal that modification of the N-glycans on CD73 by MGAT1 is critical for the dimerization of CD73 and interaction with VAMP3 which facilitates its translocation to the membrane and production of adenosine which produces an immunosuppressive environment leading to increased tumor growth and progression. The authors have characterized their reagents in some depth and tested their hypothesis from several angles. However, there are important key questions that need to be addressed to more fully support the authors' hypothesis.

We appreciate the Reviewer #1's recognition of our findings and the acknowledgment of our multi-angle approach to addressing the role of MGAT1-mediated N-glycan modification of CD73 in tumor progression and immune evasion. We also value the reviewer's suggestions to strengthen our study. Below, we provide a detailed response to the key questions raised, supported by additional experiments and analyses that further validate our conclusions.

Reviewer #1, Specific Comments:

Question #1 - *Are the branched N-glycans produced from MGAT1 responsible for any or all of the consequences of MGAT1 KD or OE? The authors should prevent branching by treating cells with kifunensin and test the effects on CD73 dimerization and adenosine production.*

We appreciate the reviewer's suggestion to further investigate the role of branched N-glycans produced by MGAT1 in CD73 dimerization and adenosine production. To address this, we performed additional experiments using TNBC cell lines (MDA-MB231 and MDA-MB468) treated with kifunensin and swainsonine, two well-established glycosidase inhibitors that effectively disrupt glycoprotein processing. The effects of these inhibitors on CD73 dimerization and adenosine production were evaluated, and the related data and description have been included

in the revised manuscript (page 16, lines 366-370): “To further determine if branched N-glycans produced from MGAT1 could directly affect CD73 dimerization and adenosine production, we treated both cell lines with kifunensin and swainsonine to ensure efficient inhibition of N-glycan branching and the results demonstrated the critical role of branched N-glycans in stabilizing CD73 dimerization and adenosine production (Supplementary Fig.6i, j).”

Supplementary Fig. 6i

Supplementary Fig. 6i: i MDA-MB231 and MDA-MB468 cells were treated with 20 μ M kifunensin or swainsonine for 48 hours and the CD73 dimerization were determined by immunoblotting with semi-native gel.

Supplementary Fig. 6j

Supplementary Fig. 6j: j MDA-MB231 and MDA-MB468 cells were treated with 20 μ M kifunensin or swainsonine for 48 hours and the adenosine concentration in the medium was measured with Adenosine Assay from Cell Biolabs.

Question #2 - *Does overexpression of MGAT1 with a point mutation that reduces activity but not medial Golgi localization have reduced effects on CD73 dimerization and adenosine production?*

See PMID: 11467936.

We thank the reviewer for highlighting this interesting research. The reference provided (PMID: 11467936) reports two critical point mutations in hamster MGAT1: the G634A mutation (D212N) and the C907T mutation (R303W), which significantly reduce enzymatic activity without affecting medial Golgi localization. To investigate whether similar mutations in human MGAT1 would impact CD73 dimerization and adenosine production, we conducted additional experiments to explore this mechanism. To this end, we aligned the MGAT1 sequence from hamster and human sources and identified the corresponding point mutations in human MGAT1: D210N and R301W. These residues are highly conserved in GlcNAc-TI across multiple species, including humans, rats, rabbits, mice, and golden hamsters. We generated D210N and R301W point mutations in human MGAT1 by cloning the mutated sequences into the pLENTI plasmid and validated the constructs through sequencing. We then established stable TNBC cell lines (MDA-MB231 and MDA-MB468) overexpressing either wild-type MGAT1 or MGAT1 with the D210N and R301W mutations, which significantly reduce enzymatic activity without affecting medial Golgi localization. The effects of these mutations on CD73 dimerization and adenosine production were evaluated, and the related data, description and reference (PMID: 11467936 cited as reference No.52) have

been included in the revised manuscript (page 16, lines 361-366): “The dimerization was further confirmed in several TNBC cell lines by running semi-native gels (Supplementary Fig. 6e) and the knockdown of MGAT1 dramatically decreased the CD73 dimerization (Fig. 5f and Supplementary Fig. 6f). The point mutations, D210N and R301W, on MGAT1 previously reported to reduce its enzymatic activity⁵² also reduce the glycosylation of CD73, leading to decreased dimerization and adenosine production (Supplementary Fig. 6g, h), which provide direct evidence that MGAT1 enzymatic activity is critical for stabilizing CD73 dimerization.”

Supplementary Fig. 6g

Supplementary Fig. 6g: g The CD73 dimerization in MDA-MB468 and MDA-MB231 cells with MGAT1 OE or R301W-MGAT1 OE or D210N-MGAT1 OE were determined by immunoblotting.

Supplementary Fig. 6h

Supplementary Fig. 6h: h The adenosine concentration in MDA-MB468 and MDA-MB231 cells with MGAT1 OE or R301W-MGAT1 or D210N-MGAT1 was measured with Adenosine Assay from Cell Biolabs.

Question #3 - Do monomeric forms of CD73 produced from control versus *Mgat1* null cells form dimers in vitro? That is, does CD73 with branched N-glycans form dimers whereas CD73 with *Man5GlcNAc2* at the 4Ns does not?

Thank reviewer for the question. Based on our data, through the treatment of kifunensin and swainsonine, we confirmed that branched N-glycans are critical for CD73 dimerization and adenosine production. This is further supported by the results of Endo H/PNGase F treatment (Fig. 5b) and the semi-native gel results obtained from MGAT1-WT/OE/KD-stable lines (Fig. 5f & Supplementary Fig. 6f). In WT and MGAT1-OE stable lines, Endo H treatment caused only minor shifts in the bands (Fig. 5b lane 1 vs lane2; lane 3 vs lane 4), indicating that most of the fully glycosylated CD73 was covered predominantly with complex N-glycans.

These glycosylated forms of CD73 demonstrated efficient dimerization. However, in MGAT1-KD stable lines, Endo H treatment significantly depleted the insufficiently glycosylated CD73, resulting in a dramatic shift in the bands (Fig. 5b lane 6 vs lane 7). These insufficiently glycosylated CD73 forms failed to efficiently form dimers (Fig. 5f and Supplementary Fig. 6f). Therefore, we conclude that branched N-glycans are critical for CD73 dimerization, whereas insufficient glycosylation undermines its dimerization efficacy. These findings directly address the question and provide mechanistic insight into the role of glycosylation in CD73 function.

Fig. 5b

Fig. 5b: b The whole cell lysates of MDA-MB468 cells with MGAT1 OE/KD were treated with endoglycosidase H and PNGase F and the CD73 protein levels and molecular sizes were determined by immunoblotting.

Fig. 5f & Supplementary Fig. 6f

Fig. 5f: f CD73 dimerization in MDA-MB468 cells with MGAT1 WT/OE/KD was determined by immunoblotting with semi-native gel.

Supplementary Fig. 6f: f CD73 dimerization in 4T1 cells with MGAT1 WT/OE/KD was determined by immunoblotting with semi-native gel.

Question #4 - Do cells producing only monomeric CD73 form fewer tumors?

Thank you for the comments. Based on our results, we observed a clear correlation between CD73 dimerization and tumor growth. In **Fig. 5f and Supplementary Fig. 6f**, we demonstrated that MGAT1 knockdown in MDA-MB-468 and 4T1 cells significantly reduced CD73 dimerization. This reduction was accompanied by decreased T cell function, suggesting that the dimeric form of CD73 plays a crucial role in modulating the tumor microenvironment. In addition, MGAT1 knockdown in 4T1 and E0771 cells resulted in markedly slower tumor growth and significantly reduced tumor weight *in vivo* compared to control models (**Fig. 7i, j and Supplementary Fig. 10 d, e**). These results provide strong evidence that impairing CD73 glycosylation, and consequently its dimerization, attenuates tumor progression. Furthermore, in **Fig. 7m, n and Supplementary Fig. 10j, k**, we assessed the impact of co-overexpressing MGAT1 and a CD73 mutant (4NQ-CD73) incapable of glycosylation and dimerization in E0771 and 4T1 cells. Tumors derived from these cells exhibited slower growth rates compared to control cells with fully glycosylated and

dimerization-competent CD73. Together, these findings clearly demonstrate that cancer cells producing predominantly monomeric CD73, exhibit reduced tumor growth and aggressiveness.

Fig. 5f & Supplementary Fig. 6f

Fig. 5f: f CD73 dimerization in MDA-MB468 cells with MGAT1 WT/OE/KD was determined by immunoblotting with semi-native gel.
Supplementary Fig. 6f: f CD73 dimerization in 4T1 cells with MGAT1 WT/OE/KD was determined by immunoblotting with semi-native gel.

Fig. 7i, j

Fig. 7i, j: i, j 4T1 control and 4T1-MGAT1 KD tumors were harvested 21 days after tumor challenge and analyzed. Tumor growth was plotted (i), and tumor weight (j) was measured at the endpoint.

Question #5 - *The CD73-4NQ mutant is not a good test molecule as it does not mimic CD73 from an Mgat1 null cell, is trapped intracellularly and probably its misfolding causes ER stress. Are CD73 KO or KD mutant cells defective in tumor formation?*

Thank you for the concern of our experimental design using CD73-4NQ mutant. We agree that the CD73-4NQ mutant has some constraints, particularly in its inability to fully mimic CD73 derived from Mgat1-null cells. However, at this stage, the CD73-4NQ mutant remains one of the most effective tools to investigate the role of N-glycosylation in CD73 function, as it enables us to isolate the influence of glycosylation on CD73 dimerization and downstream effects. This approach has been extensively used to explore the functional implications of glycosylation in various contexts (Shu et al., Nat Commun. 2024, doi: 10.1038/s41467-024-51242-8; Huang et al., Nat Commun. 2021, doi: 10.1038/s41467-021-22618-x; Song et al., Cancer Discov. 2020, doi: 10.1038/s41467-021-22618-x.). In addition, our *in vivo* data observed that knockdown of MGAT1 in both 4T1 and E0771 cells led to dramatically slower tumor growth and reduced tumor weight compared to control models. This provides direct evidence of the critical role MGAT1-CD73 axis plays in tumor formation.

Regarding your question about CD73 KO or KD cells, there is substantial evidence supporting the role of CD73 in tumor formation. Several studies (Sun et al., J Cancer Res Clin Oncol, 2024, doi: 10.1007/s00432-024-05869-1; Tang et al., Nat Commun. 2023, doi: 10.1038/s41467-023-38578-3.; Jin et al., Cancer Res. 2010, doi: 10.1158/0008-5472.CAN-09-3109) have demonstrated that either knockdown (KD) or complete knockout (KO) of CD73 results in significant impairment of tumor progression in mouse models (Fig. 7i, j and Supplementary Fig. 10d, e). Moreover, targeting CD73 with antibodies or small molecule inhibitors has consistently shown dramatic reductions in tumor formation, reinforcing its critical role in tumorigenesis.

Question #6 - The data implicating VAMP3/THSB1 and 2 and adenosine should be in the main text, not just in the supplement. These aspects of the hypothesis are referenced in the abstract and highlights and should be supported by robust evidence in the main text.

We thank reviewer's suggestion. In response to your comment, we have reorganized Fig. 5 to integrate key results related to VAMP3, THBS1 and adenosine directly into the main Fig. 5h-m. These changes ensure that the data are clearly aligned with the abstract and highlights and provide robust evidence supporting our hypothesis.

Updated Fig. 5h-m

Updated Fig. 5h-m: **h** The scheme of investigating CD73 dimerization with a split-GFP assay. The GFP 1-10 or GFP 11x7 was attached to the N-terminus of WT CD73, CD73-4NQ, or CD73Δ480-537 and co-transfected into cells to investigate their dimerization abilities. **i** The split-GFP signal generated from dimerized WT CD73, CD73-4NQ, or CD73Δ480-537 was visualized by confocal microscopy in HEK293T cells. **j** The percentage of membrane CD73⁺ cells was measured with flow cytometry in WT MDA-MB231, MDA-MB231-VAMP3-KD, WT MDA-MB468, MDA-MB468-VAMP3-KD cells. ****p < 0.0001. Data (mean ± SEM) are representative of at least three independent experiments. **k** Schematic model showing how MGAT1-mediated glycosylation of CD73 orchestrates its dimerization and further translocation to cell membranes. Failure of CD73 glycosylation impedes CD73 dimerization and membrane translocation. **l** Spearman's rank correlation analysis shows the MGAT1 protein expression is highly positively correlated with the THBS1 signaling pathways. **m** The adenosine levels were determined in MDA-MB468 and MDA-MB231 after treatment of THBS1 at indicated concentrations. ****p < 0.0001. Data (mean ± SEM) are representative of at least three independent experiments.

Additionally, we have revised the text in the Results sections to emphasize the role of VAMP3 and THBS1 in CD73-mediated regulation of adenosine production and tumor progression.

Question #7 - *Additional concerns that should be addressed:*

a. The title is very broad and should be more focused on the empirical findings.

We appreciate your suggestion and have carefully reconsidered our title. The current title is thoughtfully crafted to engage researchers across diverse disciplines, including glycosylation biology, immunology, and cancer therapy. While more focused titles have their merits, we believe the present title achieves an optimal balance between specificity and broad appeal.

b. Is there evidence in immune-cold tumors that CD73 is a dimer and carries complex N-glycans?

Thank you for the comments. While the current evidence on CD73 dimerization in immune-cold tumors remains limited, our results provide key initial findings supporting the presence of dimerized CD73 with complex N-glycans in tumor models. To directly address reviewer's concerns, we further evaluated the CD73 status in several immune suppressive TNBC cell lines including MDA-MB231, MDA-MB468 and 4T1 cells. Specifically, the new semi-native gel analysis (Supplementary Fig. 6e) revealed a significant fraction of CD73 existing in the dimerized form in three TNBC cell lines. Further newly conducted Endo H/PNGase F treatment (Supplementary Fig. 6a) demonstrates that the complex N-glycans is the most abundant type glycan on the fully glycosylated CD73 in those TNBC cells, which plays a critical role in facilitating protein dimerization and stability. These results further strengthen our observations in Fig. 5b, f that a majority of fully glycosylated CD73 is modified with complex N-glycans and glycosylation, particularly the addition of branched N-glycans via MGAT1, facilitates CD73 dimerization.

The newly added data as well as related description has been added to the revised manuscript (page 15, line 336-341): "To investigate the predominantly type of glycans on CD73, we further

tested the effect of endoglycosidase H (specifically cleaving high mannose and hybrid N-glycan) and PNGase F (an N-glycosidase removing all N-linked glycan) on CD73 in TNBC cell lines. The CD73 band shift indicates that complex type N-glycan is the major type of asparagine-linked carbohydrate attached to CD73 (Supplementary Fig. 6a), but without sufficient MGAT1, a fraction of less glycosylated CD73 will be covered with high mannose and hybrid N-glycan (Fig. 5b).” and (page 16, line 359-363): “In response to MGAT1-mediated CD73 modification, we observed a dramatic change in CD73 dimerization. We initially confirmed the dimerization of CD73 when the CD73 complex was stabilized in the presence of a crosslinking agent (Fig. 5e). The dimerization was further confirmed in several TNBC cell lines by running semi-native gels (Supplementary Fig.6e) and the knockdown of MGAT1 dramatically decreased the CD73 dimerization (Fig. 5f and Supplementary Fig. 6f).”

Supplementary Fig. 6a

Supplementary Fig. 6a: a The whole cell lysates of MDA-MB231, MDA-MB468 and 4T1 cells were treated with endoglycosidase H and PNGase F and the CD73 protein levels and molecular sizes were determined by immunoblotting.

Supplementary Fig. 6e

Supplementary Fig. 6e: e CD73 dimerization in MDA-MB231, MDA-MB468, and 4T1 cells with MGAT1 OE/KD was determined by immunoblotting with semi-native gel.

Fig. 5b

Fig. 5b: b The whole cell lysates of MDA-MB468 cells with MGAT1 OE/KD were treated with endoglycosidase H and PNGase F and the CD73 protein levels and molecular sizes were determined by immunoblotting.

Fig. 5f & Supplementary Fig. 6f

Fig. 5f: f CD73 dimerization in MDA-MB468 cells with MGAT1 OE/KD was determined by immunoblotting with semi-native gel.
Supplementary Fig. 6f: f CD73 dimerization in 4T1 cells with MGAT1 OE/KD was determined by immunoblotting with semi-native gel.

c. “MGAT1 on Golgi bodies.” Is an inappropriate term. Change to “MGAT1 in the Golgi compartment” or something similar throughout the text.

Thank you for the comments. we have revised this phrasing to “MGAT1 in the Golgi compartment” or similar context-appropriate terms. All changes have been highlighted blue in the revised text.

d. The reaction catalyzed by MGAT1 which initiates of branched N-glycan synthesis leading to complex/branched N-glycans should be shown and the hypothesis that branched N-glycans are necessary for CD73 dimerization should be clearly stated and tested.

We appreciate the reviewer’s suggestion to further investigate the role of branched N-glycans produced by MGAT1 in CD73 dimerization and adenosine production. To address this, we performed additional experiments using TNBC cell lines (MDA-MB231 and MDA-MB468) treated with kifunensin and swainsonine, two well-established glycosidase inhibitors that effectively

disrupt glycoprotein processing. The effects of these inhibitors on CD73 dimerization and adenosine production were evaluated (Supplementary Fig. 6i, j), and the related data and description have been included in the revised manuscript (page 16, lines 366-370): “To further determine if branched N-glycans produced from MGAT1 could directly affect CD73 dimerization and adenosine production, we treated both cell lines with kifunensin and swainsonine to ensure efficient inhibition of N-glycan branching and the results demonstrated the critical role of branched N-glycans in stabilizing CD73 dimerization and adenosine production (Supplementary Fig.6i, j).”

Supplementary Fig. 6i

Supplementary Fig. 6i: i MDA-MB231 and MDA-MB468 cells were treated with 20 μ M kifunensin or swainsonine for 48 hours and the CD73 dimerization were determined by immunoblotting with semi-native gel.

Supplementary Fig. 6j

Supplementary Fig. 6j: j MDA-MB231 and MDA-MB468 cells were treated with 20 μ M kifunensin or swainsonine for 48 hours and the adenosine concentration in the medium was measured with Adenosine Assay from Cell Biolabs.

e. *“Glycosylation of CD73 in the cytosol determines its membrane abundance by regulating CD73 translocation.” This statement is grossly inaccurate. The catalytic domain of MGAT1 operates in the luminal, juxta-membrane region of the medial Golgi, not in the cytosol.*

To address this, we have revised the statement for accuracy, clarifying that the glycosylation of CD73 occurs in the Golgi lumen and plays a critical role in regulating its translocation to the plasma membrane. These changes have been highlighted in the revised manuscript (page 27 line 654-657): “This work shows that, before its integration into the membrane, MGAT1-dependent glycosylation of CD73 is a critical step in the medial Golgi, ensuring CD73 membrane-bound enzymatic function in producing adenosine for down-regulating T cell activities.”

f. *“complex N-linked glycan structures.” Should be simply “complex N-glycans” throughout the text.*

We agree with reviewer’s suggestion and have revised the text to use the term “complex N-glycans” consistently throughout the manuscript for clarity and accuracy. These changes have been implemented and highlighted blue in the revised text.

g. *“The addition of N-acetylglucosamine (GlcNAc) to growing glycan chains catalyzed by MGAT1 ensures the completion of glycan chain assembly on target proteins,” This is inaccurate. MGAT1 transfers one GlcNAc to a specific Man in the substrate Man5GlcNAc2Asn which allows the removal of two Man residues and the subsequent action of several branching GlcNAc-transferases (MGAT2, MGAT3, NGAT4, MGAT5). MGAT1 does not add GlcNAc to growing glycan chains.*

Thank you for pointing out this imprecise. MGAT1 does not complete glycan chain assembly but instead transfers a single N-acetylglucosamine (GlcNAc) residue to a specific mannose on the substrate glycan, initiating the synthesis of branched N-glycans. We have revised this

statement throughout the manuscript to accurately reflect MGAT1's specific role in catalyzing the transfer of GlcNAc to the α -1,3-linked mannose of the substrate. These corrections have been implemented and highlighted blue in the revised text (page 4, lines 79-83): "MGAT1, mannosyl (alpha-1,3-)-glycoprotein beta-1,2-N-acetylglucosaminyltransferase, is a glycosyltransferase essential in the synthesis and maturation of complex N-glycans. The addition of N-acetylglucosamine (GlcNAc) to α -1,3-linked mannose of the substrate catalyzed by MGAT1 ensures the formation of complex glycans on target proteins, resulting in the orchestration of various cellular processes such as protein folding, protein translocation, and cell-cell communication¹⁰⁻¹⁵."

h. Fig 1B, C – define red and blue categories in the legend.

Thank you for your suggestion. We have revised the Figure legend for Fig. 1b, c to clearly define the red and blue categories. Specifically, we have labeled the red category as "immune-suppressive" and the blue category as "immune active."

Updated Fig. 1b, c

Updated Fig. 1b, c: **b, c** Distribution of single sample GSEA scores of BLBC separated by immune subgroup, using gene sets involved in immune functions (**b**) and glycosylation (**c**). P-values are from Wilcoxon tests. * $p < 0.05$, ** $p < 0.01$, *** $p < 0.001$, and **** $p < 0.0001$.

i. Fig. 1D – while MGAT1 is at the top, other GlcNAc-transferases are not far below. All GlcNAc-transferases contribute to increased branching and potentially affect tumor formation and progression similarly to MGAT1.

Thank you for your comment. You are correct that several other GlcNAc-transferases, including MGAT2 and MGAT4A, may play roles in the branching of N-glycans and exhibited correlation with immune checkpoints. However, our study focuses on MGAT1 because it is uniquely positioned as the most critical enzyme in the glycosylation pathway for the following reasons: (1) MGAT1 catalyzes the addition of the first N-acetylglucosamine (GlcNAc) to the core mannose structure, which is a necessary step for the synthesis of branched N-glycans. Without MGAT1 activity, downstream enzymes such as MGAT2, MGAT4A, and MGAT5 cannot further elaborate these branches. Therefore, MGAT1 acts as the gatekeeper of N-glycan branching and determines whether more complex glycans can form. (2) Among all GlcNAc-transferases, MGAT1 shows the strongest positive correlation with immune-related genes in our dataset. This includes genes critical for immune suppression, T-cell function, and adenosine signaling, which are central to the immune-suppressive tumor microenvironment. (3) MGAT1's upstream position in the glycosylation pathway, combined with its strong correlation with immune-related genes, makes it a particularly attractive therapeutic target. For these reasons, we consider MGAT1 to be a pivotal GlcNAc-transferase in the context of this study and have prioritized it for in-depth investigation.

j. Fig. 1D – Fig. 1F – MGAT1 signal seems most consistently high in Her2+ rather than TNBC tumors. Need to quantitate images and normalize to actin. Gel is not convincing.

Thank you for your comment and suggestion to improve the presentation and quantification of the data in **Fig. 1f**. In response, we have made the following improvements: (1) To further refine and differentiate MGAT1 expression levels, we included a new set of breast cancer tissue arrays with additional HER2+ and TNBC samples. This expanded dataset allowed us to more

comprehensively assess MGAT1 expression across breast cancer subtypes and provided greater statistical power to our analysis. (2) We collected fresh samples of breast cancer cell lines and repeated the immunoblotting to generate clearer bands for MGAT1 expression. These experiments included normalized quantification to actin for better comparison between samples (Supplementary Fig. 2b). (3) We revised Fig. 1f by incorporating markers and different colors to distinguish the different subtypes of breast cancer cell lines, making the findings easier to interpret and visually clear.

While some TNBC cells have relatively low MGAT1 expression. It is expected because cancer cell lines may not fully capture the complexity of primary tumors, including stromal or immune cell contributions. In addition, TNBCs can be further classified into distinct molecular subtypes, including basal-like 1 (BL1), basal-like 2 (BL2), mesenchymal (M), and luminal androgen receptor (LAR), each of which displays unique biological and molecular characteristics. These subtypes may contribute to the observed variability in MGAT1 expression levels within the TNBC group. Such heterogeneity is expected, as different subtypes may rely on diverse signaling pathways, influencing MGAT1 expression. Nevertheless, TNBC as a whole shows higher MGAT1 expression compared to luminal and Her2+ breast cancer types. This highlights the potential importance of MGAT1 as a biomarker and therapeutic target for TNBC patients with elevated MGAT1 levels.

The updated data and revised manuscript now provide a clearer and more comprehensive assessment of MGAT1 expression in HER2+ and TNBC subtypes. The revised content can be found in the manuscript on page 8, lines 155-158: “Further systematic protein expression analysis across an extensive array of breast cancer cell lines confirmed that MGAT1 protein was prevalent in HER2+ and TNBC cell lines, reinforcing the potential link to immune escape mechanisms in TNBC (Fig. 1f and Supplementary Fig. 2b).”

Updated Fig. 1f

Updated Fig. 1f: f Expression of MGAT1 in normal human mammary epithelial cells and various subtypes of breast cancer cells detected by immunoblotting using anti-MGAT1 antibody.

Supplementary Fig. 2b

Supplementary Fig. 2b: b The immunoblots of MGAT1 in normal human mammary epithelial cells and various subtypes of breast cancer cells were quantified with Image Lab and normalized to actin.

k. Ref 35 Flynn et al is not cited appropriately.

Thank you very much for reviewer's careful reading! We are aware of the inappropriate position of this reference and replaced it with more proper reference!

l. Fig. 2C. Are histograms meant to show "%IFN γ + CD8 T cells" or "%IFN γ in CD8 T cells" as labeled? Should "in" be removed from labels of all 4 histograms? Same comment for Fig. 2E and

2G and later Figures in which the % cells expressing a certain cytokine are what is being measured.

Thank you for your suggestion regarding the labeling of histograms in Fig. 2 and related Figures. We agree that the inclusion of “in” in the labels (e.g., “%IFN γ in CD8 $^+$ T cells”) could cause confusion, as the histograms are intended to show the percentage of CD8 $^+$ T cells expressing a given cytokine. In response, we have revised the labels to remove “in” from all relevant histograms in Fig. 2c, e, and g, and later Figures, including Fig. 6I, to improve clarity and consistency.

m. Fig. 2D. Taxes Red should be Texas Red.

Thank you for pointing out this error. We have corrected the label from "Taxes Red" to "Texas Red" in Fig. 2d. The updated figure has been included in the revised manuscript.

Updated Fig. 2d

Updated Fig. 2d: d The infiltration of PBMCs (red) in 3D spheroids constructed with MDA-MB468, MDA-MB468-MGAT1-OE, and MDA-MB468-MGAT1-KD was visualized with Z-stack imaging using a Lionheart microscope.

n. Fig. 3A. Any of the many, many proteins in the silver-stained gel could be a “partner” of MGAT1. Showing co-IP with very OE proteins is not a proof of specificity. What are the appropriate controls? This story is a tale of cherry-picking and following the positive results.

Thank you for the comment. Our study addressing MGAT-CD73 axis' role in immune evasion is based on a systematic approach involving multiple layers of analysis and validation, which we summarize as follows: (1) Through comprehensive TNBCs cohort and bioinformatic data analysis, we identified a strong correlation between MGAT1 overexpression and immune evasion in tumors. This provided a critical foundation for investigating MGAT1's potential role in modulating tumor immunity. (2) To understand how MGAT1 influences tumor immune evasion, we sought to explore its molecular interactions and functional pathways. Given MGAT1's role in glycosylation and its potential to affect immune checkpoint regulation, this investigation focused on identifying its interactome and downstream effects. (3) To identify MGAT1-associated proteins, we used TAP purification combined with mass spectrometry, a classical and robust method for mapping protein-protein interaction networks. The silver-stained gel served as a visualization tool to identify MGAT1-associated protein complexes. The prioritization of candidate interactors was based on mass spec identified MGAT1 interacted protein ranking of enrichment scores, reproducibility, and functional relevance to glycosylation or immune-related pathways. This approach allowed us to systematically and unbiasedly determine MGAT1's interactome under physiological conditions. (4) From the mass spectrometry data, we identified several proteins associated with MGAT1. Among these, CD73 emerged as one of the top candidates with a strong interaction score. CD73 is a well-known player in immune evasion, primarily through its role in adenosine production and suppression of anti-tumor immune responses. Importantly, this finding aligns with our earlier data analysis, further supporting the biological significance of the MGAT1-CD73 interaction in regulating immune suppression.

Regarding the specificity of TAP co-IP experiments with overexpressed (OE) proteins. While TAP co-IP involves overexpression, we ensured low-copy expression of MGAT1-FLAG using a lentiviral system, maintaining near-physiological levels to minimize artifacts. To ensure specificity, we included multiple controls, such as empty vector-transfected cells to exclude nonspecific binding, IgG isotype controls for key candidates like CD73, and quantitative mass spectrometry

analysis to identify reproducible and enriched interactors across independent experiments. TAP purification further enhances specificity through the use of pre-conjugated anti-FLAG M2 beads and peptide-based elution, which reduces nonspecific binding and ensures high-quality purification of immune complexes. From this rigorous analysis, CD73 emerged as one of the most specific interactors, consistent with its known role in immune evasion. While we recognize the limitations of overexpression systems, these controls and methodologies strongly support the specificity of our findings, ensuring an unbiased selection process and minimizing the potential for cherry-picking.

o. Fig. 3F. CD73 seems to be expressed throughout the cell. It is not confined to the Golgi and not only found in association with MGAT1.

Thank you for your observation regarding the localization of CD73 in Fig. 3f. We agree that CD73 is not exclusively localized to the Golgi but is distributed throughout the cell, including the plasma membrane, where it performs its primary enzymatic function. However, our focus in this study is on the MGAT1 interactome, specifically its role in the glycosylation of CD73 within the Golgi. MGAT1 is a Golgi-localized enzyme that catalyzes the first step in N-glycan branching, a key modification necessary for proper CD73 glycosylation. Without this step, CD73 cannot achieve the appropriate glycosylation needed for its stability, trafficking, and eventual function in other cellular compartments, including the plasma membrane. Thus, while CD73 is found in multiple locations, its functional maturation and readiness to carry out its immune-evasive role depend on proper glycosylation mediated by MGAT1 in the Golgi.

To further investigate the colocalization of MGAT1 and CD73 in the Golgi, we performed subcellular fractionation of the Golgi and membrane compartments, followed by immunoblotting analysis. The new data and related description have been added to the manuscript on page 11, line 234-240: "Additionally, super-resolution microscopy further validated the colocalization of MGAT1 and CD73, providing superior spatial resolution and reinforcing the findings observed in

confocal imaging (Fig. 3f). We also conducted a proximity ligation assay to corroborate the colocalization of MGAT1 and CD73 in the Golgi compartment (Fig. 3g). To complement the imaging data, we conducted subcellular fractionation to separate the Golgi and membrane compartments⁴²⁻⁴⁴, followed by immunoblotting. The results revealed that both MGAT1 and CD73 were enriched in the Golgi fraction, supporting their co-distribution within this compartment (Supplementary Fig. 4d). "

And the method is included in the methods section as "Plasma membrane and Golgi fractionation".

Supplementary Fig. 4d

Supplementary Fig. 4d: d The colocalization of MGAT1 and CD73 in the Golgi was validated by fractionating the Golgi and membrane sections of the cells, followed by immunoblot.

p. Fig. 3l. What are the "big datasets" from which these data were obtained?

Thank you for your question regarding the source of the "big datasets" in Supplementary Fig. 4f (previous Fig. 3i). These data were derived from the publicly available CPTAC dataset (PDC Study Identifier: PDC000408), as described in the study by Anurag M., Jaehnig EJ., Krug K., Lei JT., et al. (2022) (DOI: 10.1158/2159-8290.cd-22-0200). We have included this information in the manuscript to clarify the source of the data (page 11 line 251-253): "Consistent with the above

results, the big datasets with Spearman's rank correlation analysis using proteomic datasets from the PDC000408 cohort⁴⁵ clearly confirmed the correlation between MGAT1 protein levels with several immune regulators, especially CD73 (NT5E) (Supplementary Fig. 4f).”

q. Fig 4D. Why is there a double band for IP anti-V5(MGAT1) and IB anti-V5?

Thank you for the comments regarding the double band for IP anti-V5 (MGAT1) and IB anti-V5 in Fig. 4d. Upon reviewing the experimental conditions, we found that the use of different gel types contributed to this observation. The right panel was obtained using a plain polyacrylamide gel with a uniform acrylamide percentage, which is optimized for resolving proteins within a narrow molecular weight range but can produce broader bands for proteins with subtle differences in molecular weight. In contrast, the left panel used a gradient gel with varying acrylamide concentrations, which offers higher resolution across a broader molecular weight range and can reveal finer distinctions. These differences in gel properties explain the variation in banding patterns between the panels.

r. Fig. 5A. KD of MGAT1 only partially converts CD73 to lower MW in both cell lines. Should do MGAT1 KO to validate.

Thank you for your suggestion to validate the role of MGAT1 in CD73 glycosylation using a more effective knockdown model. In response, we generated MGAT1-knockout MDA-MB231 stable lines using the CRISPR/Cas9 system with newly design sgRNAs and evaluated CD73 glycosylation by Western blot. This new model demonstrated significantly improved knockdown efficiency compared to the original sgRNA constructs. The results showed a substantial reduction in fully glycosylated CD73, resulting in a dramatic decrease in overall CD73 glycosylation, similar to the pattern observed in MDA-MB468. The new data and related descriptions have been incorporated into the revised manuscript on page 14 line 332 to page 15 line 336: "As described above, we initially engineered MGAT1 OE and KD in MDA-MB468 and MDA-MB231 cells. As

shown in Fig. 5a, while a minor change in CD73 abundance in response to MGAT1 OE was observed, reduced glycosylation of CD73 (running smaller molecular mass ~60 kDa) was clearly detected when MGAT1 was knocked down, confirming the aforementioned CD73 glycosylation catalyzed by MGAT1."

Updated Fig. 5a

Updated Fig. 5a: a The CD73 protein levels and molecular sizes in MDA-MB468 and MDA-MB231 cells with MGAT1 OE and MGAT1 KD were determined by immunoblotting.

s. Fig. 5H-J. Removing 4 N-glycans in the 4NQ mutant traps CD73 in the cell – probably in the ER due to misfolding. This is quite different from changing the N-glycans from complex to high mannose in the medial Golgi. Cannot interpret the data in Fig 5J in relation to dimerization. In addition, the quality of the images is too poor to publish. Fig. label is MGAT1 KO but should be MGAT1 KD.

Thank you for the concern of our experimental design using CD73-4NQ mutant. We agree that the CD73-4NQ mutant has some constraints, particularly in its inability to fully mimic CD73 derived from Mgat1-null cells. However, at this stage, the CD73-4NQ mutant remains one of the most effective tools to investigate the role of N-glycosylation in CD73 function, as it enables us to isolate the influence of glycosylation on CD73 dimerization and downstream effects. This approach has been extensively used to explore the functional implications of glycosylation in various contexts

(Shu et al., Nat Commun. 2024, doi: 10.1038/s41467-024-51242-8; Huang et al., Nat Commun. 2021, doi: 10.1038/s41467-021-22618-x; Song et al., Cancer Discov. 2020, doi: 10.1038/s41467-021-22618-x.). In addition, our *in vivo* data observed that knockdown of MGAT1 in both 4T1 and E0771 cells (Fig. 7i, j and Supplementary Fig. 10d, e) led to dramatically slower tumor growth and reduced tumor weight compared to control models. This provides direct evidence of the critical role MGAT1-CD73 axis plays in tumor formation.

To address the concerns regarding the interpretation of Fig. 5j, we used the Split-GFP system, which enables real-time visualization of protein-protein interactions in living cells. This system utilizes two non-fluorescent GFP fragments: GFP 1-10 (the larger fragment) and GFP11 (a short peptide). When these fragments are brought into close proximity through dimerization of tagged proteins, they reassemble into a functional GFP, emitting green fluorescence. We engineered plasmids encoding WT CD73, CD73-4NQ, and a dimer-deficient CD73 mutant, each tagged with GFP 1-10 or GFP11x7 (an optimized GFP11). These plasmids were co-transfected into HEK293T cells, and fluorescence was visualized via confocal microscopy. Strong green fluorescence was observed for WT CD73, confirming dimerization under normal glycosylation conditions, whereas fluorescence was significantly diminished for CD73-4NQ and the dimer-deficient mutant, indicating impaired dimerization. These findings were further validated in MDA-MB231 and MDA-MB468 cells stably expressing the Split-GFP constructs, as shown in Fig. 5i and Supplementary Fig. 7b-e. The Split-GFP system provides high specificity and live-cell imaging capability, offering direct evidence that glycosylation is essential for CD73 dimerization.

To address the concern regarding the image quality, we optimized the imaging protocol to improve image quality, as shown in updated Fig. 5i. First, we removed the bright field from the overlapping image to reduce background noise and enhance the contrast of green fluorescence intensity. Second, we switched from a 60x oil objective to a 40x water objective, which is better suited for live-cell imaging in the water-based medium used during the experiment. This adjustment improved compatibility with the live-cell environment and resulted in clearer images.

All images were captured using the Leica Stellaris 8 confocal microscope, which provides advanced functionality for high-quality live-cell imaging.

Finally, we have corrected the Figure label from "MGAT1 KO" to "MGAT1 KD" to accurately reflect the experimental setup. We have also revised our manuscript to reflect these changes from page 16 line 380-386 to reflect these changes "To visualize the CD73 dimer in cells, we conducted split-GFP assays to validate the influence of glycosylation on CD73 dimerization. Plasmids of CD73 with GFP 1-10 or GFP11x7 extensions were engineered and co-transfected into HEK-293T cells (Fig. 5h), and the green fluorescence was visualized with a confocal microscope. As shown in Fig. 5i, we captured the strongest green fluorescence for WT CD73, while the signal from CD73-4NQ and dimer-deficient mutant CD73 was largely diminished. This observation was further confirmed by stably expressing CD73 with GFP 1-10 or GFP11x7 in MDA-MB231 and MD-MB468 cells (Supplementary Fig. 7b, c)."

Updated Fig. 5i

Updated Fig. 5i: i The split-GFP signal generated from dimerized WT CD73, CD73-4NQ, or CD73 Δ 480-537 was visualized by live cell confocal microscopy in HEK-293T cells.

t. Fig. 6F “Several compounds, including No.2, No.8, No.9, and No.14, showed potent inhibition of MGAT1 catalytic function (Fig. 6F).” Inhibition is only a 60% reduction in activity at most. Heterozygote *Mgat1* mice have 50% enzyme and are viable whereas *Mgat1* null mice are embryonic lethal. Only 50-70% inhibition of enzyme activity is obtained for the 3 best compounds at a concentration of 10 μ M which is very high (Fig. 6K). The “best” inhibitor W-GTF01 needs to be characterized in activity and dimerization assays like Nos.2, 8, and 9. There are no data that W-GTF01 is “best”. Also, no evidence that it is specific for MGAT1. If the inhibitors compete with UDP-GlcNAc they would be expected to inhibit any GlcNAc-transferase. The authors should discuss this point.

Thank you for your comment regarding the efficacy and specificity of the MGAT1 inhibitors. We acknowledge the limitations in the inhibiting efficacy of the leading compounds (No.2, No.8 (W-GTF01), and No.9), as the maximal inhibition achieved was only 50–70% at a concentration of 10 μ M. W-GTF01 is currently the leading compound discovered by virtual screening, and we are working on chemical modifications to enhance its potency and specificity. Regarding the specificity of No.8/W-GTF01, W-GTF01 binds to the UDP-binding domain of MGAT1, which is a conserved site shared by other GlcNAc-transferases. We recognize that this raises the possibility of off-target effects on other GlcNAc-transferases, potentially leading to unwanted toxicity. To address this, we have added a discussion in the revised manuscript about the limitations of UDP-binding domain-targeting inhibitors and the need for strategies to improve specificity (page 29 line 710-page 30 line 725): “As a key enzyme in the N-glycosylation pathway, MGAT1 is critical for the proper folding, stability, and function of numerous glycoproteins, including membrane receptors and transporters. Inhibiting MGAT1 activity *in vivo* with compounds like W-GTF01 has demonstrated promising efficacy, particularly in immune-cold TNBC models, by promoting tumor regression through the revival of cytotoxic CD8⁺ T cell responses. However, MGAT1’s integral role in glycoprotein biosynthesis also raises concerns about systemic toxicity and off-target effects, as W-GTF01 binds to the UDP-binding domain shared by other GlcNAc-transferases. This could

result in suboptimal glycosylation of essential proteins, impairing cellular signaling, and receptor-mediated interactions, ultimately leading to adverse physiological outcomes. To mitigate these risks, strategies such as antibody-drug conjugates (ADCs) and combination therapies that enhance selectivity and minimize off-target effects are being actively explored. Furthermore, comprehensive characterization of W-GTF01, including its toxicity, selectivity, and pharmacokinetics, is critical for therapeutic development. Future efforts should also focus on identifying biomarkers to monitor potential side effects and guide therapeutic interventions. These approaches will enable the development of more effective and safer MGAT1-targeting therapies, leveraging their therapeutic potential while minimizing unwanted biological consequences.”

To address the comment regarding the characterization of W-GTF01 (No.8). We agree that comprehensive testing of inhibitors is critical for validating their effects. In our study, we assessed the activities of compounds Nos. 2, 8 (W-GTF01), and 9 to determine their influence on MGAT1 enzymatic function (Fig. 6k) and membrane CD73 levels in both MDA-MB231 and MDA-MB468 cells (Fig. 6i). Among the tested compounds, W-GTF01 (No.8) consistently demonstrated the most significant reduction in membrane CD73 levels, indicating its strong effect on CD73 regulation. Furthermore, in the co-culture experiment illustrated in Fig. 6l, m, W-GTF01 (No.8) exhibited the highest and most consistent effect in boosting T cell function and proliferation compared to other compounds (No. 2 and No. 9) (Fig. 6l), leading to the most efficient tumor inhibition (Fig. 6m). Based on these results, W-GTF01 (No.8) was prioritized for further investigation as the most promising candidate.

Fig. 6k

Fig. 6k: k The normalized dose-dependent inhibition of MGAT1 by leading compounds (No.2, No.8, and No.9) was measured with our *in vitro* function-based drug screening system.

Fig. 6l

Fig. 6l: l MDA-MB468 cells were cocultured with PBMC cells and the indicated compounds. Percentages of TNF α +, IFN γ +, Granzyme B+ and Ki-67+ in CD8+ T cells were measured and quantified using flow cytometry. ***p<0.001, ****p<0.0001. Data (means \pm SEM) are representative of at least three independent experiments.

Fig. 6m

Fig. 6m: m The survival of MDA-MB468 cells in coculture with pre-activated PBMCs and No.2/No.8(W-GFT01)/No.9 was monitored by time lapse image-based quantification. The data are presented as mean \pm SEM from three replicates of a representative experiment.

u. Fig. 7. Were No. 2, 8, or 9 inhibitors tested in the tumor formation assay? If so, the data should be included.

Thank you for the comment. We assessed the activities of compounds No.2, No.8 (W-GTF01), and No.9 to determine their impact on membrane CD73 levels in both MDA-MB231 and MDA-MB468 cells (Fig. 6i). Among the tested compounds, W-GTF01 (No.8) consistently demonstrated the most significant reduction in membrane CD73 levels, indicating its strong regulatory effect on CD73. Furthermore, in the co-culture experiment (Fig. 6l, m), W-GTF01 exhibited the most consistent and robust effect in enhancing T cell function and proliferation compared to compounds No.2 and No.9. After multiple rounds of evaluation, W-GTF01 emerged as the most promising candidate among the 14 tested compounds, including No.2 and No.9.

To evaluate its efficacy *in vivo*, W-GTF01 was tested in a preclinical tumor formation assay (Fig. 7p, q and Supplementary Fig. 10l). The results demonstrated that W-GTF01 significantly enhanced the therapeutic benefits of anti-PD-L1 antibody treatment, leading to a significant reduction in tumor growth while maintaining tolerable toxicity levels. We are also actively working to optimize W-GTF01 by modifying its structure and assessing additional analogs to improve efficacy and specificity. These results will be included in future studies.

Fig. 7p, q

Fig. 7p, q: p, q 4T1-hPD-L1 cells, where the endogenous mouse PD-L1 was replaced with its human counterpart, were orthotopically injected into the left fourth mammary fat pad and allowed to grow to ~100 mm³, followed by injection of W-GTF01 (10 mg/kg, i.p.) two times/week and PD-L1 antibody durvalumab (10 mg/kg, i.p.) 3 times/week. PBS and IgG were used in the control groups. The tumor growth (p) and the survival (q) of the mice were plotted. *p<0.05, **p<0.01, ***p<0.001, and ****p<0.0001. For tumor growth statistical analysis, two-way ANOVAs followed by Tukey's multiple comparison tests were performed. Data (means ± SEM) are representative of at least two independent experiments with 5-10 independently analyzed mice per group.

Supplementary Fig. 10I

Supplementary Fig. 10I: I 4T1-hPD-L1 and 4T1-hPDL1-MGAT1 cells, where the endogenous mouse PD-L1 was replaced with its human counterpart, were orthotopically injected into the left fourth mammary fat pad and allowed to grow to ~100 mm³, followed by injection of W-GTF01 (10 mg/kg, i.p.) two times/week and PD-L1 antibody durvalumab (10 mg/kg, i.p.) 3 times/week. PBS and IgG were used in the control groups. The body weight of the mice were plotted. *p<0.05, **p<0.01, ***p<0.001, and ****p<0.0001. For tumor growth statistical analysis, two-way ANOVAs followed by Tukey's multiple comparison tests were performed. Data (means ± SEM) are representative of at least two independent experiments with 5-10 independently analyzed mice per group.

v. Fig. 8. It is not clear why the authors highlight neutrophils as TILs responsive to low MGAT1. Data for neutrophils are not significantly different in DH vs DL tumors or MGAT1 OE vs KD tumors.

We thank the reviewer for pointing this out and apologize for the incorrect labeling of neutrophils as TILs. We have corrected this error by removing the term “TILs” from Fig. 8b, c and their corresponding legends. We agree that the proportions of neutrophils do not show significant differences between MGAT1^{high}CD73^{high} (DH) vs MGAT1^{low}CD73^{low} (DL) tumors or MGAT1 OE vs. KD tumors. However, our data revealed that other immune cell populations, including CD8⁺ memory T cells and memory B cells, were significantly enriched in MGAT1^{low} areas compared to MGAT1^{high} areas (Fig. 8b) and in DL relative to DH areas within the non-epithelial stromal compartment (Fig. 8c). These findings suggest that low MGAT1 expression is associated with the presence of specific immune cell subsets that may contribute to improved immune surveillance and anti-tumor responses.

Updated Fig. 8b, c

Updated Fig. 8b, c: b, c CIBERSORTx analysis of the relative abundance of individual cell populations between MGAT1lo and MGAT1hi areas (b) or between MGAT1loCD73lo (DL) and MGAT1hiCD73hi areas (DH) (c) among nontumor compartments using transcriptomics data.

w. Discussion. “enabling CD73 on the Golgi apparatus” sounds like CD73 is facing the cytosol. Should be “enabling CD73 in the Golgi apparatus” as most of CD73 faces the lumen of the Golgi compartment. Applies throughout the manuscript.

Thank you for your suggestion. We have revised this throughout the manuscript to “enabling CD73 in the Golgi apparatus” for improved accuracy and clarity. All relevant instances have been updated, and the changes have been highlighted blue in the revised text.

w. Supplement. Fig. S3B has MGAT1-OB instead of OE. Check that all supplementary Fig. panels are referenced in the main text. Fig. S4E “Spearman’s rank correlation analysis shows the MGAT1 protein expression is highly positively correlated with the THBS1 and THBS2 signaling pathways.” Data from which the correlation analysis was performed should be described. Also there is no evidence for the source of THBS1/2 in the tumor environment.

Thank you for your suggestions. We have corrected the labeling error in Supplementary Fig. 6d (previous Figure S3B) from “MGAT1-OB” to the correct term “MGAT1-OE.” Additionally, we reviewed all supplementary Figure panels to ensure they are properly labeled and referenced in the main text.

Regarding the Fig. 5I (previous Supplementary Figure S4E) Correlation Analysis. The Spearman’s rank correlation analysis shown in Fig. 5I was performed using proteomic datasets from the PDC000408 cohort (Anurag M., et al., Cancer Discov., 2022), which includes comprehensive proteomic profiles of TNBC tumors. We have included this information in the manuscript to clarify the source of the data (page 18 line 414-418): “To investigate the upstream signaling that potentially modulates MGAT1 in the context of tumor invasion, we conducted Spearman's rank correlation analysis using proteomic datasets from the PDC000408 cohort⁴⁵ and found a tight correlation between elevated expression of MGAT1 and up-regulation of THBS1 and THBS2 signaling pathways (Fig. 5I). ”

Regarding the specific cellular sources of THBS1 and THBS2 in the tumor microenvironment, while our current study demonstrates a strong positive correlation between MGAT1 expression and the THBS1/2 signaling pathways in TNBC, the precise cellular sources of THBS1 and THBS2 within the tumor microenvironment remain to be fully characterized. Previous studies have shown that THBS1 and THBS2, members of the thrombospondin family, can be produced by a variety of cell types, including tumor cells, cancer-associated fibroblasts (CAFs), and myeloid-derived cells such as macrophages and dendritic cells. THBS1/2 play critical roles in modulating the extracellular matrix (ECM), facilitating tumor progression, angiogenesis, and immune suppression through the recruitment of immunosuppressive cells and inhibition of T cell activation. To address reviewer’s concern, we have revised our manuscript to clarify this point from page 31 line 757764: “In the context of TNBC, THBS1 could be secreted by tumor cells, cancer-associated fibroblasts (CAFs), and immune cells such as macrophages and dendritic cells, particularly in

response to hypoxia or stress signals within the tumor microenvironment. Elevated THBS1 expression is often associated with immune suppression and poor clinical outcomes. Further studies are needed to determine the precise sources and functional contributions of THBS1 within MGAT1^{high} and MGAT1^{low} tumor regions. Mechanistically, how MGAT1 is elevated by THBS1 and how THBS1 is coordinated with other CD73 regulatory pathways, for instance, TGF- β and hypoxia, requires further study.”

Reviewer #2, General Comments:

In this study, the authors investigate how the glycosyltransferase MGAT1 participates in regulating immunosuppression by influencing the glycosylation of CD73. They demonstrate that in immune-cold triple-negative breast cancer (TNBC), MGAT1 expression is closely associated with tumor immune evasion. Mechanistically, MGAT1-mediated glycosylation promotes CD73 dimerization, enhances its binding to VAMP3, and increases CD73 expression on the tumor cell membrane, which in turn suppresses the activation of anti-tumor CD8+ T cells. Furthermore, the authors identified a small molecule inhibitor, W-GTF01, through high-throughput drug screening, which inhibits CD73 glycosylation and dimerization. The combination of W-GTF01 with PD-L1 antibody therapy effectively suppresses the growth of immune-cold breast cancers. Thus, authors conclude that MGTA1 catalyzes glycosylation which is critical for CD73 membrane translocation and immunosuppression of TNBC cells.

This is an interesting paper that tries to address an important question in the field. However, some of the conclusions are not well supported by the data and there are multiple issues that authors need to address before publication.

We greatly appreciate the reviewer's constructive feedback, as well as their recognition of the significance of our study. We are pleased that the reviewer found our investigation into the role of MGAT1-mediated glycosylation in regulating immune suppression and tumor progression in immune-cold TNBC to be of interest. The acknowledgment of our work addressing an important question in the field is highly valued. To further strengthen the manuscript and address the reviewer's concerns, we have incorporated additional experimental data and analyses to address reviewer's concerns and to provide more robust support for our conclusions.

Reviewer #2, Specific Comments:

Question #1 - *In Figures 1E-F, immunohistochemistry suggests that MGAT1 expression is higher in TNBC than in other breast cancer subtypes. However, in breast cancer cell lines, MGAT1 expression is not high and is significantly lower in TNBC cell lines compared to HER2+ cell lines. This inconsistency needs to be addressed.*

Thank you for your comment and suggestion to improve the presentation and quantification of the data in Fig. 1d-f. In response, we have made the following improvements: (1) To further refine and differentiate MGAT1 expression levels, we included a new set of breast cancer tissue arrays with additional HER2+ and TNBC samples. This expanded dataset allowed us to more comprehensively assess MGAT1 expression across breast cancer subtypes and provided greater statistical power to our analysis. (2) We repeated the immunoblotting for breast cancer cell lines to generate clearer bands for MGAT1 expression. These experiments included normalized quantification to actin for better comparison between samples (Supplementary Fig. 2b). (3) We revised Fig. 1f by incorporating markers and different colors to distinguish the different subtypes of breast cancer cell lines, making the findings easier to interpret and visually clear.

While some TNBC cells have relatively low MGAT1 expression. It is expected because cancer cell lines may not fully capture the complexity of primary tumors, including stromal or immune cell contributions. In addition, TNBCs can be further classified into distinct molecular subtypes, including basal-like 1 (BL1), basal-like 2 (BL2), mesenchymal (M), and luminal androgen receptor (LAR), each of which displays unique biological and molecular characteristics. These subtypes may contribute to the observed variability in MGAT1 expression levels within the TNBC group. Such heterogeneity is expected, as different subtypes may rely on diverse signaling pathways, influencing MGAT1 expression. Nevertheless, TNBC as a whole shows higher MGAT1 expression compared to luminal and Her2+ breast cancer types. This highlights the potential

importance of MGAT1 as a biomarker and therapeutic target for TNBC patients with elevated MGAT1 levels.

The updated data and revised manuscript now provide a clearer and more comprehensive assessment of MGAT1 expression in HER2+ and TNBC subtypes. The revised content can be found in the manuscript on page 8, lines 155-158: “Further systematic protein expression analysis across an extensive array of breast cancer cell lines confirmed that MGAT1 protein was prevalent in HER2+ and TNBC cell lines, reinforcing the potential link to immune escape mechanisms in TNBC (Fig. 1f and Supplementary Fig. 2b).”

Updated Fig. 1f

Updated Fig. 1f: f Expression of MGAT1 in normal human mammary epithelial cells and various subtypes of breast cancer cells detected by immunoblotting using anti-MGAT1 antibody.

Supplementary Fig. 2b

Supplementary Fig. 2b: b The immunoblots of MGAT1 in normal human mammary epithelial cells and various subtypes of breast cancer cells were quantified with Image Lab and normalized to actin.

Question #2 – *The manuscript predominantly focuses on TNBC as an immune-desert environment and correlates MGAT1 expression with immunosuppression in TNBC. However, the co-culture experiments were conducted using HER2+ cell lines, with no experimental validation using TNBC cell lines. This limits the study's relevance to TNBC, and additional validation with TNBC models is necessary.*

We thank the reviewer for raising this important concern and for carefully evaluating our study. We agree that the proper choice of cell lines is critical for experimental design and interpretation. To clarify, the co-culture experiments were conducted using MDA-MB468 and MDA-MB231 cells, which are both well-established triple-negative breast cancer (TNBC) cell lines widely used in cancer research. We suspect that the lack of clear labeling in **Fig. 1f** may have caused some confusion regarding the subtype classification of the breast cancer cell lines. To address this, we have now added explicit subtype labels and different colors to distinguish between TNBC and HER2+ cell lines throughout the manuscript, particularly in **Fig. 1f**, to avoid any further ambiguity for readers.

To support the classification of MDA-MB-468 and MDA-MB-231 as TNBC, we refer to multiple peer-reviewed studies that extensively validate these cell lines as TNBC models: References for MDA-MB-468 include: Dai et al., Journal of Cancer. 2017. doi:10.7150/jca.18457; Lanning et al., Cancer Metab. 2017. doi: 10.1186/s40170-017-0168-x.; Borrego et al., Cancer Metab. 2016. doi: 10.1186/s40170-016-0148-6. References for MDA-MB-231 include: Dai et al., Journal of Cancer. 2017. doi:10.7150/jca.18457; Pommerenke et al., Cells. 2024. doi:10.3390/cells13040301. These references confirm that both cell lines lack ER, PR, and HER2 expression and are considered standard TNBC models.

Updated Fig. 1f

Updated Fig. 1f: f Expression of MGAT1 in normal human mammary epithelial cells and various subtypes of breast cancer cells detected by immunoblotting using anti-MGAT1 antibody.

Question #3 – *The paper lacks schematic diagrams illustrating the knockdown and overexpression results of MGAT1 in both mouse and human cell lines. These results should be included to support the conclusions drawn from the experiments.*

We thank the reviewer for the suggestion to include schematic diagrams illustrating the knockdown and overexpression results of MGAT1 to enhance clarity and support our conclusions. To address this, we have now incorporated the following:

The knockdown and overexpression results of MGAT1 in human TNBC cell lines (MDA-MB231 and MDA-MB468) are shown in **Fig. 5a**. The related description has been added to the revised manuscript from page 14 line 333 to page 15 line 336: “As shown in **Fig. 5a**, while a minor change in CD73 abundance in response to MGAT1 OE was observed, reduced glycosylation of CD73 (running smaller molecular mass ~60 kDa) was clearly detected when MGAT1 was knocked down, confirming the aforementioned CD73 glycosylation catalyzed by MGAT1”

The knockdown and overexpression results of MGAT1 in mouse TNBC cell lines (4T1 and E0771) are now included in **Supplementary Fig. 10a**, providing further validation in murine models. The related description has been added to the revised manuscript from page 20 line 491-493 line:

“To evaluate the role of MGAT1 in tumor growth *in vivo*, we created murine cell lines with modified MGAT1 expression. We developed MGAT1-OE, MGAT1-KD 4T1, and E0771 cells, with an empty vector as a control (Supplementary Fig. 10a).”

Updated Fig. 5a

Updated Fig. 5a: a The CD73 protein levels and molecular sizes in MDA-MB468 and MDA-MB231 cells with MGAT1 OE and MGAT1 KD were determined by immunoblotting.

Supplementary Fig. 10a

Supplementary Fig. 10a: a The MGAT1 protein levels in 4T1 and E0771 cells with MGAT1 OE and MGAT1 KD were determined by immunoblotting.

Question #4 – *The data presented in Figures 3D-E do not provide sufficient evidence to confirm the colocalization of MGAT1 and CD73 in the Golgi apparatus. Additional experimental data or higher-resolution images are needed to substantiate this claim.*

We appreciate the reviewer's suggestion to strengthen the evidence supporting the colocalization of MGAT1 and CD73 in the Golgi apparatus. In response, we have taken the following steps to improve the quality and rigor of the presented data: (1) We have acquired new confocal images with higher resolution to replace the previous Fig. 3d, e. (2) To further validate the colocalization of MGAT1 and CD73, we have performed super-resolution microscopy, which provides superior spatial resolution compared to conventional confocal microscopy. The super-resolution imaging data (Fig. 3f) clearly demonstrate the colocalization of MGAT1 and CD73 within the Golgi, reinforcing our findings. (3) In addition to imaging, we conducted subcellular fractionation of Golgi and membrane compartments followed by immunoblot analysis and the methodology has been included in methods (Huang et al., Cell Rep. doi: 10.1016/j.celrep.2022.111679; Tarazón et al., STAR Protoc. 2020). The results show that both MGAT1 and CD73 are enriched in the Golgi fraction (Supplementary Fig. 4d), further substantiating their colocalization in this compartment. The newly added data and related description has been added to the manuscript from page 11 line 230-242: "To examine the cellular compartmentalization of the interaction between MGAT1 and CD73, we employed multiple complementary approaches. First, we conducted immunofluorescence staining to co-stain MGAT1 and CD73 followed by imaging with confocal imaging. We observed that MGAT1 (green) colocalized with CD73 (red) in the Golgi body verified by GM-130, a well-established Golgi body indicator³⁸⁻⁴⁰ (Fig. 3d, e). Additionally, super-resolution microscopy further validated the colocalization of MGAT1 and CD73, providing superior spatial resolution and reinforcing the findings observed in confocal imaging (Fig. 3f). We also conducted a proximity ligation assay to corroborate the colocalization of MGAT1 and CD73 in the Golgi compartment (Fig. 3g). To complement the imaging data, we conducted subcellular fractionation to separate the Golgi and membrane compartments⁴¹⁻⁴³, followed by immunoblotting. The results revealed that both MGAT1 and CD73 were enriched in the Golgi fraction, supporting their co-distribution within this compartment (Supplementary Fig. 4d). Together, these approaches demonstrate that MGAT1

and CD73 colocalize in the Golgi apparatus, where MGAT1 likely mediates CD73 glycosylation to facilitate its functional maturation.”

Updated Fig. 3d

Updated Fig. 3d: d The colocalization of immunostained MGAT1 (green) and CD73 (red) in MDA-MB468 and MDA-MB231 cells was visualized by confocal imaging.

Updated Fig. 3e

Updated Fig. 3e: e The subcellular localization of MGAT1 was detected through colocalization of MGAT1 (green) and Golgi indicator, GM-130 (red), in MDA-MB468 and MDA-MB231 cells by confocal imaging.

Fig. 3f

Fig. 3f: f The intracellular interaction between MGAT1 (green) and CD73 (red) in MDA-MB468 cells was visualized by immunofluorescence stimulated emission depletion microscopy imaging followed by 3D reconstruction by Imaris.

Supplementary Fig. 4d

Supplementary Fig. 4d: d The colocalization of MGAT1 and CD73 in the Golgi was validated by fractionating the Golgi and membrane sections of the cells followed by immunoblot.

Question #5 – *The co-immunoprecipitation experiment shown in Supplementary Figure S3H only includes the IP results. The input data is missing and should be provided to fully validate the findings.*

We thank the reviewer for pointing out the need for input controls to validate the co-immunoprecipitation results in **Supplementary Fig. 8b** (previous **Supplementary Figure S3H**). In response, we have repeated the co-immunoprecipitation experiments and now provide both input and IP data to ensure a complete and rigorous presentation. The updated results are included in the revised **Supplementary Fig. 8b**

Updated Supplementary Fig.8b

b

Updated Supplementary Fig. 8b: b The biochemical interaction between CD73 and potential transport proteins was validated by coimmunoprecipitation of V5-CD73 in MDA-MB468 and MDA-MB231 cells.

Question #6 – *In several instances, the loading controls in the Western blot data are faint and unclear. The clarity of these bands should be improved for better interpretation.*

We appreciate the reviewer's observation regarding the clarity of the loading controls in the Western blot data. To address this concern, we have repeated the relevant experiments to improve the quality and clarity of the loading control bands. Specifically, we have replaced the unclear bands in **Fig. 5a, Supplementary Fig. 6d, and Supplementary Fig. 8b**, ensuring that the bands are now clear and suitable for accurate interpretation.

Question #7 – *The E0771 cell line name is repeatedly misspelled as "EO771" throughout the manuscript. This typographical error should be corrected.*

We thank the reviewer for catching this typographical error. We have now corrected all instances where the E0771 cell line name was misspelled as "EO771" throughout the manuscript, including the main text, figure legends, and supplementary materials.

Reviewer #3, General Comments:

The authors describe the identification of MGAT1, a glycosyltransferase, as a key factor in immune evasion in immune-cold tumors, particularly in triple-negative breast cancer (TNBC). The authors show that MGAT1 overexpression promotes CD73 translocation, suppressing CD8+ T cell activity. The authors show that MGAT1 adds N acetylglucosamine to CD73, which facilitates its dimerization and membrane fusion. Finally, the authors present a new inhibitor called W-GTF01, which it is shown to blocks MGAT1 activity thereby restoring immune response and improving anti-PD-L1 therapy in TNBC models. The manuscript is well written, and the results are well documented. However I have the following remarks:

We thank the reviewer for recognizing the key aspects of our study, including the identification of MGAT1 as a key factor in immune evasion, its role in facilitating CD73 dimerization and translocation, and the therapeutic potential of the W-GTF01 inhibitor in TNBC models. We greatly appreciate the reviewer's positive feedback regarding the well-written manuscript, the clarity of the results, and the rigorous documentation of our findings. To further strengthen the manuscript and address the reviewer's comments, we have incorporated new experimental data and additional analyses to validate and expand our findings.

Reviewer #3, Specific Comments:

Question #1 - *Figure 1A: the authors poorly describe the figure, assuming that the reader will understand where the data is. This is a complex panel that needs to be properly explain for the reader to understand the rest of the figure. For instance, the clusters mentioned in the results are not clear to see, since many proteins are shown and many seem to be elevated according to the results. Please explain what is what and guide the reader.*

We appreciate the reviewer's comment regarding the need for clearer explanations of **Fig. 1a** and the associated clustering analysis. We agree that additional details will help guide the reader in understanding the data and its relevance to the study. To address this concern, we

carefully revised the text to provide a clearer and more detailed explanation of Fig. 1a. Specifically: (1) We now explicitly describe the hierarchical clustering approach and the data source (TCGA) to set the context for the analysis. (2) We clarify how immune-relevant proteins such as VTCN1, CD274, and CD73 were identified and emphasize their enrichment within the basal subtype (TNBC) compared to other subtypes. (3) We describe the dendrogram clusters in more detail and highlight the presence of two distinct immune-related subgroups within TNBC.

The revised text has been added to the manuscript page 7 line 125-133: “To search for potential therapeutic targets for immune-cold breast cancer, we performed a hierarchical clustering on the gene expression data of 112 immune-relevant proteins from 935 breast cancer patients in The Cancer Genome Atlas (TCGA). Our analysis identified elevated levels of immune-relevant proteins, including VTCN1, CD274, and CD73, specifically within the basal-like subtype of breast cancer (BLBC), compared to other subtypes (Fig. 1a and Supplementary Fig. 1a). In addition, hierarchical clustering analysis suggested two distinct subgroups within TNBC patients, each characterized by a unique immune-related protein expression profile, as shown in Fig. 1a. One subgroup, highlighted in the dotted frame, exhibited a significantly higher expression of immune-related gene signatures compared to the rest of the TNBC samples.”

Updated Fig. 1a

Updated Fig. 1a: a Distribution of single sample GSEA scores of BLBC separated by immune subgroup, using gene sets involved in immune functions (b) and glycosylation (c). P-values are from Wilcoxon tests. * $p < 0.05$, ** $p < 0.01$, *** $p < 0.001$, and **** $p < 0.0001$.

Question #2 - Figure 1B: indicate what is red and blue, now it only says category. It is explained in the text, but the figure should speak by itself.

We appreciate the reviewer's suggestion. We have revised the both figure and figure legend for Fig. 1b, c to clearly define the red and blue categories. Specifically, we have labeled the red category as "immune suppressive" and the blue category as "immune active."

Updated Fig. 1b, c

Updated Fig. 1b, c: b, c Distribution of single sample GSEA scores of BLBC separated by immune subgroup, using gene sets involved in immune functions (b) and glycosylation (c). P-values are from Wilcoxon tests. *p<0.05, **p<0.01, ***p<0.001, and ****p<0.0001.

Question #3 - *What is BLBC? Don't expect the reader to know.*

We apologize for introducing the abbreviation BLBC (Basal-Like Breast Cancer) without proper explanation, which may have caused confusion. To address this, we have provided a clear definition of BLBC where it first appears in the manuscript and include the full name in Fig. 1b, c.

Updated Fig. 1b, c

Updated Fig. 1b, c: **b, c** Distribution of single sample GSEA scores of BLBC separated by immune subgroup, using gene sets involved in immune functions (**b**) and glycosylation (**c**). P-values are from Wilcoxon tests. * $p < 0.05$, ** $p < 0.01$, *** $p < 0.001$, and **** $p < 0.0001$.

Question #4 - *Figure 1D: explain what the correlation is against immune suppression genes, how is this list obtained? Manually or through the analysis?*

We appreciate the reviewer's question regarding the generation of the immune suppression gene list and its correlation analysis in Fig. 1d. The list of immune suppression genes was obtained through a combination of literature review and publicly curated databases such as MSigDB. The selected genes include well-established immune checkpoint molecules

(CD274/PD-L1, VTCN1, CTLA4) and other immune-related markers involved in T cell inhibition and immune evasion (Thorsson et al., Immunity. 2018. doi:10.1016/j.immuni.2018.03.023; Bagaev et al., Cancer Cell. 2021. doi: 10.1016/j.ccell.2021.04.014.; Bejarano et al., Cancer Cell. 2021. doi: 10.1158/2159-8290.CD-20-1808; Song et al., Nat Commun. 2024. doi: 10.1038/s41467-024-54434-4). The correlation analysis was systematically performed using Spearman correlation analysis to assess the relationship between major N-glycan biosynthesis genes expression and the expression of the immune-related genes across breast cancer datasets from The Cancer Genome Atlas (TCGA). This unbiased bioinformatic approach identified a strong positive correlation between MGAT1 expression and key immuno markers, particularly within the TNBC cohort.

To improve clarity and address this, we have added the full list of major N-glycan biosynthesis genes and the immune response-related gene list used for the analysis in the Supplementary Fig. 2a. Additionally, we have provided further details in the manuscript (Page 7 line 145 to page 8 line 151): “Given the critical role of aberrant N-glycosylation in regulating tumor immunity^{9,34-36}, we undertook Pearson correlation analysis using a curated list of immune-related genes obtained from MSigDB³⁷ to identify if specific N-glycan processing-related genes are implicated in this particular TNBC subgroup. As shown in Fig. 1d, the results clearly showed that multiple enzymes involved in N-glycosylation correlate positively with genes driving immune suppression and the detailed gene list is included in Supplementary Fig. 2a.. Notably, MGAT1 emerged as the most significantly associated enzyme, topping the list of genes linked to this effect within the TNBC subgroup (Supplementary Fig. 2a). ”

Supplementary Fig. 2a

a

Full N-glycan biosynthesis gene list (row) and immune related gene list (column) used for Pearson correlation analysis in Fig. 1d in order						
Row (1-23)	Row (24-46)	Column (1-23)	24-46	47-69	70-92	93-112
MGAT1	ALG6	BTN2A1	ADORA2A	SMAD3	IFNG	ADA
ST6GAL1	DDOST	PVR	B2M	CD274	PDCD1	CD27
MGAT4A	DOLPP1	CD80	CD55	FOXP3	TGFB2	GNLY
MAN2A1	ALG1	RNF10	CR1L	NCR1	CTLA4	IL6
MGAT2	RPN2	ADORA2B	HAVCR2	PDCD1LG2	IRF7	LY96
ALG2	DPAGT1	BTN2A2	NCR3LG1	POMC	ALCAM	TDO2
MAN1A2	ALG8	BTN3A1	TAP1	PRF1	MIF	TGFBR2
ALG11	MGAT5B	CD276	CD46	AIM2	TGFB1	IDO1
MAN1C1	ALG5	TGFBR1	CXCR4	EMB	TIGIT	IL1RN
MAN1A1	TUSC3	VTCN1	FGFR3	FASLG	TMIGD2	VIP
ALG13	STT3A	CEACAM1	HMOX1	HEBP1	BTLA	CD19
ALG9	MAN1B1	STAT1	SMAD2	HLA-A	CD226	CD40
MGAT4B	DPM1	THBS2	CD38	HLA-G	HLA-F	CALCA
MGAT3	B4GALT3	IFIT3	CD47	ICOS	NCR2	CD14
FUT8	RPN1	CD59	HLA-B	IL12A	SEC14L2	CD70
MAN2A2	GANAB	IFNGR2	HLA-C	IL1R1	ARG1	SLC9A3R2
ALG10B	MOGS	SERPINE1	IDO2	ITGAX	DPP4	LGALS1
ALG10	DAD1	ARG2	NT5E	LAG3	GZMA	GATA3
MGAT5	ALG3	CD86	ENTPD1	MX2	HHLA2	LGALS3
ALG12	DPM2	PROM1	IFNGR1	CD160	HLA-E	CSTA
STT3B	ALG14	SIRPA	KDR	CD28	IRF1	
B4GALT1	DPM3	THBS1	MMP7	DPH6	NCR3	
RFT1	B4GALT2	TLR2	PIK3CG	ENPP1	ABHD17A	

Supplementary Fig. 2a: a Full N-glycan biosynthesis gene list (row) and immune related gene list (column) used for Pearson correlation analysis in Fig. 1d in order.

Question #5 - *Figure E: Basal is missing, are samples available? If so, it should be included.*

We thank the reviewer for raising this point regarding the inclusion of Basal-like breast cancer in **Fig. 1e**. The omission of Basal-like classification is due to the nature of our tissue array data, which only includes information on estrogen receptor (ER), progesterone receptor (PR), and HER2 expression status—the standard markers used to clinically define triple-negative breast cancer (TNBC). To clarify, in **Fig. 1a-d**, the classification of Basal-like breast cancer (BLBC) is based on molecular profiling using the PAM50 classification from TCGA datasets. BLBC is a molecular subtype characterized by the expression of basal cytokeratins (e.g., CK5/6, CK14) and other basal/myoepithelial markers (Sørli et al., 2001; Perou et al., 2000). In contrast, TNBC is a clinical classification defined by the absence of ER, PR, and HER2 expression, determined via immunohistochemistry (IHC) (Foulkes et al., 2010). It is important to note that there is significant overlap between the two classifications, as 70–80% of TNBC tumors are also categorized as

Basal-like through molecular profiling (Carey et al., 2006; Foulkes et al., 2010). However, due to the absence of gene expression data in our tissue array, we rely on the clinical TNBC definition for consistency and relevance.

Question #6 - *Figure 1G: how were the interactions quantified? How many samples were analysed? How were the levels of MGAT1 assessed to divide into high and low?*

We thank the reviewer for raising this question and allowing us to clarify the methodology used in **Fig. 1g**. The touching interactions between selected cellular populations in **Fig. 1g** (e.g., PanCK+ tumor cells and CD8⁺ T cells or CD8⁺Ki67⁺ T cells) were quantified using the "Spatial Map Reviewer" tool from the R-based phenoptrReports software by AKOYA Biosciences. This tool allowed us to systematically analyze and quantify spatial interactions in a tissue microarray (TMA) containing 41 specimens. The expression levels of MGAT1 in PanCK+ tumor cells were assessed using the automated image analysis software from AKOYA Biosciences. The MGAT1 expression levels were categorized into high and low groups based on the median expression value as the threshold, ensuring an unbiased division of the samples.

To further improve clarity and transparency, we have included these details in the Methods section of the revised manuscript, specifically under the "Spatial Analysis and Quantification" subsection from page 44 line 1111 to page 45 line 1118: "The spatial interactions between PanCK⁺ tumor cells and CD8⁺ T cells or CD8⁺Ki67⁺ T cells were analyzed using the "Spatial Map Reviewer" tool within the R-based phenoptrReports software (AKOYA Biosciences). This tool was applied to quantify touching interactions between cellular populations in tissue microarray (TMA) sections comprising 41 specimens. The expression levels of MGAT1 in PanCK+ tumor cells were quantified using the automated image analysis software inForm (AKOYA Biosciences). MGAT1 expression was stratified into high and low groups based on the median expression value, serving as an unbiased threshold for comparison. All analyses were performed on multiplex-stained TMA slides to ensure precise quantification of both MGAT1 expression and cellular interactions."

Question #7 - *Figure 2B: what are the green cells? Are these really the MDA-MB468-MGAT1-KD as indicated in the legend? It looks as if the cells are cultured in the presence of a Caspase 3/7 substrate that when cleaved due to the ongoing apoptosis, the cells turn “green” fluorescent? This is a typical co-culture experiment to show that the OE and KD are differentially killed by activated T-cells. Please clarify.*

Thank you for the question regarding **Fig. 2b** and the identification of the green cells. To clarify, in the 2D co-culture model, we used MDA-MB468 MGAT1-WT/OE/KD TNBC stable cell lines co-cultured with immune cells. Quantifying cancer cell survival solely in the bright field was challenging due to the mixed cell populations. To address this, we pre-stained the cancer cells with CellTracker Green CMFDA (ThermoFisher Scientific), a non-toxic fluorescent dye that remains stable in cells for more than 72 hours. This approach allowed us to specifically track and quantify the survival of cancer cells in the presence of immune cells. The green fluorescence in the images represents the cancer cells, and its intensity reflects the relative survival of the cancer cells under immune cell challenge. In addition, we have added related text in the manuscript from page 9 line 188-192: “To this end, we established Flag-tagged MGAT1 overexpression (OE) and MGAT1 stable knockdown (KD) in MDA-MB-468 cells based on the lentivirus and CRISPR/Cas9 system^{31,33} and the stable cell lines were pre-stained with CellTracker Green CMFDA (ThermoFisher Scientific), a fluorescent dye stable for over 72 hours. The relative survival of cancer cells in the presence of immune cell-mediated cytotoxicity was quantified based on coverage of stained cells.”

Question #8 - *Figure 2B: the cell proliferation curves don't seem to correspond to the images. The OE cells are not proliferating at the same rate as the WT (as it seems the case in the quantification) according to the images. Please clarify.*

Thank you for your observation regarding the cell proliferation curves and images in **Fig. 2b**.

Upon review, we realized that the first set of images, while representative, may not have fully 55

captured the dynamic range of cell survival differences between MGAT1-WT, OE, and KD lines during immune cell-mediated killing. To address this, we have repeated the co-culture experiments and captured new time-lapse imaging data to ensure consistency between the quantification and the visual representation. The updated images in Fig. 2b clearly demonstrate that MGAT1 overexpression in breast cancer cells enhances their survival under immune cell challenge, consistent with the quantification results. Conversely, MGAT1 knockdown renders the cancer cells more vulnerable to immune cell-mediated killing.

Updated Fig. 2b

b

Updated Fig 2b: The proliferation of MDA-MB468, MDA-MB468-MGAT1-OE, and MDA-MB468-MGAT1-KD cells (green) in 2D coculture with pre-activated PBMCs was monitored by time-lapse image-based quantification. The data are presented as mean \pm SEM from three replicates from a representative experiment.

Question #9 - Figure 2F vs 2B: what is the difference between these two experiments? In 2F the T-cells are activated? But the outcome is the same?

Thank you for your question. We noticed that the experimental conditions in both figures are similar, with the primary difference being the immune cell populations used: In Fig. 2b, we used

PBMCs (Peripheral Blood Mononuclear Cells) in the co-culture system. PBMCs represent a mixed population of immune cells, including CD8⁺ T cells, CD4⁺ T cells, natural killer cells, and monocytes, providing a broader immune context. In **Fig. 2f**, we specifically used purified CD8⁺ T cells and activated by anti-CD3/CD28 beads, isolating the key effector population responsible for cytotoxicity. This setup is more focused and designed to directly assess the impact of MGAT1 expression in breast cancer cells on CD8⁺ T cell activation and proliferation. While the outcomes in both experiments appear consistent, with MGAT1 overexpression promoting cancer cell survival and MGAT1 knockdown increasing vulnerability to immune-mediated killing, **Fig. 2f** specifically demonstrates that these effects are linked to the modulation of CD8⁺ T cell function. This provides more direct evidence for the interaction between MGAT1-expressing cancer cells and cytotoxic T cells. To avoid confusion, we also added the following information in the manuscript (page 10 line 209-211): “Consistent with the PBMC coculture system, the coculture of MGAT1 stable cell lines with purified and pre-activated CD8⁺ T cells leads to a similar inhibition on CD8⁺ T cell activation (**Fig. 2f, g**).”

Updated Figure legend: “**Fig. 2f: f** The proliferation of MGAT1-WT/OE/KD MDA-MB468 cells (green) in 2D coculture with pre-activated purified CD8⁺ T cells was monitored by time-lapse image-based quantification. The data are presented as mean ± SEM from three replicates from a representative experiment.”

Question #10 - *Figure 3l: what does “big datasets” in line 261 of the text mean?*

Thank you for your question regarding the source of the “big datasets” in **Supplementary Fig. 4f** (previous **Fig. 3i**). These data were derived from the publicly available CPTAC dataset (PDC Study Identifier: PDC000408), as described in the study by Anurag M., Jaehnig EJ., Krug K., Lei JT., et al. (2022) (DOI: 10.1158/2159-8290.cd-22-0200). We have included this information in the manuscript to clarify the source of the data (page 11 line 251 to page line 254): “Consistent with

the above results, the big datasets with Spearman's rank correlation analysis using proteomic datasets from the PDC000408 cohort⁴⁵ clearly confirmed the correlation between MGAT1 protein levels with several immune regulators, especially CD73 (NT5E) (Supplementary Fig. 4f).”

Question #11 - *Figure 3K: what do CD73 and MGAT1 Area mean? What samples were measured? Please clarify.*

Thank you for your suggestion regarding Fig. 3j (previous Fig. 3k). To improve clarity and address your concern, we have updated the panel to include representative IHC images of MGAT1 and CD73 staining in human TNBC tissue sections. These updated images provide a more comprehensive view of the staining patterns and demonstrate the specific regions used for quantification. The "CD73 and MGAT1 Area" refers to the quantification of IHC staining-positive regions for CD73 and MGAT1 within the tumor tissues. The samples analyzed were derived from TNBC patient tissue sections (n=23). The quantification of positive staining areas was conducted using QuPath, a professional image analysis software, which enables precise measurement of IHC signals by detecting and quantifying the stained regions. We have updated the corresponding figure legend on page 54 line 1461-1464: “Fig. 3j: Representative IHC staining of MGAT1 and CD73 in human TNBC tissue sections (n=23). Quantification of positive staining areas using QuPath reveals a significant positive correlation between MGAT1 and CD73 expression.”

Updated Fig. 3j

j

Updated Fig. 3j: j Representative IHC staining of MGAT1 and CD73 in human TNBC tissue sections (n=23). Quantification of positive staining areas using QuPath reveals a significant positive correlation between MGAT1 and CD73 expression.

Question #12 - Figure 5 In line 314 of the main text, the authors state that MGAT1 KD resulted in a reduced CD73 glycosylation. While this seems to be true for MDA-MD468 cells, it does not seem to be the case for MDA-MB231, or at least, there seems to be equally much CD73 glycosylated and non-glycosylated levels. Can the authors speculate about these cell-specific differences?

We thank reviewer for pointing out the cell-specific differences regarding CD73 glycosylation levels in Figure 5. The observed differences between MDA-MB468 and MDA-MB231 cells were likely due to variations in knockdown efficiency in the initial MDA-MB231 stable cell line batch, which resulted in residual MGAT1 expression. This partial knockdown may have allowed the presence of both glycosylated and non-glycosylated CD73, leading to an unclear pattern. To address this, we have reconstructed the MGAT1-knockdown MDA-MB231 stable cell lines with improved efficiency and validated the CD73 glycosylation levels using Western blot analysis. In this updated experiment, we observed a significant reduction of fully glycosylated CD73,

consistent with the pattern seen in MDA-MB468 cells. In addition, the updated description has been added to the revised manuscript from page 14 line 332 to page 15 line 336: “As described above, we initially engineered MGAT1 OE and KD in MDA-MB468 and MDA-MB231 cells. As shown in Fig. 5a, while a minor change in CD73 abundance in response to MGAT1 OE was observed, reduced glycosylation of CD73 (running smaller molecular mass ~60 kDa) was clearly detected when MGAT1 was knocked down, confirming the aforementioned CD73 glycosylation catalyzed by MGAT1.”

Updated Fig. 5a

Updated Fig. 5a: a The CD73 protein levels and molecular sizes in MDA-MB468 and MDA-MB231 cells with MGAT1 OE and MGAT1 KD were determined by immunoblotting

Question #13 - *Figure 5D: what is the membrane dye for? Please clarify what is the meaning of this observation in the text.*

Thank you for your question regarding Fig. 5d. The membrane dye used in this experiment is Biotium’s CellBrite® NIR680 Cytoplasmic Membrane Dye, which specifically labels the cytoplasmic membranes in live or formaldehyde-fixed cells. This dye was used to visualize and highlight the membrane localization of CD73 in our confocal imaging experiments. The purpose of this observation was to further validate the results obtained by flow cytometry, which demonstrated that MGAT1 overexpression significantly enhances the membrane distribution of

CD73, while MGAT1 knockdown reduces its membrane localization. The confocal imaging with the membrane dye provides spatial confirmation of these findings, visually reinforcing the role of MGAT1 in regulating CD73 translocation to the cell membrane. In addition, we have added related information to the manuscript in page 15 line 353-357: “The above result was further confirmed by confocal microscopy imaging using Biotium’s CellBrite® NIR680 Cytoplasmic Membrane Dye, which specifically labels the cytoplasmic membranes of live or formaldehyde-fixed cells (Fig. 5d) and whole-cell lysate fractionation followed by immunoblot (Supplementary Fig. 6d).”

Question #14 - *In line 376 of the main text, the authors again elude to a Spearman’s rank correlation, but it is totally unclear what data was used. Please clarify.*

Regarding the Fig. 5l (previous Supplementary Fig. 4e) Correlation Analysis, the Spearman’s rank correlation analysis shown in Fig. 5l (previous Supplementary Fig. 4e) was performed using proteomic datasets from the PDC000408 cohort (Anurag M., et al., Cancer Discov., 2022), which includes comprehensive proteomic profiles of TNBC tumors. We have included this information in the manuscript to clarify the source of the data (page 18 line 414-418): “To investigate the upstream signaling that potentially modulates MGAT1 in the context of tumor invasion, we conducted Spearman’s rank correlation analysis using proteomic datasets from the PDC000408 cohort⁴⁵ and found a tight correlation between elevated expression of MGAT1 and up-regulation of THBS1 and THBS2 signaling pathways (Fig. 5l).”

Question #15 - *Figure 6: What exact compound library was used for this screen? Is it an “in-house” library? A repurposing library? Please clarify.*

Thank you for your question regarding the compound library used in Fig. 6b. The library we utilized for this screening is the Anti-Cancer Compound Library (HY-L025) from MedChemExpress, a widely used and well-curated collection of bioactive molecules. This library contains compounds designed to target key regulators of cancer progression, including kinases,

cell cycle components, and pathways related to tumorigenesis, immune modulation, and metabolic reprogramming. This library was chosen for its diversity and relevance to cancer research, providing an extensive set of potential candidates for identifying inhibitors of MGAT1 or its related pathways. Importantly, it contains both investigational compounds and clinically approved molecules, enabling the identification of candidates with translational potential for therapeutic development.

To ensure clarity, we have included detailed information about the compound library in the revised manuscript from page 18 line 428-431: “We initially developed an *in vitro* high-throughput screening assay to monitor the effect on MGAT1 enzymatic activity of a library of putative inhibitors (Fig. 6a). An anti-cancer compound library (HY-L025 from MedChemExpress) was subjected to screening of inhibitors that could block the assembling glycan chain.”

Question #16 - *What is known about TW-37? Any references to its specificity and uses? Please provide references.*

Thank you for your question regarding TW-37 and its role in our study. TW-37 is a non-peptide small-molecule inhibitor targeting the Bcl-2 family proteins, specifically Bcl-2 and Mcl-1, by binding to the BH3-binding groove (Mohammad et al., Clin Cancer Res. 2007. doi: 10.1158/1078-0432.CCR-06-1574). It has also been demonstrated to have anti-angiogenic effects, inducing apoptosis in endothelial cells. (Zeitlin et al., Cancer Res. 2006. doi: 10.1158/0008-5472.CAN-05-3691). Interestingly, our study is the first to report that TW-37 directly inhibits MGAT1 activity, revealing a novel function beyond its canonical role. Through screening efforts, we identified TW-37 as a compound capable of modulating glycosylation-dependent pathways by targeting MGAT1. Building on this discovery, we performed structural simulations and virtual screening based on the docking information between TW-37 and MGAT1, identifying 15 novel compounds as potential candidates capable of targeting MGAT1. These compounds were further screened and

experimentally validated to assess their ability to modulate MGAT1-mediated glycosylation processes.

We have included these findings in the revised manuscript, along with relevant references, (page 18 line 428-433): “We initially developed an *in vitro* high-throughput screening assay to monitor the effect on MGAT1 enzymatic activity of a library of putative inhibitors (Fig. 6a). An anti-cancer compound library (HY-L025 from MedChemExpress) was subjected to screening of inhibitors that could block the assembling glycan chain. We obtained several hits, and compound named TW-37, an inhibitor of recombinant Bcl-2, Bcl-xL, and Mcl-1^{54,55}, was the most potent candidate for inhibiting MGAT1 activity (Fig. 6b).”

Question #17 - *Why trying to identify other compounds based on the structure of TW-37? What was the rationale?*

Thank you for your question regarding our efforts to identify additional compounds based on the interaction information between TW-37 and MGAT1. Our journal to develop MGAT1 small molecule inhibitor started with *in vitro* MGAT1 enzymatic function screening (Fig. 6a), which could quantify the efficiency of compound to inhibit the enzymatic function of MGAT1. In the screening with Anti-Cancer Compound Library (HY-L025) from MedChemExpress, a well-curated collection of bioactive molecules, we surprisingly discovered TW-37 as an effective inhibitor to MGAT1 function *in vitro*. We utilized docking-predicted binding of TW-37 to MGAT1 (Fig. 6c and Supplementary Fig. 9a, b) and use this compound that has been approved to be effective in inhibiting MGAT1 in the *in vitro* system as a reference to identify more potent compounds targeting MGAT1. Through FTMap analysis, we identified high-affinity probe-binding residues of MGAT1, notably R115, Y182, I185, E209, L267, W288, S320 and R413. The information from TW-37 strengthened the construction of efficient virtual screening assay and eventually lead to the discovery of an effective MGAT1 inhibitor, W-GTF01, with more superior inhibiting efficacy. The

structure information of a compound with verified inhibiting function in the experiment strengthens the efficiency of virtual screening to discover more potent inhibitor.

Question #18 - *Figure 6M: have the authors run this experiment for compounds No.2 and No.9? If so, the results should be shown too.*

Thank you for your comment. We have conducted additional experiments to evaluate the effects of compounds No.2, No.8, and No.9 in the cancer and immune cell co-culture model to assess their impact on the antitumor immune response. These updated results have been included in Fig. 6m. The related description has been added to the manuscript from page 19 line 466 to page 20 471: “These candidates were further evaluated using cancer and immune cell coculture to assess the effect on the antitumor immune response. The CD8⁺ T cell function was detected using flow cytometry (Fig. 6l) and cancer cell survival was examined with image-based viability (Fig. 6m). Based on evaluation standards from multiple layers, W-GTF01 (No.8 compound) was confirmed as a potent inhibitor of MGAT1 activity and stimulator of CD8⁺ IFN γ -producing T-cell response, leading to lowest cancer cell survival.”

Updated Fig. 6m

Updated Fig. 6m: m The survival of MDA-MB468 cells in coculture with pre-activated PBMCs and No.2/No.8/No.9 was monitored by time lapse image-based quantification. The data are presented as mean \pm SEM from three replicates of a representative experiment.

Question #18 - *Figure 7: In line 478 of the main text, the authors conclude that there is an increased TAM presence in the in vivo experiment with MGAT1 OE cells, and that this is a sign of a pro-inflammatory M2-like macrophage response. However, the levels of IFN γ + and TNF α +, two pro-inflammatory cytokines are reduced. Can the authors please clarify or elaborate?*

We thank the reviewer for this comment. Our data show that MGAT1 overexpression (OE) in tumor cells enhances the presence of tumor-associated macrophages (TAMs) with a tumor-promoting M2-like phenotype (Fig. 7g). M2-like macrophages are characterized by their anti-inflammatory and immunosuppressive properties, which support tumor progression by inhibiting CD8⁺ cytotoxic T cell activity, promoting angiogenesis, and enhancing tissue remodeling. This is distinct from the M1-like macrophages, which exhibit pro-inflammatory properties and contribute to antitumor immune responses.

In alignment with the M2-like phenotype, we observed a significant reduction in IFN γ and TNF α secretion by infiltrating CD8⁺ T cells in MGAT1 OE tumors (Fig. 7h). These cytokines are critical for antitumor immune responses, and their decreased levels suggest a suppression of CD8⁺ T cell activity in the tumor microenvironment. Together, these findings support the notion that MGAT1 expression in tumor cells promotes immune evasion by shifting TAMs toward a tumor-promoting M2-like phenotype and suppressing the functionality of infiltrating CD8⁺ T cells.

Question #19 - *Figure 7K: why did the authors measure Tim3+PD1+ in CD8+ T cells? Please elaborate. It's just that it comes a bit as a surprise and there is no explanation.*

We appreciate reviewer's comment. Tim-3⁺PD-1⁺ cells within the CD8⁺ T cell population are widely recognized as a hallmark of T cell exhaustion, characterized by impaired functionality, reduced cytokine production, and diminished ability to control tumor progression (PMCID: PMC2947065). These exhausted T cells are often associated with poor prognosis and tumor immune evasion. In our study, we analyzed Tim-3⁺PD-1⁺ cells to assess the exhaustion status of CD8⁺ T cells in the tumor microenvironment. Interestingly, we found that there was a decreased

fraction of exhausted-like Tim3⁺PD1⁺ and an increased fraction of proliferating CD8⁺ T cells expressing a higher level of Ki67 (Fig. 7k, l) in MGAT1 knockdown tumor-bearing mice compared to that in the control mice, indicating great potential of inhibiting MGAT1 expression to revive antitumor CD8⁺ T cells and suppress tumor growth.

Question #20 - *Figure 7Q: these results are remarkable. Did the authors measure the amount of exhausted T-cells on the mice treated with W-GTF01+anti-PDL1?*

Thank you for your positive comment on the results in Fig. 7p (previous Fig. 7q). To address your question, we performed additional IHC analyses to evaluate the levels of exhausted T cells in mice treated with W-GTF01 combined with anti-PD-L1 antibody. These analyses specifically assessed markers associated with T cell exhaustion, such as TOX and TCF, within the tumor microenvironment. The new data and related description have been added to the manuscript page 22 line 550-page 23 line 558: “In addition, we also observed that the combination of W-GTF01 and anti-PD-L1 significantly improves survival outcomes compared to either treatment alone (Fig. 7q), reinforcing the conclusion that targeting MGAT1 enhances the efficacy of immune checkpoint blockade. The IHC images showed that the combination of W-GTF01 and PD-L1 antibody markedly increased CD8⁺ T cell infiltration and activation (Supplementary Fig. 10m). Additionally, we observed a moderate decrease in T cell exhaustion, as evidenced by reduced Tox staining, when compared to treatment with anti-PD-L1 antibody alone. These outcomes suggest the therapeutic potential of targeting MGAT1 pharmacologically to enhance immunogenic cell death in response to current immunotherapy.”

Supplementary Fig. 10m

Supplementary Fig. 10m: m Representative images of IHC staining of mouse CD8, granzyme B and TOX level of mouse tumors after the treatment of IgG, or anti-PD-L1 antibody treatment or W-GTF01 + anti-PD-L1 antibody combination. Scale bar = 40 μm.

Question #21 - Figure 8: What data was used for Figures 8H to 8I?

Thank you for your question regarding Fig. 8h, i. These panels represent data derived from pathway enrichment analyses, including KEGG, Reactome, and Gene Ontology (GO), based on spatial transcriptomic profiling performed using the GeoMX Digital Spatial Profiler (DSP) platform. This advanced technique allows high-resolution spatial transcriptome analysis, enabling the evaluation of gene expression and pathway activation within specific regions of the tumor microenvironment. The spatial transcriptomics data were obtained from tumor samples categorized by high and low MGAT1 expression. Using the DSP platform, we identified distinct molecular signatures and pathways enriched in different tumor compartments. The data from Fig. 8h, i highlight significant pathway enrichments linked to cleavage of basic residues, palmitate, disulfide bond and glycoprotein pathways in regions with double elevated MGAT1 and CD73 expression. These findings align with the broader narrative of MGAT1's role in modulating the tumor immune microenvironment and its potential as a therapeutic target.

Further methodological details regarding spatial transcriptomics, pathway enrichment analyses, and data processing have been added to the Materials and Methods section for clarity and transparency.

Minor point #1 - *Figure 2A: in my personal opinion, the comical cells are unnecessary and don't bring rigor to the manuscript, on the contrary.*

Thank you for the suggestion! We have removed the comical cells in **Fig. 2a**.

Updated Fig. 2a

Updated Fig. 2a: a Schematic diagram of the 2D and 3D coculture assays.

Minor point #2 - *Figure 3A: the mass spectrometry results are impossible to read, suggestion to make them bigger and put them in the supplementary.*

We thank reviewer's suggestion. To address this, we have enlarged the mass spectrometry spectrum for improved readability and moved it to the supplementary materials as recommended.

Please see the **Supplementary Fig. 4a** in the supplementary section.

Supplementary Fig. 4a

a

Supplementary Fig. 4a: a Representative peptides of CD73 and MGAT1 discovered by mass spectrometry.

Minor point #3 - Figure 5H comes after I in the text, suggestion to consider swapping.

Thank you for pointing this out. We have reorganized the figure panels to ensure they are called out in sequential order within the text. Fig. 5h now appears before Supplementary Fig. 7a as suggested.

Minor point #4 - *Line 392, it is not needed in a screen result to say that the hit was compound number 72...it could have been compound number 450 and it would have still been irrelevant. The important here is the TW-37 compound name*

We thank reviewer's suggestion. To address this, we have removed "compound number 72" from the text and now only use the compound name, TW-37, as recommended.

Reviewer #4, General Comments:

In the manuscript titled 'MGAT1-Mediated Glycosylation Orchestrates Immune Checkpoints and Antitumor Immunity', the authors investigate MGAT1/GnT1, a glycosyltransferase involved in CD73-mediated inhibition of CD8+T cells in immune-cold triple-negative breast cancer. The authors performed bioinformatic analysis of a subset of the TCGA database, and found that a number of enzymes in protein N-glycosylation pathway, particularly MGAT1, are associated with unfavourable tumor immune responses and poor prognosis in immune-cold breast cancers. Next, the authors co-cultured PBMCs with a breast cancer cell line overexpressing or knocked down for MGAT1 and conducted flow cytometry analysis. They found that MGAT1 expression in cancer cells modulates the expression of pro-inflammatory genes, including TNF α and IFN γ in CD8+ T cells. Confocal microscopy analysis further demonstrated the colocalization of MGAT1 and CD73 in the Golgi apparatus.

Based on these findings, the authors explored the molecular mechanism of MGAT1 and CD73 interactions using in silico docking and simulation. They validated the predicted MGAT1-CD73 binding and proposed a model of the complex. Notably, the authors discovered that MGAT1-mediated glycosylation of CD73 promotes its dimerization and localization to plasma membrane. Finally, the authors screen a small compound library and identified W-GTF01 as a potent inhibitor of MGAT1 activity.

Protein N-glycosylation is prevalent in most membrane receptors and transporters. MGAT1/GnT1 is a key glycosyltransferase in N-glycosylation biosynthesis pathway, controlling the conversion of oligomannose glycan to hybrid/complex N-glycan structures. Previous studies have underscored the importance of MGAT1/GnT1 in various diseases. The loss of function mutations of MGAT1/GnT1 leads to aberrant glycosylation of cell surface receptors and transporters, resulting in dysregulated down-stream pathways. Here, the authors focus on elucidating MGAT1-mediated glycosylation of CD73 and investigate its biological consequences. In summary, these results links MGAT1 activity and CD73 glycosylation to immune checkpoints and antitumor

immunity. I find this work is not suitable for publication in Nature Communications in current version.

We thank the reviewer for their thoughtful assessment and for recognizing the key aspects of our work. We appreciate the reviewer's expertise in structural modeling, protein-protein interactions, and glycobiology. To address the reviewer's comments and strengthen the study, we have incorporated new experimental data and additional analyses to further support our findings and address the raised concerns. These revisions aim to provide a more rigorous and complete understanding of the MGAT1-CD73 axis and its role in immune evasion.

Reviewer #4, General Comments:

Question #1 - *CD73 is a nucleotidase with at least three N-glycans. Therefore, it is a plausible substrate for MGAT1. However, the authors presented some conflicting results regarding the proposed interface of MGAT1- CD73 interactions (Figure 4E, 4F 4G and 4H). The authors should re-analysis the in silico predictions of MGAT1-CD73 interaction. If MGAT and non-glycosylated CD73 can form a stable complex (Figure 4F and 4G), does this imply CD73 is an essential component of the glycosylation machinery? Does non-glycosylated CD73 exist in cells? Is the glycan conformation rigid in Figure 4H? Furthermore, do the authors have any control for their in silico predictions? Have they performed AlphaFold prediction of the truncated MGAT1/CD73 (Figure 4A and 4C). Can the truncated MGAT/CD73 maintain their native structures and conformations?*

Thank you for your insightful comments. Below, we address each of your concerns systematically and incorporate additional analysis and clarifications based on your suggestions.

(1) Reanalysis of MGAT1-CD73 Interaction and Binding States (Fig. 4e-h)

Fig. 4e represents the proposed binding pose of MGAT1 and CD73, derived from the mapping results shown in **Fig. 4a-d**. These initial mapping results provided a strong foundation for further analyzing the potential interaction states between MGAT1 and CD73.

Our analysis suggests that MGAT1, as an enzyme, recognizes CD73 in a specific binding state (Fig. 4e), which likely represents the initial interaction required for substrate recognition. This binding pose is supported by both experimental evidence and computational modeling, as further detailed below. Additionally, Fig. 4f, g provide quantitative data on the stability of the complex. To ensure the robustness of our simulations, we extended the simulation duration and performed five independent runs, all of which consistently supported the binding pose illustrated in Fig. 4e. The binding state simulations involved multiple iterations to optimize the stability and reproducibility of the results. Furthermore, we propose that during the catalytic reaction, MGAT1 may adopt another conformation, distinct from the initial binding state, as illustrated in previous Fig. 4h (current Supplementary Fig. 5b). This alternate conformation potentially reflects a state involved in glycosylation of specific asparagine residues on CD73. Alternate poses shown in the previous Fig 4h (Supplementary Fig. 5b) represent additional plausible modes of interaction, which require further experimental validation. Therefore, these alternative poses have been moved to the Supplementary Figures (Supplementary Fig. 5b). Finally, while CD73 and MGAT1 do form a stable complex, this does not necessarily imply that CD73 is an essential component of the glycosylation machinery. Our findings suggest that CD73 functions primarily as a substrate in the context of this interaction, rather than as a structural or catalytic component of the glycosylation process.

(2) Does Non-Glycosylated CD73 Exist in Cells?

While non-glycosylated CD73 is not typically observed under normal physiological conditions, we hypothesize that such a form might exist transiently during specific cellular processes or stress conditions. This hypothesis is supported by studies showing that CD73 can localize to multiple cellular compartments, including the plasma membrane, endoplasmic reticulum (ER), Golgi apparatus, and endosomes. These localizations suggest that CD73 might exist in different post-translationally modified states, including non-

glycosylated forms, as it traverses the secretory pathway or under stress conditions that disrupt glycosylation machinery. However, determining the biological relevance and functional implications of non-glycosylated CD73 will require further experimental investigation, including direct detection using glycosylation-specific assays, localization studies, and functional analyses, which is beyond the scope of the current study.

(3) Is the Glycan Conformation Rigid in Figure 4H?

Yes, the glycan conformation in Supplementary Fig. 5b (previous Fig. 4h) was modeled as rigid. The binding poses in Supplementary Fig. 5b were generated using ZDock. We have included this clarification in the manuscript in page 41 line 1033-page 42 line 1035: "The binding poses of active hMGAT1 to CD73 glycoprotein were predicted in Supplementary Fig. 5b was generated using ZDOCK server⁵⁰, which is based on a rigid-body docking algorithm."

(4) Controls for In-Silico Predictions.

To ensure the reliability of our computational predictions, we adopted the following controls: (a) Multiple runs were performed to test the stability of the structures and verify reproducibility. (b) When experimental data are available, we check the consistence, or we use the data to refine and validate in silico predictions. This approach was applied to generate the binding pose in Fig. 4e. (c) In the case of ZDock simulations, the proposed poses were ranked based on their Z-scores, and only those consistent with experimental data were considered for analysis. Additionally, in the absence of experimental validation, we have now chosen to present this data as supplementary material. These strategies ensure that the computational predictions presented are robust and reproducible.

(5) AlphaFold Predictions of Truncated MGAT1/CD73

Individual structures of MGAT1 and CD73 (truncated forms) were predicted by AlphaFold3. The root-mean-square deviation (RMSD) between the AlphaFold3-predicted proteins and their counterparts generated using homologous modelling software SwissModel were found

to be 0.6 Å and 0.2 Å, respectively. We have added a sentence to the manuscript to acknowledge the close agreement between AlphaFold3- and SwissModel-predicted structural models (page 12 line 279 to page 13 line 281): “The structures predicted by SwissModel for MGAT1 and CD73 were observed to be almost identically reproduced by AlphaFold3 (with respective RMSDs of 0.6 and 0.2Å).”

Furthermore, AlphaFold3 was also used to generate potential models for the complex formed by CD73 and MGAT1 using their truncated forms (see Supplementary Fig. 5c and caption for details). None of these models were compatible with experimental data on the identified interfacial residues. Therefore, we generated a ‘guided’ model for the complex, using ClusPro and experimental data, presented in Fig. 4e. The following excerpt from the text clarifies these issues (page 13 line 285-289): “AlphaFold3 was also employed to predict the binding interface between CD73 and MGAT1 (Supplementary Fig. 5a). When compared with experiments, ClusPro seems to provide a better prediction (No.1 in Supplementary Fig. 5a) than AlphaFold3 (No.2-6 in Supplementary Fig. 5a) where the interacting interface observed in co-IP position closely in ClusPro predicted model. ”

Supplementary Fig. 5a

(6) Can Truncated MGAT1/CD73 Maintain Their Native Structures and Conformations?

This is an excellent point. We checked them in two ways: by comparing the structures of the intact and truncated proteins, and by comparing the dynamics of the intact and truncated proteins:

Comparison of intact and truncated conformations: Based on AlphaFold3 predictions, the structure of the truncated MGAT1 (fragment of interest; residues 220-445) is almost identical to that of WT (intact) MGAT1. We further tested the stability of MGAT1 220-445 Δ 321-370, which yielded an RMSD of 0.249Å with respect to its counterpart in intact MGAT1. Likewise, CD73 1-224 structure was verified to be almost identical to that of CD73 WT, and CD73 Δ 79-128 differed from the WT by an RMSD 0.855Å which is typically within the resolution of protein structures.

Comparison of the conformational dynamics of the intact and truncated forms: We carried out, MD simulations up to 200 ns for the truncated proteins in explicit water to assess their stability as well as possible refolding/collapsing of flexible regions, in comparison to the dynamics of the corresponding regions in the intact proteins. Results showed that MGAT1 fragments were stable within an RMSD of 3.5Å with respect to the starting conformation. CD73 WT, CD73 1-224 and CD73 Δ 79-128, on the other hand, showed departures of up to 9Å from the initial state. Notably, this behavior was shared by the wildtype CD73 as well as its truncated fragments, pointing to the higher conformational flexibility of CD73 in general. Further examination of the initial and final conformations of CD73 WT, CD73 1-224 and CD73 Δ 79-128 confirmed that the change in conformations during the course of simulations maintained the overall fold, with major rearrangements originating from the reconfiguration of the flexible (loop) regions as well as the relative position of the two domains.

These results are presented in the Supplementary Material as well as the main text. The newly added paragraphs can be seen from page 13 line 306 to page 14 line 325 of the main text of

manuscript, “Since our data suggests that MGAT1 interaction with CD73 relies on the critical interfacial residues regions, it is worth validating if the truncated MGAT1 and CD73 maintain their native strictures and conformation. We examined them in two ways: 1. by comparing the structural alignment of the intact and truncated proteins, and 2. by comparing the dynamics of the intact and truncated proteins. In comparison of intact and truncated conformation, the structure of the truncated MGAT1 and CD73 (fragment of interest; residues 220-445 and 27-224, respectively) predicted by AlphaFold3 predictions is almost identical to that of WT (intact) MGAT1. For the smaller fragments, when superimposed on its wildtype counterpart, MGAT1 220-445 Δ 321-370 yielded an RMSD of 0.249Å and CD73 Δ 79-128 differed from the WT by an RMSD 0.855Å, which is typically within the resolution of protein structures. In the second way, we carried out, MD simulations up to 200 ns for the truncated proteins in explicit water to assess their stability as well as possible conformational change and refolding/collapsing of flexible regions, in comparison to the dynamics of the corresponding regions in the intact proteins. Results showed that MGAT1 fragments were stable within an RMSD of 3.5Å with respect to the starting conformation (Supplementary Fig. 5d). CD73 WT, CD73 1-224 and CD73 Δ 79-128, on the other hand, showed departures of up to 9Å from the initial state (Supplementary Fig. 5e). Notably, this behavior was shared by the wildtype CD73 as well as its truncated fragments, pointing to the higher conformational flexibility of CD73 in general. Further examination of the initial and final conformations of CD73 WT, CD73 1-224 and CD73 Δ 79-128 confirmed that the change in conformations during simulations maintained the overall fold, with major rearrangements originating from the reconfiguration of the flexible (loop) regions as well as the relative position of the two domains.” and the results can be viewed in the Supplementary Fig. 5d, e. Finally, we note that the stability of AlphaFold3 models in MD simulations is not necessarily a good metric for assessing the fragments’ fold and stability. However, the structural models and simulations, combined with experimental evidence showing that these fragments stabilize folded forms and

migrate normally on the gel, appear to be in support of the utility of the fragments for probing interfacial interactions.

Supplementary Fig. 5d, e

Supplementary Fig. 5d, e: **d** Different constructs of MGAT1 are predicted by AlphaFold3 and MD simulation was carried out to assess these predicted constructs stability. RMSD progression over time is plotted for MGAT1 WT (residue sequence from P101 to N445), MGAT1 220-445 (residue sequence from E220 to N445), and MGAT1 220-445Δ (residue sequence from E220 to N445 with the deletion from H321 to V370) α-carbon position during the 200ns MD simulation. The superposition of the first and last frame of each of these MD runs were shown for visualization of the actual structural fluctuation. **e** Similar to **d**, the following constructs of CD73 were also predicted by AlphaFold3 to assess the stability of the fragments CD73 WT (residue sequence from W27 to S549, PDB ID: 4H1S), CD73 27-224 (residue sequence from W27 to E224), and CD73 Δ (residue sequence from W27 to S549 with the deletion from L79 to I128). RMSD over time is plotted over 200ns for CD73 WT, CD73 27-224, and 70ns for CD73 Δ.

Question #2 - MGAT1 is the key enzymes in protein glycosylation pathway. Inhibiting its activity in vivo leads to dysregulation of various membrane receptors and transporters across all cell types. The authors should expand their discussion about the biological consequence of inhibiting MGAT1 activity using W-GTF01.

We thank reviewer for the suggestion. We agree that the critical role of MGAT1 in the N-glycosylation pathway indicates the potential for widespread biological consequences when its activity is inhibited. To address this, we have expanded our discussion to include the broader implications of MGAT1 inhibition and propose targeted strategies to mitigate associated toxicities.

The newly added information has been inserted to the manuscript from page 29 line 710 to page

30 line 725: “As a key enzyme in the N-glycosylation pathway, MGAT1 is critical for the proper folding, stability, and function of numerous glycoproteins, including membrane receptors and transporters. Inhibiting MGAT1 activity *in vivo* with compounds like W-GTF01 has demonstrated promising efficacy, particularly in immune-cold TNBC models, by promoting tumor regression through the revival of cytotoxic CD8⁺ T cell responses. However, MGAT1’s integral role in glycoprotein biosynthesis also raises concerns about systemic toxicity and off-target effects, as W-GTF01 binds to the UDP-binding domain shared by other GlcNAc-transferases. This could result in suboptimal glycosylation of essential proteins, impairing cellular signaling, and receptor-mediated interactions, ultimately leading to adverse physiological outcomes. To mitigate these risks, strategies such as antibody-drug conjugates (ADCs) and combination therapies that enhance selectivity and minimize off-target effects are being actively explored. Furthermore, comprehensive characterization of W-GTF01, including its toxicity, selectivity, and pharmacokinetics, is critical for therapeutic development. Future efforts should also focus on identifying biomarkers to monitor potential side effects and guide therapeutic interventions. These approaches will enable the development of more effective and safer MGAT1-targeting therapies, leveraging their therapeutic potential while minimizing unwanted biological consequences. ”

Reviewer #4, Specific Comments:

Question #1 - *Figure 3A, there are so many other bands darker than the CD73 band in the right lane (MGAT1-OE). Is there any other glycotransferase or immune-response related protein that co-precipitated with MGAT1? The quality of MS/MS spectra can be improved. Did you only observe one peptide from CD73 and one peptide from MGAT1? How many replicates did you repeat?*

Thank you for your comments. We appreciate the opportunity to elaborate on the mass spectrometry (MS) data and the identification of MGAT1-associated proteins. Indeed, **Fig. 3a**

reveals several bands in the right lane (MGAT1-OE), indicating the presence of additional proteins that may co-precipitate with MGAT1. To identify significant binding partners, we applied stringent criteria: (1) at least five peptides must be detected in the experimental group, and (2) the number of peptides in the test group must be at least twice that of the control group. Using these parameters, MGAT1 (the bait for purification) yielded 53 total peptides, including 14 unique peptides. Among these, CD73 emerged as one of the top candidates with 6 total peptides and 3 unique peptides and none of which were detected in the control group. In addition, although other immune-related proteins, such as members of the HLA family, were detected, their co-presence in the control group suggests non-specific binding. The identification of CD73 as an MGAT1 interactor also aligns with our earlier findings (Fig. 1), where MGAT1 strongly correlated with immune regulator expression. This supports a mechanistic link between MGAT1 and CD73 in immune modulation through glycosylation. Given CD73's role in facilitating immune evasion by promoting adenosine production and suppressing anti-tumor immune responses, its interaction with MGAT1 further underscores its biological significance.

Due to space constraints in the figure panels, we highlighted CD73 as the most relevant candidate to advance the manuscript's mechanistic framework. To ensure transparency and accessibility, we have uploaded the complete mass spectrometry data online and **included a full list of potential MGAT1-associated proteins in the supplementary materials**. To further address reviewer's concern, we have also attached additional spectrum in the **Supplementary Fig. 4a**.

Supplementary Fig. 4a

Supplementary Fig. 4a: a Representative peptides of CD73 and MGAT1 discovered by mass spectrometry.

Question #2 - *MGAT1 facilitates N-glycosylation across a range of glycosylated membrane receptors and transporters. Did you observe any other highly glycosylated membrane receptors involved in immune check point and antitumor immunity in the TAP experiment? For example, CD39 is also highly glycosylated and it is in the same pathway with CD73.*

Thank you for your insightful comment regarding other glycosylated immune checkpoint proteins, such as CD39, which operates within the same adenosine production pathway as CD73. Indeed, CD39 is a highly glycosylated protein and plays a critical role in immune modulation. However, based on our TAP experiment, CD73 was the only specific immune checkpoint protein identified as an MGAT1 interactor. Other immune-related proteins, such as members of the HLA family, were also detected, but their co-presence in the control group indicates non-specific binding.

To further explore whether MGAT1 regulates CD39 membrane distribution, we conducted additional experiments using MGAT1-WT, MGAT1-OE, and MGAT1-KD stable cell lines. The flow cytometry analysis revealed that, unlike CD73, the membrane distribution of CD39 was not dramatically influenced by MGAT1 expression like CD73. This suggests that MGAT1 does not significantly regulate CD39 glycosylation, or that alternative pathways may govern its glycosylation status. These findings highlight the specificity of the MGAT1-CD73 interaction and its unique role in regulating immune evasion via glycosylation. The new data and description have been included in the revised manuscript from page 17 line 394-398: "To examine whether MGAT1 also influences the membrane distribution of other critical regulators in adenosine production, we measured membrane CD39 in MGAT1-WT, MGAT1-OE, and MGAT1-KD cell lines with flow cytometry and the results (Supplementary Fig. 7h) showed no significant alteration in CD39 membrane distribution across these conditions, indicating that MGAT1 does not play a prominent role in regulating CD39 glycosylation."

Supplementary Fig. 7h

Supplementary Fig. 7h: h Membrane CD39 was measured with flow cytometry in MDA-MB468 and MDA-MB231 cells with MGAT1 OE/KD. Representative analyses are shown.

Question #3 - *It is essential to perform a glycoproteomics experiment of determine the glycan types in CD73, and confirm that its glycosylation is regulated by MGAT-KO and OE.*

We thank reviewer's insightful suggestion! We agree that a glycoproteomic analysis would provide a comprehensive characterization of CD73 glycosylation, enabling us to identify specific glycan structures and their abundances. However, despite multiple attempts, we encountered technical challenges in optimizing the glycomics workflow. Additionally, the primary author of this manuscript has graduated and transitioned to a new position, which limits our ability to conduct further glycoproteomic experiments at this time.

To address this limitation, we performed complementary experiments to elucidate the role of MGAT1 in regulating CD73 glycosylation and its functional consequences. Specifically, our N-glycan trimming experiments using Endo H and PNGase F with three types of TNBC cell lines, MDA-MB231, MDA-MB468 and 4T1 (Supplementary Fig. 6a) demonstrated that the complex type of glycan is the most predominant type N-glycan on CD73 since the PNGase F treatment lead to a much significant drop in CD73 molecular weight than that caused by Endo H. To address if MGAT1 significantly influences the glycan complexity of CD73, our Endo H and PNGase F treatment on MGAT-WR/OE-KD stable lines (Fig. 5b) demonstrates that, in MGAT1-

overexpressing cells, CD73 predominantly exhibited complex glycans, as evidenced by reduced sensitivity to Endo H treatment. In contrast, the lower molecular band of CD73 in MGAT1-KD stable lines were highly sensitive to Endo H trimming, indicating the knockdown of MGAT1 leads to insufficient glycosylation of CD73 and cause increased high-mannose or hybrid forms of glycan .

Further functional assays revealed that MGAT1-mediated glycosylation is essential for CD73 dimerization and membrane localization (Fig. 5c-l, Supplementary Fig. 6f and Supplementary Fig. 7a-e), directly linking MGAT1 activity to CD73-mediated immune suppression. We have included all the newly generated data and related information in the revised manuscript from page 14 line 332 to page 15 line 341 “As described above, we initially engineered MGAT1 OE and KD in MDA-MB468 and MDA-MB231 cells. As shown in Fig. 5a, while a minor change in CD73 abundance in response to MGAT1 OE was observed, reduced glycosylation of CD73 (running smaller molecular mass ~60 kDa) was clearly detected when MGAT1 was knocked down, confirming the aforementioned CD73 glycosylation catalyzed by MGAT1. To investigate the predominantly type of glycans on CD73, we further tested the effect of endoglycosidase H (specifically cleaving high mannose and hybrid N-glycan) and PNGase F (an N-glycosidase removing all N-linked glycan) on CD73 in TNBC cell lines. The CD73 band shift indicates that complex type N-glycan is the major type of asparagine-linked carbohydrate attached to CD73 (Supplementary Fig. 6a), but without sufficient MGAT1, a fraction of less glycosylated CD73 will be covered with high mannose and hybrid N-glycan (Fig. 5b). “

Supplementary Fig. 6a

Supplementary Fig. 6a: a The whole cell lysates of MDA-MB231, MDA-MB468 and 4T1 cells were treated with endoglycosidase H and PNGase F and the CD73 protein levels and molecular sizes were determined by immunoblotting.

Fig. 5b: b The whole cell lysates of MDA-MB468 cells with MGAT1 OE/KD were treated with endoglycosidase H and PNGase F and the CD73 protein levels and molecular sizes were determined by immunoblotting.

Fig. 5f & Supplementary Fig. 6f

Fig. 5f: f CD73 dimerization in MDA-MB468 cells with MGAT1 OE/KD was determined by immunoblotting with semi-native gel.
Supplementary Fig. 6f: f CD73 dimerization in 4T1 cells with MGAT1 OE/KD was determined by immunoblotting with semi-native gel.

Question #4 - *It would be beneficial to clarify the reason for using two different software, Cluspro and ZDOCK for the prediction of MGAT1-CD73 complex.*

Thank you for your question regarding our rationale of using two different software tools, ClusPro and ZDOCK. Both ClusPro and ZDOCK are protein-protein rigid-body docking algorithms that utilize Fast Fourier Transform (FFT) for rapid docking and energy calculations. ZDOCK was developed earlier and features a straightforward pipeline, ranking docked poses based on scores that reflect binding affinity and geometric complementarity. In contrast, ClusPro, developed later, incorporates a more advanced clustering process to identify biologically relevant docked poses. ClusPro has been demonstrated to yield highly accurate results in many challenging cases, especially for large complexes. For pure protein-protein binding prediction, ClusPro provides greater accuracy by allowing slight overlaps between proteins, thereby mimicking protein conformational flexibility. This capability makes ClusPro particularly well-suited for refining docking results and identifying more biologically relevant complexes. However, ClusPro cannot predict interactions involving glycosylated proteins. In contrast, ZDOCK can effectively handle interactions with glycosylated proteins. Therefore, we used Cluspro for pure protein-protein interactions, while we utilized ZDOCK for interactions involving MGAT1 and glycosylated CD73.

By combining these tools, we were able to leverage the strengths of ZDOCK for broad initial exploration and ClusPro for precise refinement, ensuring robust and reliable predictions for the MGAT1-CD73 complex.

Question #5 - *Figure 4F, is this the best RMSD from one MD simulation or averaged RMSD from several MD simulations?*

We thank reviewer for the insightful question. The RMSD data initially presented in Figure 4F were derived from a single molecular dynamics (MD) simulation to assess the structural stability of the CD73-MGAT1 complex. To address reproducibility and provide a more comprehensive analysis, we have extended our simulations and performed multiple independent runs. Specifically, we conducted five 200-nanosecond MD simulations to ensure consistent results and validate the stability of the complex. We have now updated Fig. 4f to include the RMSD results from all five runs, demonstrating the overall reproducibility and stability of the CD73-MGAT1 complex. Additionally, Fig. 4g has been revised to present the results from a representative long MD run, providing a clearer illustration of the dynamic behavior of the complex over an extended timescale. The revised figures and corresponding descriptions have been incorporated into the updated manuscript in page 13 line 291-297 “In order to assess the stability of the predicted complex conformation, we performed five runs of 200 ns MD simulations for the top-ranked cluster predicted by ClusPro. The root-mean-square deviations (RMSDs) of the MGAT1-CD73 complexes were evaluated by structurally aligning the simulated complexes with the initial ClusPro-predicted model. A RMSD profile over 200 ns simulation time indicated a stable binding of MGAT1 and CD73 in 4 out of 5 runs (Fig. 4f). The result from Fig. 4g illustrates the time evolution of the most persistent inter-residue interactions at the interface between MGAT1 and CD73 over the 200 ns time frame of the MD simulations.”

Updated Fig. 4f, g

Updated Fig. 4f, g: f, g The time evolution of the RMSDs of the MGAT1-CD73 complex in five 200-ns MD simulations of the complex formed with MGAT1-CD73 using the start points in (e) is shown in (f). The RMSD was evaluated after structurally aligning the conformers observed during MD trajectories with respect to the initial ClusPro-predicted complex. The corresponding time evolution of residue-residue interactions between MGAT1 and CD73 residues is shown in (g), and regions shaded in gray refer to time intervals during which the indicated residue pairs made interfacial contacts.

Question #6 - Line 304, how do you know all four Asn residues can be glycosylated by MGAT1?

The proximity of two protein in space does not necessarily imply a catalytic reaction. Additionally, why does ZDOCK predict four different complex structure? Does this mean that all four structures are plausible or that none of them are significant?

We appreciate reviewer for providing the opportunity to address this important point. First, we acknowledge that the proximity of MGAT1 and CD73 in space, as predicted by docking, does not necessarily confirm catalytic activity or imply that all four Asn residues can be glycosylated. The docking results were presented as an illustration of potential glycosylation mechanisms, illustrating the spatial alignment that could enable glycosylation at these sites. However, given the uncertainties associated with these computational predictions, we have moved the original **Fig. 4h** to the Supplementary Materials to maintain clarity and focus.

To further validate whether all four Asn residues on CD73 are regulated by MGAT1, we conducted additional experiments. Specifically, we generated a triple-mutant CD73 construct, where three of the four N-glycosylation sites were mutated to glutamine (Q), mimicking a condition where only one site could be glycosylated. The triple-mutant CD73 was transfected into wild-type (WT) or MGAT1 knockdown (MGAT1-KD) 293T cells, and Western blotting was used to analyze the molecular size of CD73 under these conditions. Our results demonstrated MGAT1 knockdown significantly decreased the molecular weight (band height) of CD73, suggesting that MGAT1 indeed plays a critical role in catalyzing N-glycan addition on CD73. When three of the four Asn sites were mutated (triple-mutant CD73), the band height of CD73 was reduced compared to WT-CD73, indicating a reduction in glycosylation. However, the band height in the triple-mutant was still higher than in the completely glycosylation-deficient 4NQ-CD73 mutant, suggesting that each of the four Asn sites on CD73 can potentially be glycosylated by MGAT1. These findings provide experimental evidence that MGAT1 regulates the glycosylation of all four N-glycan sites on CD73. This new data has been included in the revised manuscript from page 15, line 341 to line 348: "CD73 owns four N-glycosylation motifs including ⁵³NAS, ³¹¹NSS, ³³³NYS, and ⁴⁰³NGT⁵¹. To

investigate whether all four N-glycan sites on CD73 are regulated by MGAT1, we transfected the triple-mutant CD73 construct into WT and MGAT1-KD HEK-293T cells and analyzed CD73 molecular size via Western blotting. The results demonstrated that glycosylation at all four sites requires the participation of MGAT1, as MGAT1 knockdown significantly reduced the CD73 glycosylation (Supplementary Fig. 6b). Additionally, the triple-mutant CD73 retained partial glycosylation, further supporting the hypothesis that each of the four Asn sites on CD73 can independently be glycosylated by MGAT1."

Supplementary Fig. 6b

Supplementary Fig. 6b: b The protein molecular size of CD73 carried indicated triple-mutation at N-glycan sites in MGAT1-WT and KD 293T cells were determined by immunoblotting.

Question #7 - Figure 5E, there appear to be two contradictory results in a single gel. The middle lines (MGAT1-OE enhances CD73 dimerization) suggests that CD73 carries oligomannose type N-glycans and the overexpression of MGAT1 promotes the formation of complex N-glycans in CD73. However, the right lanes (MGAT1-KD suppresses CD73 dimerization) indicate CD73 carries complex-type N-glycans, and that MGAT1-KD results in oligomannose type N-glycan in CD73.

We thank the reviewer for the insightful comment and the opportunity to clarify our findings. MGAT1 plays a pivotal role in the N-glycosylation pathway, catalyzing the critical step in converting oligomannose to hybrid glycans, which are subsequently processed by downstream

glycosyltransferases to form complex N-glycans. Thus, MGAT1 indirectly facilitates the formation of complex glycans by providing substrates for downstream enzymes, but its overexpression does not result in the accumulation of oligomannose glycans. Instead, MGAT1 overexpression promotes the overall progression of glycosylation, increasing complex glycan abundance.

Under MGAT1 overexpression, CD73 predominantly carries complex-type glycans, as indicated by the high molecular weight (~70 kDa) band that is largely resistant to Endo H digestion (Fig. 5b). These complex glycans enable CD73 to efficiently form dimers, as shown in the semi-native gel analysis (Fig. 5f). This result reflects the combined action of MGAT1 and downstream glycosyltransferases, facilitating the maturation of glycan chains to complex forms. In contrast, MGAT1 knockdown significantly impairs CD73 glycosylation, resulting in a predominance of oligomannose glycans (~60 kDa). While a small fraction of CD73 retains complex glycans (~70 kDa), this is drastically reduced compared to MGAT1-WT or MGAT1-OE conditions. The impaired glycosylation is corroborated by the increased sensitivity of CD73 to Endo H digestion (Fig. 5b), which shifts the molecular weight band significantly. This insufficient glycosylation undermines the ability of CD73 to form dimers, as observed in Fig. 5f.

Taken together, the overexpression of MGAT1 does not lead to an accumulation of oligomannose glycans but rather drives their conversion into complex glycans, supporting efficient CD73 dimerization. Conversely, MGAT1 knockdown results in a glycosylation bottleneck, leaving CD73 predominantly modified with oligomannose glycans, which compromises its ability to dimerize. These findings align with MGAT1's established role as an initiator of hybrid and complex glycan synthesis, ensuring the proper maturation of glycan chains. We have clarified these findings in the revised manuscript on page 14 line 332 to page 15 line 341 as follows: "As described above, we initially engineered MGAT1 OE and KD in MDA-MB468 and MDA-MB231 cells. As shown in Fig. 5a, while a minor change in CD73 abundance in response to MGAT1 OE was observed, reduced glycosylation of CD73 (running smaller molecular mass ~60 kDa) was clearly detected when MGAT1 was knocked down, confirming the aforementioned CD73

glycosylation catalyzed by MGAT1. To investigate the predominantly type of glycans on CD73, we further tested the effect of endoglycosidase H (specifically cleaving high mannose and hybrid N-glycan) and PNGase F (an N-glycosidase removing all N-linked glycan) on CD73 in TNBC cell lines. The CD73 band shift indicates that complex type N-glycan is the major type of asparagine-linked carbohydrate attached to CD73 (Supplementary Fig. 6a), but without sufficient MGAT1, a fraction of less glycosylated CD73 will be covered with high mannose and hybrid N-glycan (Fig. 5b).” and page 15 line 359-page 16 line 363: “In response to MGAT1-mediated CD73 modification, we observed a dramatic change in CD73 dimerization. We initially confirmed the dimerization of CD73 when the CD73 complex was stabilized in the presence of a crosslinking agent (Fig. 5e). The dimerization was further confirmed in several TNBC cell lines by running semi-native gels (Supplementary Fig. 6e) and the knockdown of MGAT1 dramatically decreased the CD73 dimerization (Fig. 5f and Supplementary Fig. 6f). ”

Fig. 5b

Fig. 5b: b The whole cell lysates of MDA-MB468 cells with MGAT1 OE/KD were treated with endoglycosidase H and PNGase F and the CD73 protein levels and molecular sizes were determined by immunoblotting.

Fig. 5f & Supplementary Fig. 6f

Fig. 5f: f CD73 dimerization in MDA-MB468 cells with MGAT1 WT/OE/KD was determined by immunoblotting with semi-native gel.

Supplementary Fig. 6f: f CD73 dimerization in 4T1 cells with MGAT1 WT/OE/KD was determined by immunoblotting with semi-native gel.

Question #8 - *Can two well-known tool compounds, namely kifunensine and swainsonine also inhibit CD73 dimerization?*

We appreciate the reviewer's suggestion to further investigate the role of branched N-glycans produced by MGAT1 in CD73 dimerization and adenosine production. To address this, we performed additional experiments using TNBC cell lines (MDA-MB231 and MDA-MB468) treated with kifunensin and swainsonine, two well-established glycosidase inhibitors that effectively disrupt glycoprotein processing. The effects of these inhibitors on CD73 dimerization and adenosine production were evaluated (Supplementary Fig. 6h, i), and the related data and description have been included in the revised manuscript (page 16 line 366-370): "To further determine if branched N-glycans produced from MGAT1 could directly affect CD73 dimerization and adenosine production, we treated both cell lines with kifunensin and swainsonine to ensure efficient inhibition of N-glycan branching and the results demonstrated the critical role of branched N-glycans in stabilizing CD73 dimerization and adenosine production (Supplementary Fig.6i, j)."

Supplementary Fig. 6i

Supplementary Fig. 6i: i MDA-MB231 and MDA-MB468 cells were treated with 20 μ M kifunensine or swainsonine for 48 hours and the CD73 dimerization were determined by immunoblotting with semi-native gel.

Supplementary Fig. 6j

Supplementary Fig. 6j: j MDA-MB231 and MDA-MB468 cells were treated with 20 μ M kifunensine or swainsonine for 48 hours and the adenosine concentration in the medium was measured with Adenosine Assay from Cell Biolabs.

Reviewer #5, General Comments:

The study focuses on a glycosyltransferase, MGAT1, which the authors identify as a regulator of CD73 glycosylation. This regulation drives the transport and dimerization of CD73 at the cell surface, enabling its activity and the generation of adenosine, which leads to immunosuppression. Notably, the study includes the development of an inhibitor that disrupts this interaction, preventing MGAT1-CD73-mediated immunosuppression. This is a key highlight of the research; however, data supporting the inhibitor's effects should be further validated. Overall, the study is comprehensive, encompassing molecular biology, biochemistry, as well as preclinical and clinical validations. While the data is compelling, additional analyses and revisited claims are needed before it can be considered for publication.

We thank the reviewer for the assessment of our work. We are pleased that the reviewer recognizes the key strengths of our study, including its comprehensive approach spanning molecular biology, biochemistry, and preclinical validations, as well as its focus on MGAT1 as a critical regulator of CD73 glycosylation and its implications for immunosuppression. In response to the reviewer's constructive comments, we have incorporated new experimental results and additional analyses validating the inhibitor's mechanism of action and its impact on MGAT1-mediated CD73 glycosylation and therapeutic relevance.

Reviewer #5, Specific Comments:

Question #1 - *In the introduction the authors should contextualize better TNBC as one of the most aggressive subtypes due to lack of therapies (in general, not only immunotherapy). Indeed, HER2 BC is more aggressive but has efficient therapies reducing the mortality and aggressiveness of this subtype. Authors should also mention that immunotherapy has only been approved in TNBC, but not in the other subtypes since these are immune cold tumors. TNBCs are not as cold as the rest of BC tumors.*

We appreciate the reviewer's suggestion and have carefully revised the introduction to better contextualize TNBC and its therapeutic challenges. The newly added text has been incorporated into the revised manuscript from page 4 line 64-77: "Despite the benefits of immunotherapy in treating certain types of cancers, the variability in patient response still remains¹⁻³. Triple-negative breast cancer (TNBC) is among the most aggressive subtypes of breast cancer, primarily due to the lack of effective targeted therapies and limited treatment options beyond chemotherapy. Importantly, TNBC is the only breast cancer subtype for which immune checkpoint inhibitors⁴⁻⁶. However, even in TNBC, many patients exhibit resistance to immunotherapy, driven by mechanisms of immune evasion and tumor-mediated immunosuppression⁷⁻⁹. These factors underscore the critical need for innovative therapeutic strategies to enhance the efficacy of immunotherapy in TNBC and address the persistent unmet clinical needs of this patient population. Our studies using multiomics analyses and experimental validation to identify novel targets for improving anticancer immune responses have drawn our attention to MGAT1."

Question #2 - *Authors should discuss and consider in their statements why MGAT1 is more expressed in TNBC rather ER+BC which is much colder than TNBC. It could help to provide a similar analysis as in Fig. 1b including all type of BC patients.*

Thank you for raising this important question about why MGAT1 expression is significantly higher in TNBC compared to ER+ breast cancer, which is often considered more "immune-cold." We have expanded our analysis (Supplementary Fig. 1b-e) to address this observation and provide a comprehensive explanation below:

First, in TNBC, we observed significantly elevated expression of immune regulators such as CD73 and PD-L1. These immune-related factors are known to exert inherent immunosuppressive effects on the tumor microenvironment. Furthermore, their stability and activity are enhanced by MGAT1-mediated glycosylation, which reinforces immune suppression. Specifically, the elevated expression of MGAT1 in TNBC facilitates the glycosylation and functional optimization of these

immune checkpoint proteins, contributing to the robust immune evasion observed in TNBC. This suggests that the pronounced immunosuppressive microenvironment in TNBC is closely linked to the higher expression of MGAT1.

Second, our additional analysis of ER+ or Her2+ breast cancer revealed a different immune profile. While ER+ or Her2+ tumors are often considered "immune-cold," the overall expression of immune regulatory factors in this subtype was found to be relatively low. A small subset of ER+ patients exhibited elevated immune-related gene expression, similar to certain immunosuppressive subpopulations observed in TNBC. However, glycosylation-related pathways, including MGAT1 and its related glycosylation enzymes, did not show significant upregulation in ER+ breast cancer. Instead, the immune coldness in ER+ or Her2+ tumors is likely driven by alternative mechanisms, such as reduced tumor-infiltrating lymphocytes (TILs), impaired antigen-presenting cell function, or diminished tumor antigen expression, rather than glycosylation-dependent pathways.

In addition, we have included the newly generated results in the revised manuscript from page 7 line 137-145: "To further validate these findings, we conducted the similar analysis to other breast cancer subtypes including LumA, LumB, Her2+, and Normal-like. As shown in **Supplementary Fig. 1b-e**, the data revealed that within these non-TNBC subtypes, a small subset of patients exhibited increased expression of immune-related genes, resembling the immune-cold subpopulation observed in TNBC. Interestingly, these samples also demonstrated significant enrichment in pathways associated with the negative regulation of the immune system, indicating the presence of a similar immunosuppressive environment. However, when analyzing glycosylation and glycan processing pathways, we found no significant differences compared to their main groups within the non-TNBC cohort (**Supplementary Fig. 1b-e**). "

Supplementary Fig. 1b-e

Supplementary Fig. 1b-e: b-e Distribution of single sample GSEA scores of LumA (**b**), LumB (**c**), Her2+ (**d**), and Normal-like (**e**) separated by immune subgroup, using gene sets involved in immune functions and glycosylation. P-values are from Wilcoxon tests. * $p < 0.05$, ** $p < 0.01$, *** $p < 0.001$, and **** $p < 0.0001$.

Question #3 - In fig.3 please show all candidates obtained with the MS data alongside CD73. We appreciate the importance of providing comprehensive insights into the mass spectrometry data. To identify MGAT1-associated proteins, we employed tandem affinity purification (TAP) combined with MS. This approach allowed us to systematically and unbiasedly identify MGAT1-

associated protein complexes. Candidate interactors were prioritized based on their enrichment scores, reproducibility across biological replicates, and functional relevance to glycosylation or immune-related pathways. From the MS data, we identified several proteins associated with MGAT1. Among these, CD73 emerged as one of the top candidates. CD73 is known to facilitate immune evasion by promoting adenosine production and suppressing anti-tumor immune responses. The identification of CD73 as an MGAT1 interactor aligns with our earlier data in Fig. 1 where MGAT1 is highly correlated with immune regulator protein expression.

Due to space constraints within the figure panels, we highlighted CD73 as the most relevant candidate to support the mechanistic framework of this manuscript. However, to ensure a comprehensive view of the data, we have included the complete list of potential binding partners identified through MS, in the supplementary materials.

Question #4 - *In order to validate that MGAT1 influence adenosine synthesis through its interaction with CD73. MGAT1 OE and KD experiments of Fig.3 should be also done in CD73-KD cells.*

We appreciate the reviewer's suggestion to investigate MGAT1 overexpression (OE) and knockdown (KD) in CD73-KD stable lines to validate whether MGAT1 influences adenosine synthesis through its interaction with CD73. Based on our previous findings (Fu et al., 2023), CD73 is critical in regulating adenosine production, with its knockdown significantly reducing adenosine levels. To directly address this question, we generated CD73-KD stable lines and subsequently engineered MGAT1-OE and MGAT1-KD within these models. Adenosine production was quantified under these conditions. This new experimental data has been incorporated into the revised manuscript on page 11 line 246-251: "To investigate the dependency of MGAT1's role in adenosine synthesis on CD73, we engineered MGAT1 overexpression and knockdown in CD73-KD stable lines. The results revealed that, in the absence of CD73, MGAT1 overexpression or knockdown did not affect adenosine levels (Supplementary Fig. 4e). These

findings demonstrate that MGAT1's influence on adenosine synthesis is mediated specifically through its regulation of CD73."

Data from previous publication (Fu et al., Sci Adv. 2023. doi: 10.1126/sciadv.add6626.)

Figure 2I: (I) Extracellular adenosine levels in MDA-MB468-shCD73, MDA-MB231-shCD73, and HS578T- shCD73 cells.

Supplementary Fig. 4e

Supplementary Fig. 4e: e Adenosine levels were determined in CD73-KD MDA-MB231 and MDA-MB468 breast cancer cells with MGAT1 OE or MGAT1 KD. **p, ***p, ****p<0.0001. Data (mean ± SEM) are representative of at least three independent experiments.

Question #5 - *In addition, all functional experiments should include this condition of CD73-KD to validate that MGAT1 is mediating all the claimed effects through CD73 or not. This is partially addressed with CD73-4NQ in in vivo experiments, however its own CD73-4NQ control is missing.*

We thank the reviewer for their insightful comment. We agree that including comprehensive controls, such as the CD73-KD condition, would provide additional clarity. However, our study specifically focuses on the crucial role of MGAT1-mediated glycosylation of CD73 in driving its dimerization and membrane localization, which are essential for CD73-mediated enzymatic activity and immune evasion. While CD73-KD serves as a valuable control to demonstrate the dependency of MGAT1 on CD73, it does not directly address the biochemical impact of MGAT1-mediated glycosylation on CD73's functional modifications.

To address this, we confirmed the biochemical role of MGAT1-mediated glycosylation in regulating CD73 dimerization by comparing WT-CD73, 4NQ-CD73 (a glycosylation-deficient mutant), and dimer-deficient CD73 under experimental conditions (Fig. 5h, I and Supplementary Fig. 7a-e). Additionally, we validated the importance of the MGAT1-CD73 interaction for tumor growth and immune evasion *in vivo* using CD73-4NQ as a model (Fig. 7m-o and Supplementary Fig. 10g-k).

Taken together, our comprehensive *in vitro* and *in vivo* data strongly support the conclusion that MGAT1-catalyzed glycosylation regulates CD73's membrane translocation, dimerization, and immune suppressive functions. While the inclusion of CD73-KD as an additional control would further complement our findings, we believe the evidence presented sufficiently demonstrates the critical role of MGAT1-mediated glycosylation in CD73-dependent adenosine-driven tumor immune suppression.

Question #6 - *Importantly, the effect of the inhibitor should be shown alongside the CD73-KD conditions to claim that is mediated to CD73 and not by other MGAT1 effects or by other targets of the designed inhibitor (i.e. teste whether it also affects Bcl-2,...). Indeed, it would be strongly*

recommended to perform RNA-seq of the treated cells with the inhibitor to have a full picture of transcriptomic changes mediated by the inhibitor.

We thank reviewer's comment and suggestion. In our initial in vitro screening, TW-37, a previously identified Bcl-2 family protein inhibitor, was found to effectively inhibit MGAT1 enzymatic activity. Structural docking analysis revealed that TW-37 binds to the UDP-binding domain of MGAT1, thereby inhibiting its function. Based on this discovery, we performed virtual screening and designed improved compounds with higher specificity for MGAT1, reduced off-target effects, and enhanced pharmacological properties. These newly identified compounds, including No.2, No.8 (W-GTF01), and No.9, were optimized to prioritize specificity toward MGAT1 and minimize off-target effects. To validate the specificity of these compounds, we assessed Bcl-2 abundance following treatment with No.2, No.8, and No.9 in MDA-MB231 cells using Western blot analysis. These findings have been incorporated into the revised manuscript **on page 20, line 475 to line 478**: "Furthermore, to determine if No.2, No.8 (W-GTF01), and No.9 exhibit off-target effects against Bcl-2, we measured Bcl-2 abundance following treatment in MDA-MB231 and MDA-MB468 cells. The results confirmed that these compounds do not affect Bcl-2 expression (Supplementary Fig. 9h), further supporting their specificity toward MGAT1."

Additionally, we recognize the importance of incorporating CD73-KD as a condition to validate that the inhibitor's effects are mediated through CD73. To address this, we established CD73-KD stable lines in MDA-MB468 cells and repeated the coculture with immune cells under W-GTF01 treatment. These findings have been added to the revised manuscript on page 20 line 471-475: "To investigate whether W-GTF01 directly affects the MGAT1-CD73 axis, we established CD73-KD stable lines in MDA-MB468 cells and repeated the coculture experiment with immune cells under W-GTF01 treatment. The data demonstrated that the addition of W-GTF01 did not present further strengthened efficacy to immune cells in the absence of CD73 (Supplementary Fig. 9g)."

Finally, we are collaborating with synthetic chemists to optimize W-GTF01 for enhanced potency and selectivity. As part of our ongoing efforts, we plan to conduct RNA-seq experiments

in the next phase of this study to comprehensively evaluate the transcriptomic changes induced by these inhibitors, providing a global understanding of their specificity and biological effects.

Supplementary Fig. 9h

h

Supplemental Fig. 9h: h The protein levels of Bcl-2 in MDA-MB468 and MDA-MB231 after treatment of DMSO, compound No.2, No.8 and No.9 at were determined by immunoblotting.

Supplementary Fig. 9g

g

Supplementary Fig. 9g: g The survival of WT or CD73-KD MDA-MB468 cells in coculture with pre-activated PBMCs and No.8 was monitored by time lapse image-based quantification. The data are presented as mean \pm SEM from three replicates of a representative experiment.

Question #7 - *CX3CR1* reflects a transitory T cell effector state and *CD101* a differentiated exhausted state, thus authors should show percentages of these combined populations, since the increase of *CX3CR1*⁺ *CD8* T-cells do not support their conclusions.

We appreciate the reviewer's valuable comment. As suggested, we have replaced the original panel with updated data showing the percentages of *PD-1*⁺*TOX*⁺ and *CX3CR1*⁺*CD101*⁺ *CD8*⁺ T cell populations in Fig. 7f. This approach provides a more accurate representation of the differentiation and exhaustion states of *CD8*⁺ T cells.

The updated results and associated descriptions have been incorporated into the revised manuscript as follows: (page 21 line 517 to line 518): "Furthermore, the fractions of *PD-1*⁺*TOX*⁺ and *CX3CR1*⁺*CD101*⁺ *CD8*⁺ cells were significantly increased in *MGAT1* OE tumors (Fig. 7f)."

Updated Fig. 7f

f

Updated Fig. 7f: f Comparison of the percentage of *PD-1*⁺*TOX*⁺ *CD8*⁺ and *CX3CR1*⁺*CD101*⁺ *CD8*⁺ T cells among live *CD45*⁺ between 4T1 control tumors and 4T1 *MGAT1* OE tumors.

Question #8 - *The immune-checkpoint experiments in fig.7 should include survival events. In addition, waterfall plots would help the interpretation of the data.*

We appreciate the reviewer’s suggestion to include survival data and waterfall plots in the immune-checkpoint experiments presented in Fig. 7. To address this, we have revisited our data in Fig. 7p (previous Fig. 7q) and the survival curves have been included in the revised Fig. 7q, and corresponding descriptions have been added to the manuscript (page 22 line 550 to page 23 line 553): “In addition, we also observed that the combination of W-GTF01 and anti-PD-L1 significantly improves survival outcomes compared to either treatment alone (Fig. 7q), reinforcing the conclusion that targeting MGAT1 enhances the efficacy of immune checkpoint blockade.” Fig.

7q

Fig. 7q: q 4T1-hPD-L1 cells, where the endogenous mouse PD-L1 was replaced with its human counterpart, were orthotopically injected into the left fourth mammary fat pad and allowed to grow to ~100 mm³, followed by injection of W-GTF01 (10 mg/kg, i.p.) two times/week and PD-L1 antibody durvalumab (10 mg/kg, i.p.) 3 times/week. PBS and IgG were used in the control groups. The tumor growth (p) and the survival (q) of the mice were plotted. *p<0.05, **p<0.01, ***p<0.001, and ****p<0.0001. For tumor growth statistical analysis, two-way ANOVAs followed by Tukey’s multiple comparison tests were performed. Data (means ± SEM) are representative of at least two independent experiments with 5-10 independently analyzed mice per group.

Question #9 - *The immune-checkpoint experiments in fig.7 should include gain-and-loss of function conditions with MGAT1-KD/OE and CD73-4NQ (or KD).*

We thank the reviewer for the suggestion to incorporate gain- and loss-of-function experiments with MGAT1-KD/OE and CD73-4NQ into the immune-checkpoint studies presented in Figure 7. To address this, we performed additional experiments to comprehensively evaluate the roles of MGAT1 and CD73 in modulating immune checkpoint blockade responses. The new data and related descriptions have been incorporated into the revised manuscript from page 23 line 560 to

line 566 “To further validate the role of the MGAT1-CD73 axis in immune checkpoint blockade responses, we conducted gain- and loss-of-function experiments in MGAT1-KD, MGAT1-OE, and CD73-4NQ tumor models (Supplementary Fig. 11a-f). MGAT1 overexpression impaired the anti-PD-L1 ICB efficacy (Supplementary Fig. 11a, b). Conversely, the MGAT1 knockdown significantly enhanced the efficacy of anti-PD-L1 therapy, characterized by reduced tumor progression (Supplementary Fig. 11c, d). Similarly, the CD73-4NQ tumors exhibited significantly tumor growth inhibition under anti-PD-L1 therapy compared to WT-CD73 tumors (Supplementary Fig. 11e, f). These findings highlight the critical role of MGAT1-mediated CD73 glycosylation in driving immune suppression and resistance to ICB therapy.”

Supplementary Fig. 11a-f

Supplementary Fig. 11: The gain- and loss-of-function experiments with MGAT1-KD/OE and CD73-4NQ to validate the critical role of MGAT1-mediated CD73 glycosylation in the resistance to ICB therapy. a-f 4T1-hPDL1-MGAT1-OE, 4T1-hPDL1-MGAT1-KD, 4T1-hPDL1-4NQ-CD73 cells, where the endogenous mouse PD-L1 was replaced with its human counterpart, were orthotopically injected into the left fourth mammary fat pad and allowed to grow to ~100 mm³, followed by injection of PD-L1 antibody durvalumab (10 mg/kg, i.p.) 3 times/week. PBS and IgG were used in the control groups. The tumor growth (a, c, e) of the mice were plotted and the tumor weight were measured (b, d, f). *p<0.05, **p<0.01, ***p<0.001, and ****p<0.0001. For tumor growth statistical analysis, two-way ANOVAs followed by Tukey's multiple comparison tests were performed. Data (means ± SEM) are representative of at least two independent experiments with 5-10 independently analyzed mice per group.

Minor point #1 - - Indicate category legend in fig. 1B

Thank you for your suggestion. We have revised the figure legend for Fig. 1b, c to clearly define the red and blue categories. Specifically, we have labeled the red category as "immune-suppressive" and the blue category as "immune-active."

Updated Fig. 1b, c

Updated Fig. 1b, c: Distribution of single sample GSEA scores of BLBC separated by immune subgroup, using gene sets involved in immune functions (b) and glycosylation (c). P-values are from Wilcoxon tests. * $p < 0.05$, ** $p < 0.01$, *** $p < 0.001$, and **** $p < 0.0001$.

Minor point #2 - Fig. 1G not very convincing data. Please indicate better the number of tumors, and the number of regions per tumor analyzed.

Thank you for your suggestion! Fig. 1g showed only the representative composite image (left panel) with inset (right) of a breast cancer specimen stained by multicolor IHC comprising CD8, Ki67, CD3, MGAT1, PanCK, and DAPI. The number of tumors (n=41) analyzed have now been indicated in the figure legend. The entire region per core specimen of 1 mm in diameter was scanned and analyzed using a Vectra 3 Automated Quantitative Pathology Imaging System

(AKOYA). Based on the spatial analysis from Fig. 1g, we particularly measured the touching events between CD3⁺CD8⁺ T cells and PanCK⁺ tumor cells, or CD3⁺CD8⁺Ki67⁺ T cells and PanCK⁺ tumor cells among MGAT1^{hi} relative to MGAT1^{lo} tumor regions (n=41). We found the heightened interactions between CD8⁺ T cells and MGAT1^{lo} tumor cells, as well as between proliferating CD8⁺Ki67⁺ T cells and MGAT1^{lo} tumor cells, compared to interactions with MGAT1^{hi} tumor cells (Fig. 1h).

Minor point #3 - - *Fig. 1l is not novel since the differences are likely due to high or low CD8 as known in the literature. Show better the supp data fig. 1 with MGAT1 high vs low CD8 infiltration and the prognosis of MGAT1 high and low.*

Thank you for your comment. In Fig. 1i, our analysis highlights that TNBC patients with low MGAT1 expression derive significant survival benefits from high CD8⁺ T cell infiltration (p=0.047). Conversely, this survival advantage is notably diminished in patients with elevated MGAT1 expression (p=0.112), suggesting that high MGAT1 expression may suppress the anti-tumor function of CD8⁺ T cells. To further address the reviewer's concern and provide a clearer representation, we have also replaced Fig. 1i with previous Supplementary Fig. 1i that specifically compares MGAT1-high and MGAT1-low groups prognosis. The updated descriptions have been incorporated into the revised manuscript from page 9 line 174 to line 181: "To further determine the prognostic value of MGAT1 expression in TNBC patients, we analyzed a cohort of 156 TNBC samples from the TCGA database. We observed a strong correlation between elevated MGAT1 expression and negative overall survival outcomes (Fig. 1i). Moreover, when evaluating cumulative survival with respect to both MGAT1 expression levels and the CD8⁺ T cell population, it was evident that patients with a high CD8⁺ T cell presence coupled with reduced MGAT1 expression exhibited a significant survival advantage and patients with low CD8⁺ T cell and high MGAT1 expression were associated with worse prognosis (Supplementary Fig. 2g). This data

compellingly links increased MGAT1 expression to a worse prognosis in the context of immunosuppressive TNBC.”

Updated Fig. 1i

Updated Fig. 1i: i Kaplan-Meier curves of TNBC and BC patients corresponding MGAT1 expression. The high MGAT1 expression leads to poor prognosis in both TNBC and BC patients.

Minor point #4 - Fig. 1F indicate better cell lines per subtype. In addition, authors should discuss why HER2 BC show higher expression of MGAT1.

Thank you for your comment and suggestion to improve the presentation in Fig. 1e, f. In response, we have made the following improvements: (1) To further refine and differentiate MGAT1 expression levels, we included a new set of breast cancer tissue arrays with additional HER2+ and TNBC samples. This expanded dataset allowed us to more comprehensively assess MGAT1 expression across breast cancer subtypes and provided greater statistical power to our analysis. (2) We collected fresh cell samples and repeated the immunoblotting for breast cancer cell lines to generate clearer bands for MGAT1 expression (Fig. 1f). These experiments included normalized quantification to actin for better comparison between samples (Supplementary Fig. 2b). (3) We revised Fig. 1f by incorporating markers and different colors to distinguish the different subtypes of breast cancer cell lines, making the findings easier to interpret and visually clear.

While some TNBC cells have relatively low MGAT1 expression. It is expected because cancer cell lines may not fully capture the complexity of primary tumors, including stromal or immune cell contributions. In addition, TNBCs can be further classified into distinct molecular subtypes, including basal-like 1 (BL1), basal-like 2 (BL2), mesenchymal (M), and luminal androgen receptor (LAR), each of which displays unique biological and molecular characteristics. These subtypes may contribute to the observed variability in MGAT1 expression levels within the TNBC group. Such heterogeneity is expected, as different subtypes may rely on diverse signaling pathways, influencing MGAT1 expression. Nevertheless, TNBC as a whole shows higher MGAT1 expression compared to luminal and Her2+ breast cancer types. This highlights the potential importance of MGAT1 as a biomarker and therapeutic target for TNBC patients with elevated MGAT1 levels.

The updated data and revised manuscript now provide a clearer and more comprehensive assessment of MGAT1 expression in HER2+ and TNBC subtypes. The revised content can be found in the manuscript on page 8, lines 155-158: “Further systematic protein expression analysis across an extensive array of breast cancer cell lines confirmed that MGAT1 protein was prevalent in HER2+ and TNBC cell lines, reinforcing the potential link to immune escape mechanisms in TNBC (Fig. 1f and Supplementary Fig. 2b).”

Updated Fig. 1f

Updated Fig. 1f: f Expression of MGAT1 in normal human mammary epithelial cells and various subtypes of breast cancer cells detected by immunoblotting using anti-MGAT1 antibody.

Supplementary Fig. 2b

Supplementary Fig. 2b: b The immunoblots of MGAT1 in normal human mammary epithelial cells and various subtypes of breast cancer cells were quantified with Image Lab and normalized to actin.

Minor point #5 - Fig. 2B indicate % of tumor cell survival.

We thank reviewer for the suggestion! We have updated the figure labels on all the cancer-immune cell coculture experiments, including Fig. 2b, 2f, and 6m.

Minor point #6 - Fig. 2 should be addressed with an additional TNBC model

We thank reviewer for the suggestion! To strengthen our findings and address the reviewer's suggestion, we have included MDA-MB231, another well-characterized TNBC model, in additional experiments (Supplementary Fig. 3a, b and Fig. 2d). These analyses confirmed that MGAT1 expression similarly impacts cancer cell survival and CD8⁺ T cell activity in MDA-MB231 cells, aligning with the observations made in MDA-MB468 (Fig. 2). The new data, along with the corresponding description, have been incorporated into the revised manuscript from page 10 line 200: "Similar results were observed in MDA-MB231 stable lines (Supplementary Fig. 3a, b)."

Supplementary Fig. 3a, b

Supplementary Fig. 3: Elevated expression of MGAT1 inhibits tumor immune response in 2D and 3D *in vitro* coculture systems. **a** The proliferation of MDA-MB231, MDA-MB231-MGAT1-OE, and MDA-MB231-MGAT1-KD cells (green) in 2D coculture with pre-activated PBMCs was monitored by time lapse image-based quantification. The data are presented as mean \pm SEM from three replicates from a representative experiment. **b** MDA-MB231, MDA-MB231-MGAT1-OE, and MDA-MB231-MGAT1-KD cells were cocultured with pre-activated human PBMCs, and IFN γ ⁺, TNF α ⁺, Granzyme B⁺, Ki-67⁺ CD8⁺ T cell populations were measured and quantified using flow cytometry. * p <0.05, ** p <0.01, *** p <0.001. Data (mean \pm SEM) are representative of at least three independent experiments.

Fig. 2d

d

Fig. 2d : d The infiltration of PBMCs (red) in 3D spheroids constructed with MDA-MB468, MDA-MB468-MGAT1-OE, and MDA-MB468-MGAT1-KD or MDA-MB231, MDA-MB231-MGAT1-OE, and MDA-MB231-MGAT1-KD was visualized with Z-stack imaging using a Lionheart microscope.

Minor point #7 - Fig. 6K should include the analysis CD8 T cell activation with Granzyme B to be consistent with previous figures. Additional activation markers would also reinforce the data such as CD69 or 41BB. Again, coculture tumor cell survival should be shown.

We thank reviewer for the suggestion! To address this, we have added data on Granzyme B expression in Fig. 6I to ensure consistency with previous figures and to provide additional insights into CD8⁺ T cell activation.

Updated Fig. 6l

Updated Fig. 6l: I MDA-MB48 cells were cocultured with PBMC cells and the indicated compounds. Percentages of TNFα⁺, IFNγ⁺, Granzyme B⁺ and Ki-67⁺ in CD8⁺ T cells were measured and quantified using flow cytometry. ***p<0.001, ****p<0.0001. Data (means ± SEM) are representative of at least three independent experiments.

Regarding the inclusion of additional activation markers such as CD69 or 41BB, we acknowledge their importance in further characterizing T cell activation. To maintain consistency with our current flow cytometry analyses, we focused on Granzyme B in this manuscript. However, we are planning a follow-up study that will systematically investigate the function of the leading compounds, incorporating these additional markers to provide a more comprehensive understanding of T cell activation.

Additionally, to address the tumor cell survival in coculture experiments, we have updated Fig. 6m to include the survival of tumor cells cocultured with immune cells in the presence of the leading compounds No.2, No.8 (W-GTF01), and No.9. The new data and related description has been added to the manuscript from page 20 line 468 to line 471: “The CD8⁺ T cell function was detected using flow cytometry (Fig. 6l) and cancer cell survival was examined with image-based viability (Fig. 6m). Based on evaluation standards from multiple layers, W-GTF01 (No.8 compound) was confirmed as a potent inhibitor of MGAT1 activity and stimulator of CD8⁺ IFNγ-producing T-cell response, leading to lowest cancer cell survival.”

Updated Fig. 6m

Updated Fig. 6m: m The survival of MDA-MB468 cells in coculture with pre-activated PBMCs and No.8 was monitored by time-lapse image-based quantification. The data are presented as mean \pm SEM from three replicates of a representative experiment.

Minor point #8 - In line 515, include a header for the section of spatial analysis of clinical samples

We thank reviewer's suggestion! We have added the header "Spatially resolved signatures associated with $MGAT1^{hi_g^h}/CD73^{hi_g^h}$ correlate with unfavorable immune responses in clinical samples"

Minor point #9 - Title change: change "orchestrates immune checkpoints" to "modulated CD73" or similar.

We appreciate your suggestion and have carefully reconsidered our title. The current title is thoughtfully crafted to engage researchers across diverse disciplines, including glycosylation biology, immunology, and cancer therapy. While more focused titles have their merits, we believe the present title achieves an optimal balance between specificity and broad appeal.

Response Letter

Dear Reviewers,

We sincerely appreciate your thorough evaluation of our manuscript, including both the positive feedback and the insightful critiques, technical concerns, and mechanistic questions. We are pleased to report that in this second revised version, we have comprehensively addressed all reviewer concerns and implemented corrections and improvements based on their suggestions to further enhance the quality of our manuscript. These revisions have strengthened our conclusions regarding: (1) Multiomic spatial profiling reveals distinct tumor-immune interactions associated with MGAT1 and CD73, offering insights into mechanisms shaping immune-cold phenotypes. (2) MGAT1-catalyzed complex N-glycan addition to CD73 at the Golgi is essential for its dimerization and membrane translocation, driving immune evasion through elevated adenosine production in immune-cold TNBC. (3) Clinically, the MGAT1^{low}/CD73^{low} signature correlates with favorable immune profiles and better outcomes in breast malignancies, emphasizing its translational significance. (4) W-GTF01, a novel MGAT1 inhibitor, disrupts CD73 dimerization, restores CD8+ T cell function, and sensitizes tumors to anti-PD-L1 therapy. In addition to these experimental advancements, we have revised the manuscript to enhance clarity, address all reviewer comments comprehensively. Below, we provide a detailed *point-by-point* response to each reviewer's comment:

In addition to these experimental advancements, we have refined the manuscript for improved clarity and have carefully addressed all reviewer comments in detail. Below, we provide a point-by-point response to each reviewer's comment.

Reviewer #1: Pages 2-6

Reviewer #4: Pages 7-9

Reviewer #5: Pages 10-17

REVIEWER COMMENTS

Reviewer #1 (Remarks to the Author):

The authors have responded to each of my concerns and added considerable new data to support their conclusions. However, on re-reading the manuscript, I was struck by the awkwardness of the logic. The focus of the manuscript is tumor immune evasion and, while MGAT1 did top the list of molecules to target, the manuscript could just as easily focus on CD73. In fact, CD73 provides a specific focus and mechanism whereas MGAT1 can be replaced by swainsonine or kifunensine, and most probably by the deletion of any of a number of N-glycan branching GlcNAc-transferases. A 2024 paper on CD73 in TNBC and immune evasion must be quoted (J Clin Invest (2024) 10.1172/JCI180914). This paper shows that CD73 levels controlled by ubiquitylation are essential to its functions in tumor immune evasion. The present manuscript makes a strong case that the complex N-glycans on CD73 are essential to its functions in immune evasion, a complementary and elegant story. The authors should rewrite this manuscript in the context of the published JCI paper and any other literature on CD73 in breast cancer. A search for “CD73 TNBC” in Pubmed brings up 27 papers. All literature on CD73 in cancer should be accounted for in the context of the authors’ findings and with respect to the novel role for complex N-glycans on CD73 they have uncovered. The main contribution of the present manuscript is to show that complex N-glycans on CD73 (generated by MGAT1 (or MGAT2, MGAT4, MGAT5)) are important regulators of its role in tumor immune evasion via inducing the dimerization of CD73 and its plasma membrane localization. MGAT1 is in several senses a red herring as a “regulator” and a focus. It is the complex N-glycans themselves that regulate CD73. In trying to simplify the narrative, the authors have instead made it much more difficult for cancer biologists to comprehend in relation to the big picture. It is important that the literature be presented as clearly as possible in this complex area of cancer biology. The title should reflect the central role of CD73 in immune evasion and the importance of its complex N-glycans. Thus “Complex N-glycans on CD73 Regulate Immune Evasion in TNBC” would be an appropriate title. Evidence that complex

N-glycans promote tumor growth and tumor progression/metastasis has been published over the last 30 years, particularly by J. W. Dennis and colleagues. This literature should be appropriately discussed.

Response to Reviewer #1

We appreciate reviewer's comments. We acknowledge your concerns, and we have carefully addressed these issues by refining our narrative, incorporating relevant literature, and clarifying the significance of MGAT1 as a regulator of immune evasion in TNBC in the revised manuscript.

Our goal for this project is to systematically search for potential critical genes/proteins governing tumor immune evasion and progression, especially in triple negative breast cancers (TNBCs). Results from our multiomic analyses initially attracted large attention on immune-suppressive subpopulation characterized by hyperactivation of the N-glycan biosynthesis pathway and further identified MGAT1 as a top-ranked candidate. The subsequent MGAT1 protein complex purification coupled mass spectrometry identified CD73 as a putative downstream substrate. To confirm the biochemical consequence and physiological relevance of MGAT1-CD73 axis in modulating tumor immunogenicity and malignancy, we have comprehensively validated the impact of MGAT1-CD73 axis in immune evasion through mIHC analysis of clinical samples, functional assays in TNBC cell lines, and co-culture experiments with CD8⁺ T cells. Our data demonstrated that MGAT1 overexpression suppresses CD8⁺ T cell function and promotes immune evasion, making MGAT1 a compelling therapeutic target for further study. Further immunoprecipitation, immunofluorescence, and PLA assays confirmed that MGAT1-mediated glycosylation promotes CD73 dimerization and membrane localization, enhancing its immunosuppressive function. While we acknowledge that complex N-glycans are generated through the coordinated action of multiple glycosyltransferases, MGAT1 remains a crucial and specific regulator within this pathway. MGAT1 functions at the initial step of complex N-glycan biosynthesis, making it a pivotal control point for the glycosylation process. Thus, while our study

reveals an important role for CD73 in immune evasion, it fundamentally centers on MGAT1 as the upstream regulator, with CD73 serving as the mechanistic link through which MGAT1 exerts its effects.

Regarding the relationship between our study and the JCI (2024) publication as well as 27 related publications on CD73 in TNBC pointed by the reviewer, we have expanded the discussion section (page 27 line 645 - page 28 line 673) to better contextualize post-translational modifications of CD73 within TNBC immune evasion. “Extensive research has explored the transcriptional and etiological regulation of CD73 at the gene level^{30,67,68}. During malignant tumor progression, CD73 expression is dynamically modulated, particularly in response to hypoxic signaling^{65,69-71}. Beyond HIF-1 α -mediated regulation, CD73 levels in tumor cells are also influenced by Wnt and cyclic AMP (cAMP) signaling, as well as cytokine-driven pathways, highlighting its integration into multiple oncogenic signaling networks⁷²⁻⁷⁵. While significant research has focused on the transcriptional regulation of CD73, limited attention has been given to CD73 protein regulation, especially how CD73 is regulated by post-translational modifications. A recent publication demonstrated that CD73 expression level is regulated through ubiquitylation, which controls its protein stability and turnover, thereby influencing its overall abundance on the tumor cell surface³⁴. In contrast, our study reveals a distinct mechanism that CD73 is tightly regulated by glycosylation regarding its cellular distribution, including its protein synthesis/quality control, trafficking, and membrane translocation. Our fluorescent imaging studies revealed that, once CD73 passed the quality control checkpoint in the endoplasmic reticulum, MGAT1 interacted with CD73 in the Golgi compartment, an essential cellular apparatus allowing protein trafficking to a functional destination. Using molecular docking and simulation coupled with biochemical and cell biological validation, we deciphered the in-depth dynamics of how MGAT1 recognizes CD73, how MGAT1 interacts with the substrate CD73, and how GlcNAc is added to CD73. The functional interference analyses based on the four asparagine residues (glycosylation sites) on CD73 identified by mass spectrometry further revealed the biochemical consequences of MGAT1-

mediated CD73 glycosylation in terms of its membrane translocation, where the replacement of all asparagine residues by glutamine disrupted translocation to the plasma membrane, as demonstrated by biochemical fractionation, fluorescent image, and flow cytometry. This work shows that, before its integration into the membrane, MGAT1-dependent glycosylation of CD73 is a critical step in the medial Golgi, ensuring CD73 membrane-bound enzymatic function in producing adenosine for down-regulating T cell activities. Unlike ubiquitylation, which controls CD73 degradation, N-glycosylation fine-tunes its spatial organization and functional output, underscoring the complexity of its post-translational regulation. By revealing a previously unrecognized role for MGAT1 in CD73 dimerization and immune evasion, our study suggests that disrupting CD73 glycosylation through MGAT1 inhibition may serve as a novel strategy to enhance immunotherapy responses in TNBC patients.”

We also acknowledge the critical contributions of J.W. Dennis and colleagues, who have extensively studied the role of complex N-glycans in tumor progression and immune evasion. To integrate these insights, we have revised our discussion (page 28, lines 681-692) as follows: “Glycosylation is a fundamental post-translational modification that regulates protein folding, stability, localization, protein-protein interactions, and enzymatic activity⁷⁶⁻⁷⁸. Highly conserved across species, glycoconjugates—including O-fucose, O-mannose, N-glycans, mucin-type O-glycans, proteoglycans, and glycosphingolipids—play essential roles in cellular function^{76,77,79,80}. Over the past three decades, studies have demonstrated that complex N-glycans are key drivers of tumor progression, metastasis, and immune evasion. These branched glycans, covalently linked to the membrane and secreted proteins, regulate cell-cell and cell-extracellular matrix (ECM) interactions, facilitating tumor proliferation, invasion, and immune escape⁸¹⁻⁸⁵. Aberrant N-glycosylation has been shown to impair immune surveillance, allowing tumor cells to evade recognition and destruction by the immune system^{34,35,37,78,86}. Given their critical role in tumor biology, understanding the mechanisms that regulate complex N-glycan biosynthesis has become

a major focus in cancer research, particularly for identifying novel therapeutic targets in cancer immunotherapy.”

Based on the suggestion from the reviewer, we now changed the title of our manuscript to:

"MGAT1-Guided Complex N-Glycans on CD73 Regulate Immune Evasion in TNBC"

Reviewer #4 (Remarks to the Author):

The authors have addressed all my previous comments. However, I regret to note that the first author has graduated and taken a new position, making glycoproteomics experiments unavailable. Instead, the authors performed PNGase F and Endo H digestions to investigate the glycosylation status of CD73 under MGAT-KO and OE conditions. They also conducted point mutations to validate the presence of complex-type N-glycans at four potential glycosylation sites.

One further comment: the quality of the mass spectra in Figure 3a is still poor. The authors should re-plot these spectra to improve clarity.

Response to Reviewer #4

We appreciate reviewer's recognition of the efforts we have made to address the previous concerns. Regarding your comment on the quality of the mass spectra in Fig. 3a, we have taken the following steps to improve clarity and presentation: (1) We have replaced the mass spectra in Fig. 3a, ensuring improved resolution and clarity for better visualization. The updated figure provides a more distinct representation of the key peaks. (2) To facilitate a more comprehensive assessment, we have included enlarged versions of the spectra in Supplementary Fig. 4a which provide a closer view of the relevant details.

Updated Fig. 3a

Fig. 3a: a MGAT1 complex was purified with a tandem-affinity purification protocol followed by mass spectrometry analysis in MDA-MB468-Flag/HA-MGAT1 cells. Silver staining of the purified MGAT1 complex is illustrated. CD73 was identified as a binding partner of MGAT1, and the representative spectra are shown.

Supplementary Fig. 4a

Supplementary Fig. 4a: a Representative peptides of CD73 and MGAT1 discovered by mass spectrometry.

Reviewer #5 (Remarks to the Author):

The authors have addressed many of the concerns raised by this reviewer and others. As highlighted in my previous revision, a key novelty of this study is the development of an inhibitor targeting MGAT1-CD73-mediated immunosuppression. Part of the newly provided data reinforce this message; however, some minor concerns still persist.

Response to Reviewer #5

Question #1 - *Based on the new data presented in Figure 1, which compare this mechanism across various BC subtypes, the authors should clarify whether the mechanism studied is specific to TNBC. It does not appear to be significantly enriched in immune-cold tumors, such as ER+ BC, thus this distinction should be explicitly stated.*

We thank the reviewer's comment. In general, ER+ breast cancers, including Luminal A and Luminal B subtypes, are considered the "coldest" of all breast cancer subtypes due to their low tumor mutation burden (TMB), low levels of immune cell infiltration, and reduced expression of immunogenic neoantigens (Onkar et al. *Cancer discovery*. 2023. doi:10.1158/2159-8290.CD-22-0475). These tumors lack sufficient immune stimulatory signals to recruit effector T cells, rendering them largely unresponsive to immunotherapy. Specifically, ER+ tumors exhibit low PD-L1 expression, minimal CD8⁺ T cell infiltration, and a predominance of immunosuppressive elements such as regulatory T cells (Tregs) and tumor-associated macrophages (TAMs), which contribute to their immune-excluded phenotype (Wagner et al. *Cell*. 2019. doi: 10.1016/j.cell.2019.03.005). Additionally, estrogen signaling itself has been shown to play a direct role in suppressing immune activation by modulating cytokine production and antigen presentation pathways. In contrast, TNBC is the most immunogenic breast cancer subtype, characterized by the highest TMB, increased infiltration of immune cells (CD8⁺ T cells, CD4⁺ T cells, Tregs, and myeloid-derived suppressor cells (MDSCs)), and significant expression of

immune checkpoint molecules such as PD-L1, B7-H4, and CD73. These features make TNBC the most responsive to immunotherapy among all breast cancer subtypes.

To explicitly clarify the distinction between TNBC and ER+ breast cancers regarding MGAT1 and CD73 regulation, we have revised the Discussion section (page 29, lines 707–722) as follows:

“Our data indicate that the MGAT1-CD73 axis is not significantly enriched in ER+ breast cancers, likely due to the distinct immune landscape and immune evasion strategies of these tumors. ER+ tumors exhibit poor T cell infiltration and low antigen presentation, resulting in minimal selective pressure for cancer cells to upregulate immunosuppressive molecules such as CD73. Consequently, the necessity for MGAT1-mediated glycosylation to enhance CD73 function is reduced. Additionally, emerging evidence suggests that estrogen receptor signaling suppresses glycosylation-associated pathways, including MGAT1 regulation, further limiting the role of this axis in ER+ tumors. In contrast, TNBC lacks ER signaling and relies on alternative oncogenic pathways that promote glycosylation changes, enhancing CD73-mediated immunosuppression. While CD73 plays a crucial role in shaping the immunosuppressive tumor microenvironment in TNBC, ER+ tumors predominantly evade immune attack through metabolic and hormonal crosstalk rather than direct immune suppression via checkpoint molecules like CD73. These findings underscore the specificity of MGAT1-mediated CD73 upregulation in TNBC, where immune evasion is driven by high checkpoint expression and adenosine-mediated suppression, whereas ER+ breast cancers employ distinct, non-CD73-dependent immune escape mechanisms. This distinction highlights the need to develop therapeutic strategies tailored to the immune landscape of each breast cancer subtype.”

Question #2 - *As previously requested, the authors should elaborate on the criteria used to select CD73 from among other interacting partners. Additionally, they should reference the table containing the list of proteins in the main text and provide examples of other immunosuppressive*

proteins, since it is mentioned in the text. The table should also include FDR values, as CD73 is listed at position 226.

We appreciate the reviewer's comments. To address the concerns, we have now explicitly referenced the table of potential interacting proteins in the main text and included examples of other immune-related proteins identified in our analysis, such as antigen-presenting molecules HLA-A and HLA-B. Among the potential MGAT1 interactors identified through mass spectrometry, we prioritized candidates based on their functional relevance to immune regulation, reproducibility across experiments, and ranking in the interactome analysis. While multiple immune-related proteins were identified, CD73 was the only immune-suppressive protein found in our purification and mass spectrometry (MS) analysis, making it a strong candidate for further investigation.

The revised text on page 10, lines 222-227 states: “The interactome of MGAT1, resolved through mass spectrometry, identified several tumor immune-responsive proteins, including antigen-presenting proteins such as HLA-A and HLA-B, as well as CD73 (Table 1). Among the potential MGAT1 binding partners, our analysis identified several antigen-presenting proteins (e.g., HLA-A and HLA-B) but only one key immune-suppressive molecule—CD73. Given its well-established role in adenosine-mediated immunosuppression and tumor immune evasion, we prioritized CD73 as the primary focus for mechanistic investigation.”

In addition, to further enhance clarity and rigor, we have now included experimental q-values (FDR thresholds) in the table. The Exp. q-value represents the minimum FDR at which a given protein is considered a significant interactor. In our dataset: Proteins with q-values <0.01 are regarded as high-confidence hits, Proteins with q-values <0.05 are considered medium-confidence hits, CD73 had an Exp. q-value of 0, indicating a high-confidence interaction.

Question #3 - *The new data in Supplementary Fig. 4e should be integrated into Fig. 3H. The functional assays lack a CD73-KD condition, which is necessary to evaluate whether the moderate changes in adenosine concentration result in functional effects.*

We appreciate the reviewer's suggestions. As recommended, we have now integrated the new data previously shown in Supplementary Fig. 4e into Fig. 3h to consolidate the findings and enhance clarity.

Updated Fig. 3h

h

Fig. 3h: h Adenosine levels were determined in WT or CD73-KD MDA-MB231/MDA-MB468 breast cancer cells with MGAT1 OE or MGAT1 KD. **, ***, **** $p < 0.0001$. Data (mean \pm SEM) are representative of at least three independent experiments.

Question #4 - The new data presented in Supplementary Fig. 9g should be incorporated into the main figures for better visibility and emphasis.

We appreciate the reviewer's suggestion to enhance the visibility and emphasis of the new data presented in Supplementary Fig. 9g. As recommended, we have now incorporated this data into the main figures as Fig. 6q.

Updated Fig. 6p-r

Fig. 6p-r: **p** The survival of MDA-MB468 cells in coculture with pre-activated PBMCs and No.2/No.8(W-GTF01)/No.9 was monitored by time lapse image-based quantification. The data are presented as mean \pm SEM from three replicates of a representative experiment. **q** The survival of WT or CD73-KD MDA-MB468 cells in coculture with pre-activated PBMCs and No.8 was monitored by time lapse image-based quantification. The data are presented as mean \pm SEM from three replicates of a representative experiment. **r** Binding of compound W-GTF01 (MolPort-004-851-686) to human MGAT1. Notably, π -stacking interaction (marked by blue transparent oval) is formed between F324 (yellow stick) and W-GTF01 (magenta stick).

Question #5 - *The combined T-cell state populations for different CX3CR1^{+/−} vs. CD101^{+/−} combinations should be shown. These results appear inconsistent with the authors' claims, as CX3CR1 is not a marker of exhaustion and is shown to be higher with MGAT1-OE.*

We appreciate the reviewer's careful assessment and the opportunity to clarify and improve our data presentation. We apologize for mislabeling "PD-1⁺CD101⁺" as "CX3CR1⁺CD101⁺" in the original Fig. 7f, and this has now been corrected. To further address this concern, we have incorporated newly analyzed results (now presented as Supplementary Fig. 10d) that specifically compare different CX3CR1^{+/−} vs. CD101^{+/−} combinations, providing additional validation and context for the role of CX3CR1 and CD101 expression in CD8⁺ T cell states under MGAT1-OE conditions. Additionally, we have updated the gating strategy in Supplementary Fig. 14 to ensure the proper identification of CX3CR1^{+/−} and CD101^{+/−} subpopulations, aligning with standard immunophenotyping methodologies. The updated text (page 21 line 518 to page 22 line 523) states as "Furthermore, the fractions of PD-1⁺TOX⁺ and PD-1⁺CD101⁺ CD8⁺ cells were significantly increased in MGAT1 OE tumors (Fig. 7f). Consistent with the Fig. 7f, our additional

analysis in Supplementary Fig. 10d further supports the observation that MGAT1-OE tumors exhibit an increased percentage of exhausted-like CX3CR1-CD101⁺ CD8⁺ T cell populations compared to control tumors, reinforcing the impact of MGAT1 overexpression on T cell exhaustion and immunosuppressive mechanisms in the tumor microenvironment.”

Updated Fig. 7f

Fig. 7f: f Comparison of the percentage of PD-1⁺TOX⁺ CD8⁺ and PD-1⁺CD101⁺ CD8⁺ T cells among live CD45⁺ between 4T1 control tumors and 4T1 MGAT1 OE tumors.

Updated Supplementary Fig. 10d

Supplementary Fig. 10d: d Comparison of the percentage of CX3CR1⁺/CD101⁻, CX3CR1⁺/CD101⁺ and CX3CR1⁻/CD101⁺ CD8⁺ T cells between 4T1 control tumors and 4T1 MGAT1 OE tumors.

Updated Supplementary Fig. 14

Supplementary Fig. 14. Gating strategy. a Gating strategies for the Cytek spectral flow cytometry data analysis in tumor infiltrated lymphocytes.

Question #6 - *Statistical analysis is missing in Fig. 7q and should be included.*

We appreciate the reviewer's comments. To address this, we have now included the statistical analysis in Figure 7Q. Statistical significance has been determined using Log-rank (Mantel-Cox) test, and the corresponding p-values are now indicated in the figure.

Updated Fig. 7q

q

Fig. 7q. q 4T1-hPD-L1 cells, where the endogenous mouse PD-L1 was replaced with its human counterpart, were orthotopically injected into the left fourth mammary fat pad and allowed to grow to ~100 mm³, followed by injection of W-GTF01 (10 mg/kg, i.p.) two times/week and PD-L1 antibody durvalumab (10 mg/kg, i.p.) 3 times/week. PBS and IgG were used in the control groups. The tumor growth (p) and the survival (q) of the mice were plotted. *p<0.05, **p<0.01, ***p<0.001, and ****p<0.0001. For tumor growth statistical analysis, two-way ANOVAs followed by Tukey's multiple comparison tests were performed. Data (means ± SEM) are representative of at least two independent experiments with 5-10 independently analyzed mice per group.

Question #7 - *Finally, the title of the study overstates its findings. This could lead to confusion regarding the scope of immune checkpoints tested, which have not been tested in the study.*

We appreciate the reviewer's suggestion regarding the study title. To better align the title with the scope and findings of our research, we have revised it to "MGAT1-Guided Complex N-Glycans on CD73 Regulate Immune Evasion in TNBC".

Dear Reviewers,

We greatly appreciate the opportunity to publish our article in *Nature Communications*. In this revised version, we have carefully addressed all concerns from the editor and reviewers, making necessary corrections and improvements in accordance with the editor's guidelines.

We sincerely thank all the reviewers for their thoughtful feedback and for acknowledging the significance of our manuscript.

Reviewer #1 (Remarks to the Author):

The authors have addressed my comments appropriately.

Reviewer #2 (Remarks to the Author)

The authors addressed all point raised, and did good work on the revised manuscript.

Reviewer #3 (Remarks to the Author)

I thank the authors for addressing my remarks and concerns.

Reviewer #4 (Remarks to the Author):

The authors have addressed all my comments.

Reviewer #5 (Remarks to the Author):

The authors have addressed all my concerns. The manuscript has been improved.